# Forking-Sequences: Statistically and Computationally Efficient Multi-Horizon Forecasting with Reduced Volatility

**Willa Potosnak**
SCOT Forecasting Science, Amazon
Auton Lab, School of Computer Science, Carnegie Mellon University          `wpotosna@andrew.cmu.edu`

**Malcolm Wolff**
SCOT Forecasting Science, Amazon          `wolfmalc@amazon.com`

**Mengfei Cao**
SCOT Forecasting Science, Amazon          `mfcao@amazon.com`

**Ruijun Ma**
SCOT Forecasting Science, Amazon          `ruijunma@amazon.com`

**Tatiana Konstantinova**
SCOT Forecasting Science, Amazon          `tkonst@amazon.com`

**Dmitry Efimov**
SCOT Forecasting Science, Amazon          `defimov@amazon.com`

**Michael W. Mahoney**
SCOT Forecasting Science, Amazon          `zmahmich@amazon.com`

**Boris Oreshkin**
SCOT Forecasting Science, Amazon          `oreshkin@amazon.com`

**Kin G. Olivares**
SCOT Forecasting Science, Amazon          `kigutie@amazon.com`

**Reviewed on OpenReview:** `https://openreview.net/forum?id=dXdycy7WCX`

## Abstract

While accuracy is a critical requirement for time series forecasting, an equally important desideratum is reasonable forecast volatility across forecast creation dates (FCDs). Even highly accurate models can produce erratic revisions between FCDs, undermining trust and disrupting downstream decision-making. To improve the volatility of forecast revisions, state-of-the-art models like `MQCNN`, `MQT`, and `SPADE` employ a powerful yet underexplored neural network architectural design: *forking-sequences*. This architectural design jointly encodes and decodes the entire time series across all FCDs, producing an entire multi-horizon forecast grid in a single forward pass. This approach contrasts with conventional neural forecasting methods that process FCDs independently, generating only a single multi-horizon forecast per forward pass. In this work, we formalize the forking-sequences design and motivate its broader adoption by introducing a metric for quantifying excess volatility in forecast revisions and by providing theoretical and empirical analysis. We theoretically motivate three key benefits of forking-sequences: (i) reduced forecast volatility through ensembling; (ii) gradient variance reduction, improving the statistical efficiency of the training procedure; and (iii) improved inference computational efficiency. We validate the benefits of forking-sequences compared to baseline window-sampling on the M-series benchmark, using 16 datasets from the M1, M3, M4, and Tourism competitions. We observe median sCRPS improvements across datasets of 46.2%, 49.3%, 28.6%, 24.7%, and 6.4% for `RNN`, `LSTM`, `CNN`, `Transformer`, and `State Space`-based architectures, respectively. We then show that forecast ensembling during inference can reduce median forecast volatility by 13.2%, 13.0%, 10.9%, 10.2%, and 11.2% for these respective models trained with forking-sequences, while maintaining accuracy.

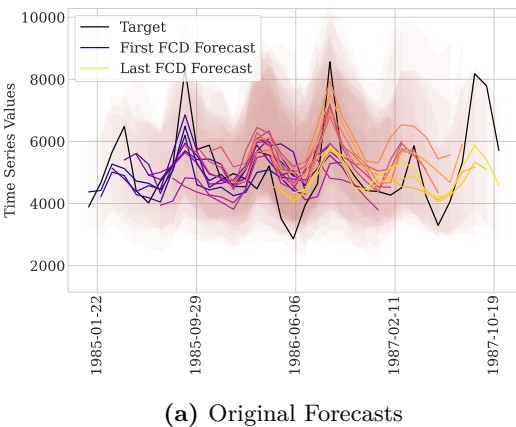
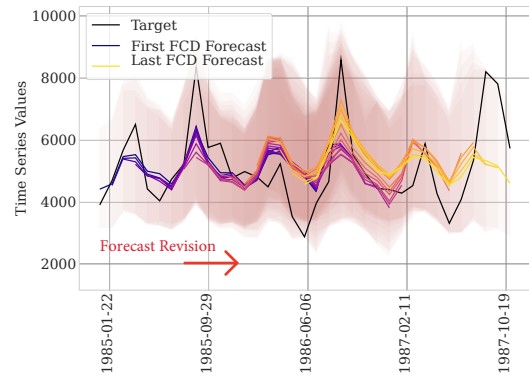

**(a)** Original Forecasts

**(b)** Forking-sequence Ensembled Forecasts

**Figure 1:** Forecast comparison for a series from the `M1` dataset (Makridakis et al., 1982). (a) Forecasts generated without the forking-sequences ensemble. (b) Forecasts with the forking-sequences ensemble applied. Augmenting the model with the forking-sequences ensemble significantly reduces forecast variability across FCDs, resulting in more stable and consistent forecast distributions. We marked the direction of forecast revisions in red. The lines show P50 (median) forecasts across different FCDs. By reusing encoder computations, forking-sequences enables computationally efficient ensembling with negligible additional computational cost.

## 1 Introduction

Forecasting plays an important role in a wide range of domains, including energy systems (Hong et al., 2016), finance (Maung and Swanson, 2025), economics (Elliott and Timmermann, 2008), product supply chain (Wen et al., 2017; Eisenach et al., 2021; Wolff et al., 2024), and healthcare analytics (Nikolopoulos et al., 2021; Potosnak et al., 2025). It typically serves as a foundation for predicting uncertain events, enabling informed decision-making in various scenarios, including demand anticipation, production scheduling, and resource allocation. To meet diverse operational needs and reduce the accumulation of forecast errors, multi-horizon forecasting tools have become the default, offering improved support for short, medium, and long-term planning. However, as these systems operate over time, they generate multiple overlapping forecasts for each target date. This sequence of revisions raises considerations regarding the consistency of these forecasts, which we refer to as *forecast volatility*.

The balance between forecast accuracy (of a single prediction) and forecast volatility (as predictions are updated) presents a new challenge: while new temporal information should naturally refine predictions, excessive fluctuations in forecasts can complicate operational planning and decision-making. For example, electrical grid operators rely on load forecasts to determine the power supply. In the event of an upcoming intense heat wave, forecast revisions from 45 to 65 GW reflect a useful revision that enables preparations such as activating reserve plants. Conversely, if forecasts are overly stable (i.e., not revised), it can result in inadequate power supply, where insufficient power when needed can cause preventable blackouts affecting large populations. Forecast volatility raises two research questions: 1) how can we measure forecast volatility in a way that captures benign revisions while identifying problematic revisions?; and 2) are there neural forecasting architectural designs that can reduce forecast volatility, with minimal degradation or even improvement in forecast accuracy?"

In this paper, we formally introduce *forking-sequences*, an encoder-agnostic network architectural design. The forking-sequences design offers two distinct use paradigms: during training, it augments loss computation with additional samples, leading to reduced gradient variance and better preservation of gradients for early time steps in recurrent architectures; and during inference, it enables efficient ensemble forecasting across multiple FCDs, reducing forecast volatility.

Our main contributions are as follows:

(i) **Forking-Sequences Statistical Efficiency (Section 3.3).** We introduce forking-sequences and formally analyze its effects on gradient quality. We show that, under reasonable temporal correlation assumptions, the gradient variance decreases linearly with the number of FCDs ($\mathcal{O}(\frac{1}{T})$), echoing results

from the weak law of large numbers), and its signal-to-noise ratio (SNR) improves linearly with the number of FCDs. These gradient improvements lead to more stable training steps, which in turn accelerates convergence in optimization, in both convex and nonconvex loss landscapes. We empirically demonstrate that these optimization benefits also translate into improved generalization.

(ii) **Forking-Sequences Computational Efficiency (Section 3.4).** We show that forking-sequences enable an order of magnitude improved computational regime for encoder-decoder architectures by supporting efficient inference across FCDs in a single forward pass. Unlike standard approaches that recompute the encoder for each FCD, forking-sequences reuse encoder outputs across all FCDs, significantly reducing redundant computation. In the unrestricted context setting, inference complexity improves from quadratic to linear ($\mathcal{O}(T^2)$ to $\mathcal{O}(T)$). In the restricted setting, where each FCD uses an $L$-length context, complexity improves by a factor of $L$ ($\mathcal{O}(LT)$ to $\mathcal{O}(T)$).

(iii) **Forking-Sequences Ensembling (Section 3.5).** By generating multi-horizon forecasts for all FCDs in a single forward pass, forking-sequences naturally produce overlapping forecasts for each target date. We show that forecast ensembling during inference can reduce forecast volatility compared to without ensembling for all encoders, with median percentage improvements of 10.8%, 13.2%, 13.0%, 10.9%, 10.2%, and 11.2% for RNN, LSTM, CNN, Transformer, and State Space-based architectures, respectively, while maintaining accuracy. We additionally show that forecast ensembling can be used in zero-shot inference tasks with pretrained time series foundation models (TSFMs), demonstrating median reduction in forecast volatility of approximately 10% with less than 0.1% degradation in forecast accuracy for Chronos-2, Toto 2.0, TimesFM, pretrained PatchTST, and pretrained NBEATS.

(iv) **Novel Forecast Volatility Metrics (Section 3.6).** We introduce two complementary metrics to assess forecast volatility across FCDs: the *Symmetric Forecast Percentage Change* (sFPC), which measures raw point forecasts' revision magnitude independently of ground truth, and *Scaled Excess Volatility* (sEV), that retroactively measures volatility of probabilistic forecasts, distinguishing accuracy-improving revisions from detrimental ones.

(v) **Forking-Sequences Empirical Validation (Section 4.2).** We evaluate both accuracy and forecast volatility across FCDs, comparing forking-sequences against the standard window-sampling approach on the M-series benchmark: 16 large-scale datasets from the Tourism competition (Athanasopoulos et al., 2011) and the M-forecast competitions (Makridakis et al., 1982; Makridakis and Hibon, 2000; Makridakis et al., 2020). We observe that forking-sequences yield improvements in sCRPS over window-sampling for all encoders, though with varied magnitude, with median percentage improvements across datasets of 46.2%, 49.3%, 28.6%, 24.7%, and 6.4% for RNN, LSTM, CNN, Transformer, and State Space-based architectures, respectively.

The remainder of this paper is organized as follows: Section 2 explores related work and relevant literature. Section 3 introduces the architectural design of forking-sequences and its properties. Section 4 presents the main empirical results. Finally, Section 5 discusses our key findings, and outlines future research directions.

## 2 Related Work

Ideas similar to forking-sequences have been present in the forecasting literature for decades, despite not inspiring neural network architectural design or training. Most notably, Hart (1994) introduced time-series cross-validation as a model selection technique that accounts for temporal dynamics and data dependencies, addressing the limitations of classical cross-validation; this approach assumes independent FCD observations, and forking-sequences bears several similarities with it (Bergmeir and Benítez, 2012). (Figure 7 provides a depiction of the temporal cross-validation evaluation.)

More recently, forking-sequences has informed the design of MQForecaster neural networks for industrial applications, with notable examples including MQCNN, MQT, and SPADE (Wen et al., 2017; Olivares et al., 2023; Eisenach et al., 2021; Wolff et al., 2024). However, this prior work largely adopted forking-sequences as a practical heuristic, without providing formal presentation, theoretical justification, or systematic empirical validation of its benefits. In some cases, its theoretical motivation was attributed to the martingale properties

of forecast revisions (Chen et al., 2022; Foster and Stine, 2021), rather than its simpler and more natural role in reducing gradient variance during training.

Beyond forecasting; in the context of healthcare and natural language processing (NLP), the concept of *target replication* has been considered to train LSTM time series classifiers for medical diagnosis (Lipton et al., 2016); and a similar scheme to train text document classifiers has also been considered (Dai and Le, 2015). We distinguish target replication and the standard *teacher forcing* technique (Williams and Zipser, 1989).

The first method for training a recurrent network in a fully online, continuous manner was the Real-Time Recurrent Learning (RTRL) algorithm, which updates the network's weights at every time step rather than waiting for the full sequence to finish (Williams and Zipser, 1989). Williams and Zipser later coined the term *teacher forcing* to describe the practice of feeding ground-truth inputs to the recurrent network during training, instead of using its own predictions. This approach improved the training stability of models trained with backpropagation through time by reducing the accumulation of prediction errors across time steps.

The *teacher forcing* method, most notably through the next-token prediction objective (Bengio et al., 2003), is an autoregressive learning strategy which trains models to predict the next token given previous ones. Architectures that reuse computation to generate the dense supervision signal in a single forward pass, laid the foundation for large language models like GPT-1 and GPT-3 (Radford et al., 2018; Brown et al., 2020).

A key difference between the standard *teacher-forcing* technique used in NLP autoregressive models and the *forking-sequences* approach used in neural forecasting is the implicit target replication that arises in multi-horizon forecasts. Whereas NLP models are typically trained to predict only the next token, forecasting models must predict longer future trajectories, generating multiple future steps for each FCD. This temporal overlap in targets is precisely what exacerbates the need for forecast volatility metrics and evaluations.

Forecast volatility differs from the notion of forecast degradation. While multi-horizon forecasts do generally experience gradual degradation over time steps, forecast volatility refers specifically to consistency across overlapping forecasts for the same target date. Neural forecasting has focused almost exclusively on improving predictive accuracy without concern for forecast volatility. The few works that address volatility typically rely on specialized loss (Belle et al., 2023), dynamic loss weighting algorithms (Caljon et al., 2025), or post-processing techniques such as linear interpolation (Belle et al., 2023) that require additional parameters and processing steps. In contrast, *forking-sequences* offers a much simpler way to efficiently ensemble forecasts for overlapping target dates generated across multiple FCDs in a single forward pass, without modifying the loss or neural architecture.

## 3 Methodology

### 3.1 Multi-Horizon Forecasting

We start by introducing the general multi-horizon forecasting task notation. Let the FCDs collection be $[t] = [1, \ldots, T]$ and the forecast horizon be denoted by $[h] = [1, \ldots, H]$. The target random variables are

$$\text{single target} \quad \mathbf{Y}_{t,[h]} = [Y_{t+1}, \ldots, Y_{t+H}] \in \mathbb{R}^H, \tag{1}$$

$$\text{and grid target} \quad \mathbf{Y}_{[t][h]} \in \mathbb{R}^{T \times H}. \tag{2}$$

The forecasting task that we tackle aims to estimate the following conditional distribution:

$$\mathbb{P}\left(\mathbf{Y}_{t,[h]} \mid \boldsymbol{\theta},\ \mathbf{y}_{[t]}\right), \tag{3}$$

where $\boldsymbol{\theta}$ denotes the model parameters and $\mathbf{y}_{[t]} = [y_1, y_2, \ldots, y_t]$ denotes the autoregressive realizations until $t$. In our work, we denote the forecasts for the target variable $\mathbf{Y}_{t,[h]}$ as $\hat{\mathbf{Y}}_{\mathbf{t},[\mathbf{h}]}$.

**Forecast revisions.** Our work focuses on multi-horizon forecasting across FCDs ($H > 1$). This setting is essential for studying forecasts' volatility. In single-horizon forecasting, each prediction corresponds to a unique future time point, so forecasts never overlap, and revisions cannot occur. In contrast, multi-horizon

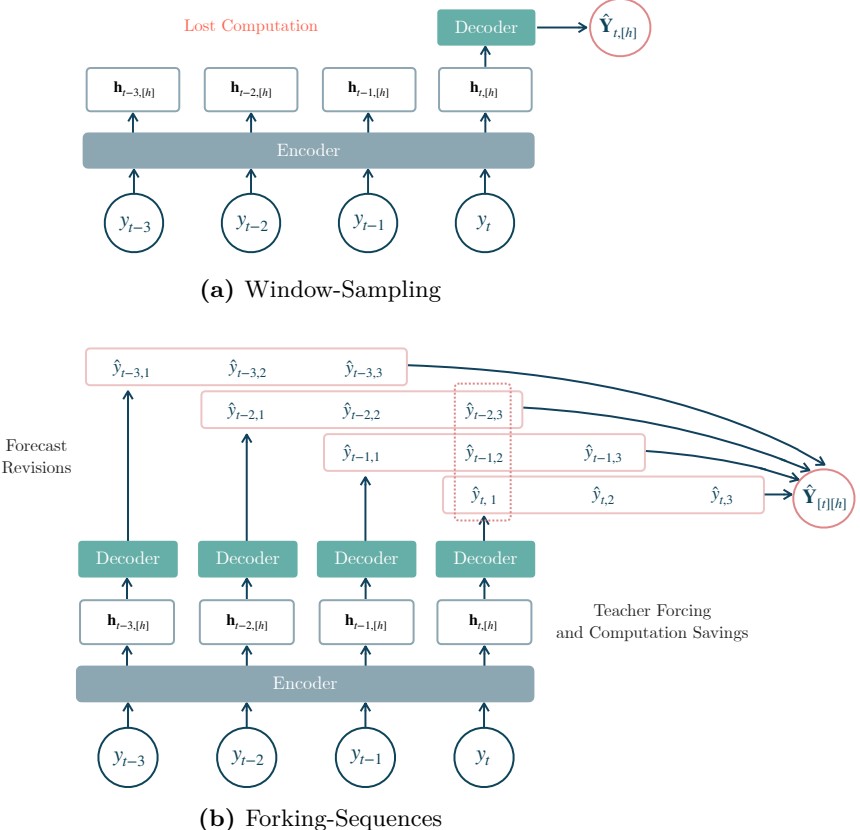

**(a)** Window-Sampling

**(b)** Forking-Sequences

**Figure 2:** (a) A *window-sampling* model, where its encoder outputs a single hidden state $\mathbf{h}_{t,[h]}$ (blue rectangles) for each input sequence $\mathbf{y}_{[t]}$. Blue rectangles indicate encoded series, green rectangles denote the decoders, and the red circle denotes the prediction target. (b) A *forking-sequences* model, where its encoder outputs hidden states $\mathbf{h}_{[t][h]}$, reusing computations from $\mathbf{h}_{[t-1][h]}$. During training, forking-sequences collects multi-horizon errors from all intermediate FCDs. Additionally, decoded forecasts across FCDs can be easily ensembled (dotted lines), saving most of the encoding computational costs and reducing forecast volatility across forecast revisions.

forecasting produces overlapping predictions: the same target value is forecast multiple times at successive FCDs as new information arrives. This naturally forms a sequence of revisions for each target value. Formally,

$$\text{For the target random variable} \quad Y_{t+h}, \quad \hat{\mathbf{Y}}_{t+1,h} \quad \text{is a revision of} \quad \hat{\mathbf{Y}}_{t,h+1}. \tag{4}$$

**Parameter Learning.** To tackle the multi-horizon forecasting task, we learn neural network parameters $\boldsymbol{\theta}$ by minimizing the empirical expectation of the Multi Quantile Loss (QL; (Koenker and Bassett, 1978)),

$$\hat{\boldsymbol{\theta}} := \operatorname*{arg\,min}_{\boldsymbol{\theta}\in\Theta} \hat{\mathbb{E}}\left[ \frac{1}{|Q|} \sum_{[q]\in Q} \text{QL}_q(Y, \hat{Y}^{(q)}(\boldsymbol{\theta})) \right], \tag{5}$$

where QL is the quantile loss for a given quantile level $q \in \mathcal{Q} = \{0.1, 0.2, \ldots, 0.9\}$

$$\text{QL}_q(y, \hat{y}^{(q)}) = q(y - \hat{y}^{(q)})_+ + (1 - q)(\hat{y}^{(q)} - y)_+. \tag{6}$$

### 3.2 Forking-Sequences Architecture Design

Most neural forecasting models use a standard (and intuitive) architectural design known as *window-sampling* (Oreshkin et al., 2020; Challu et al., 2023; Nie et al., 2023; Salinas et al., 2020; Lim et al., 2021;

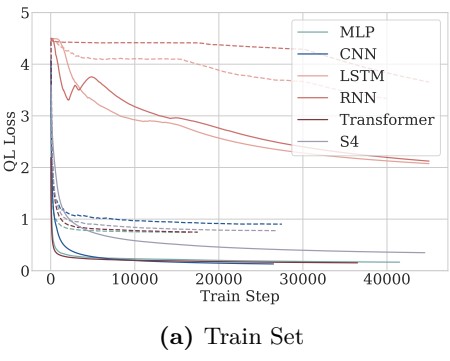
**(a)** Train Set

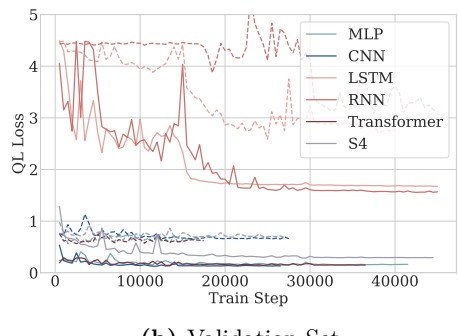
**(b)** Validation Set

**Figure 3:** MQForecaster model optimization convergence on the M4 hourly data, using either *forking-sequences* (solid) or *window-sampling* (dashed) techniques. Quantile loss on the **(a)** train set and **(b)** validation set as a function of train steps are consistently lower for forking-sequences models. Across architectures forking sequences show validation quantile loss improvements. Quantile loss for the train set of other frequency-specific datasets is shown in Fig. 14.

Das et al., 2024; Zhou et al., 2020; Ansari et al., 2024; Woo et al., 2024; Garza and Mergenthaler-Canseco, 2023). The window-sampling approach segments the series into windows of size $L$; and windows are treated independently across FCDs. This approach is illustrated in Figure 2a, and it is formalized in Equation 7:

$$\mathbf{h}_{t,[h]} = \text{Encoder}\left(\mathbf{Y}_{[t]}\right) \qquad \text{and} \qquad \hat{\mathbf{Y}}_{t,[h]} = \text{Decoder}\left(\mathbf{h}_{t,[h]}\right). \tag{7}$$

The Encoder in Eqn. 7 outputs a hidden state for each position in the forecast horizon, with the full hidden state tensor having shape $B \times T \times H \times D$, where D denote the embedding dimension. We use $[h] = [1, \ldots, H]$ to index the embedding corresponding to each horizon step $h$.

In contrast, *forking-sequences* is a neural architectural design that jointly encodes and decodes the time series across all FCDs. Specifically, the model computes a single hidden representation $\mathbf{h}_{[t][h]}$ that is then decoded into forecasts $\hat{\mathbf{y}}_{[t][h]}$ for all FCDs $[1, \ldots, T]$. This approach is illustrated in Figure 2b, and it is formalized in Equation 8:

$$\mathbf{h}_{[t][h]} = \text{Encoder}\left(\mathbf{Y}_{[t]}\right) \qquad \text{and} \qquad \hat{\mathbf{Y}}_{[t][h]} = \text{Decoders}\left(\mathbf{h}_{[t][h]}\right). \tag{8}$$

### 3.3 Forking-Sequences Gradient Improvements

In this subsection, we analyze how the *forking-sequences* design influences parameter learning. We theoretically motivate the design by showing that leveraging each forecast across multiple FCDs reduces the variance of the gradient estimator in a manner analogous to the statistical efficiency gains obtained from full autoregressive factorization in *teacher-forcing* (Williams and Zipser, 1989; Bengio et al., 2003). Beyond variance reduction, we demonstrate that forking-sequences provide denser supervision signals, which accelerate learning, improve convergence, and mitigate vanishing-gradient effects across a range of encoder architectures.

During training, the gradients are computed with respect to the model parameters for SGD updates to minimize the loss in Equation 5. The gradient is computed on a $B$-length minibatch. With a window-sampling architecture, multi-horizon losses are gathered in a batch $\mathcal{B}$ for individual FCDs:

$$\nabla\mathcal{L} = \frac{1}{B \times H} \sum_{b=1}^{B} \sum_{h=1}^{H} \nabla\text{QL}\left(y_{b,h}, \ \hat{y}_{b,h}\right). \tag{9}$$

Using a forking-sequences design, multi-horizon losses are gathered across all FCDs:

$$\nabla \mathcal{L}_T = \frac{1}{B \times T \times H} \sum_{b=1}^{B} \sum_{t=1}^{T} \sum_{h=1}^{H} \nabla \text{QL} \left( y_{b,t,h}, \ \hat{y}_{b,t,h} \right). \tag{10}$$

We include additional details on the difference between forking-sequences and window-sampling optimization in Appendix A. The proof of the Theorem 1 result is included in Appendix B.

**Theorem 1.** *(Forking-Sequences SNR Gains under $M$-Dependence) Consider the forking-sequence gradient estimator from Equation 10. If the $T$ gradient samples are $M$-dependent (Shumway and Stoffer, 2017), the estimator's variance decreases linearly $\mathcal{O}\left(1/T\right)$ and its Signal to Noise Ratio (SNR) grows linearly $\mathcal{O}\left(T\right)$.*

Forking-sequences improves the gradient SNR by multiplying the number of effective training signals extracted from each series. As with *teacher forcing* in autoregressive language models, which turns one text sequence into many next-token supervision events, forking-sequences converts a time series into a full set of multi-horizon targets, multiplying available supervision at a negligible computational cost. This data augmentation has a twofold effect. First, model training with forking-sequences reduces the variance of the stochastic gradient around its mean at a linear rate ($\mathcal{O}(1/T)$), with FCDs; see Fig. 11b in Appendix B. Second, forking-sequences help preserve gradients in early FCDs, thus helping alleviate vanishing gradient issues, as shown in Appendix C. As shown in Figure 3, a natural consequence of the reduced gradient variance is a faster and more stable convergence of the training procedure. In Appendix D, we verify this effect using linear autoregressive models optimized with convex quantile loss on synthetic data, demonstrating consistent improvements in both convex and non-convex regimes. The gradient variance reduction shown in **Theorem 1** can also be achieved with window-sampling by accumulating gradients across forward passes or increasing the batch size, but at greater computational cost than forking-sequences as shown in Figure 4.

Practical implications of these results are that models trained with forking-sequences can tolerate larger learning rates, require fewer iterations to reach comparable loss levels, and tend to follow smoother optimization trajectories. This leads to training procedures that are more robust to hyperparameter choices, less sensitive to initialization, and more computationally efficient.

### 3.4 Forking-Sequences Computational Complexity

As illustrated in Figure 2, the forking-sequences approach enhances efficiency by reusing the encoder outputs across all forecast creation dates (FCDs), which allows the model to propagate hidden states $\mathbf{h}_{[t][h]}$ from $\mathbf{h}_{[t-1][h]}$). In contrast to standard methods that must recompute the encoder's hidden states separately for each FCD, incurring repeated and redundant encoder passes, this design achieves its efficiency by reusing intermediate computations within a single forward pass. This substantially reduces the total number of encoder operations required to generate the full set of multi-horizon forecasts.

Forking-sequences naturally suit encoders like CNNs, LSTMs, RNNs, and Transformers capable of outputting intermediate encoder outputs, while also being adaptable to MLPs through appropriate input reformatting.

We show that this architectural design yields a computational speedup factor of $T$, where $T$ is the number of FCDs. See the complexity summary in Table 1 and complete calculation in Appendix E. Figure 4 empirically validates these complexity results by measuring the cross-validation inference time across series of different lengths. This experiment compares the computational cost of an MQCNN architecture cross validation inference when using three different variants: the standard full windows-sampling, the more efficient restricted windows-sampling, and our proposed forking-sequences approach.

**Table 1:** Theoretical Encoder Complexity

|         | MLP | CNN | RNN | Att. | S4 |
|---------|-----|-----|-----|------|-----|
| FS | $\mathcal{O}(T)$ | $\mathcal{O}(T)$ | $\mathcal{O}(T)$ | $\mathcal{O}(T^2)$ | $\mathcal{O}(T \log T)$ |
| WS restr. | $\mathcal{O}(TL)$ | $\mathcal{O}(TL)$ | $\mathcal{O}(TL)$ | $\mathcal{O}(TL^2)$ | $\mathcal{O}(TL \log L)$ |
| WS full | $\mathcal{O}(T^2)$ | $\mathcal{O}(T^2)$ | $\mathcal{O}(T^2)$ | $\mathcal{O}(T^3)$ | $\mathcal{O}(T^2 \log T)$ |

**Figure 4:** Empirical validation of computational complexity through wall-clock measurements across different inference methods. $T$ represents the time series length, and $L$ denotes the window size in restricted window-sampling.

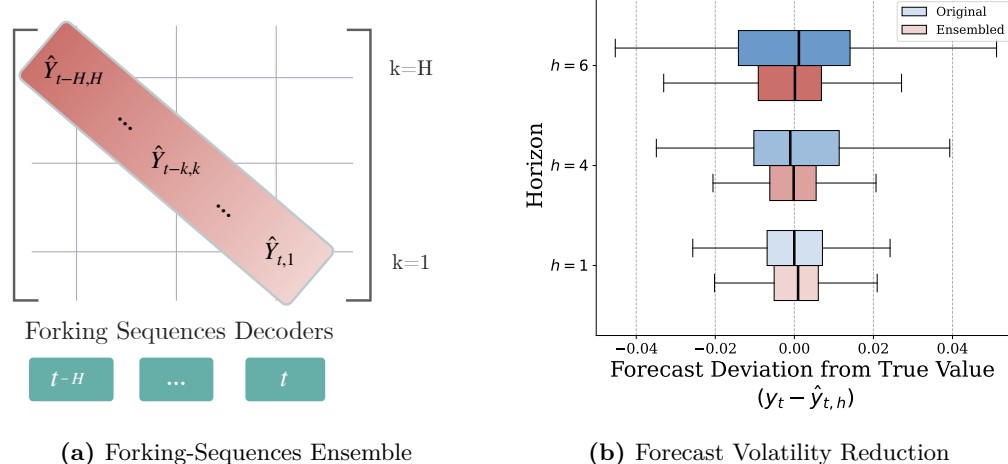

**(a)** Forking-Sequences Ensemble    **(b)** Forecast Volatility Reduction

**Figure 5:** a) We adapt forking-sequences during inference to ensemble multiple forecasts of the same future date $\tau$ by computing a function (ex., moving average) across predictions generated from previous FCDs. b) Forking-sequences ensembling reduces forecast volatility, reducing the estimators variance with a linear convergence rate analogous to the weak law of large numbers.

In the case of convolutional models, the forking-sequences scale linearly with the time series length, achieving a complexity of $\mathcal{O}(T)$ by producing forecasts at each FCD while reusing previous computations. In contrast, window-sampling incurs a higher computational cost, scaling quadratically as $\mathcal{O}(T^2)$ in the WS (full) case, or linearly as $\mathcal{O}(TL)$ when restricting the encoder to a window of length $L$ in the WS (restricted) case.

### 3.5 Forking-Sequences Ensembling

The forking-sequences architectural design can be extended to serve as an efficient forecasting ensemble technique to help reduce volatility of the forecast revisions in Equation 4. By generating predictions for all FCDs in a single forward pass, forking-sequences naturally produce overlapping predictions for each target date $t$, which can be assembled to reduce the volatility of the forecast as follows:

$$\widetilde{\mathbf{Y}}_{t,h} = \frac{1}{H} \sum_{k=0}^{H} \widehat{\mathbf{Y}}_{t-k,h+k} \qquad \text{for } t \geq H. \tag{11}$$

The forecast revisions are illustrated in Fig. 5a as the diagonal within the forecast grid $\widehat{\mathbf{Y}}_{[t][h]}$, and the temporally overlapped forecasts in Fig. 2b.

Although it is tempting to expect a variance-reduction behavior similar to the results of Theorem 1, it is important to recognize that forecast variance naturally increases the further a forecast is from its corresponding observation. As a result, there is an inherent limit to how much ensembling can reduce volatility: older forecast revisions carry substantially higher uncertainty, whereas more recent revisions are both more accurate and less variable. This makes it desirable for an ensemble to place greater weight on newer forecasts rather than treating all revisions equally. We investigate alternative ensembling strategies that account for this effect in Appendix I. Building on this discussion, ensemble techniques such as Equation 11 can also be applied to window-sampling variants, though at additional computational cost. We detail the impact of ensembling on forecast volatility for forking-sequences models in Appendix I.1 and for window-sampling pre-trained models in Fig. 8 and Appendix J.

We acknowledge that ensembling can be integrated during both training and inference with forking-sequences, and could be further extended with learnable parameters (as explored in (Eisenach et al., 2021)). We defer the exploration of incorporating forecast ensembling directly into the training phase to future work.

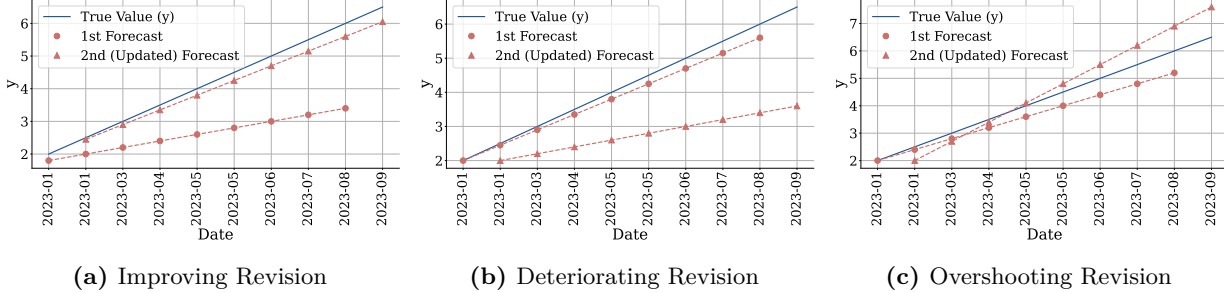

**(a)** Improving Revision        **(b)** Deteriorating Revision        **(c)** Overshooting Revision

**Figure 6:** Example penalty behavior of the Excess Volatility (EV) metric. EV distinguishes accuracy-improving revisions from accuracy-degrading ones, assigning no penalty when revisions improve accuracy, while asymmetrically penalizing both deteriorating and overshooting revisions according to their impact on accuracy.

### 3.6 Excess Volatility

Here we introduce a novel metric for assessing forecast volatility across FCDs: *Excess Volatility* (EV). EV is designed to reward accuracy-improving forecast revisions while distinguishing them from harmful volatility. EV addresses a limitation of measures such as forecast percentage change, which quantify revisions without considering forecast accuracy. Excess Volatility is defined as

$$\text{EV}(y, \ \hat{\mathbf{y}}_1, \ \hat{\mathbf{y}}_2) = \text{QL}(\hat{\mathbf{y}}_2, \hat{\mathbf{y}}_1) - (\text{QL}(y, \hat{\mathbf{y}}_1) - \text{QL}(y, \hat{\mathbf{y}}_2)). \tag{12}$$

For two consecutive forecasts $\hat{\mathbf{y}}_1$ and $\hat{\mathbf{y}}_2$, the EV metric in Equation 12 evaluates the quality of forecast revisions by asymmetrically rewarding revisions that improve accuracy and penalizing those that degrade it. Unlike percentage-change metrics such as sFPC, EV incorporates ground-truth values to judge whether a revision is beneficial or harmful.

The proof of the EV properties is included in Appendix F.

**Theorem 2.** (*Excess Volatility Metric Properties*) Taking the ground truth $y$ and two consecutive forecasts $\hat{\mathbf{y}}_1, \hat{\mathbf{y}}_2$, the EV metric has the following properties:

- **Positivity:** $\text{EV}(y, \ \hat{\mathbf{y}}_1, \ \hat{\mathbf{y}}_2) \geq 0$

- **Zero-Penalty for Improving Revisions:** When the revision from $\hat{\mathbf{y}}_1$ to $\hat{\mathbf{y}}_2$ represents a direct proportional improvement and $\hat{\mathbf{y}}_2$ lies on the linear path from $y$ to $\hat{\mathbf{y}}_1$, then $\text{EV}(y, \ \hat{\mathbf{y}}_1, \ \hat{\mathbf{y}}_2) = 0$.

- **Maximum penalty for Deteriorating Revisions:** When $\hat{\mathbf{y}}_1$ lies on the direct path between $y$ and $\hat{\mathbf{y}}_2$; that is, when the revision from $\hat{\mathbf{y}}_1$ to $\hat{\mathbf{y}}_2$ moves linearly away from the truth, then $\text{EV}(y, \ \hat{\mathbf{y}}_1, \ \hat{\mathbf{y}}_2) = \text{QL}(y, \hat{\mathbf{y}}_2) - \text{QL}(y, \hat{\mathbf{y}}_1)$. In this fully misdirected case, EV assigns the maximum penalty, equal to the entire accuracy degradation.

- **Overshoot-Revision Penalty Property:** When the ground truth $y$ lies on the line segment between $\hat{\mathbf{y}}_1$ and $\hat{\mathbf{y}}_2$; that is, when $\hat{\mathbf{y}}_2$ revises $\hat{\mathbf{y}}_1$ in the correct direction but overshoots, then $\text{EV} = \text{QL}(y, \hat{\mathbf{y}}_2)$. In this excess-revision case, EV penalizes only the overshoot.

## 4 Experiments

We now evaluate the forking-sequences architecture design in two separate but related studies: we first assess how forking-sequences impact model performance in comparison to window-sampling. We then assess how ensembling techniques impact forecast accuracy and volatility. We present results for forking-sequences and window-sampling without ensembling as well as forking-sequences with ensembling in individual columns in Tables 2 and 3. Our main study aims to answer the following research questions: (i) Accuracy: Do models trained with forking-sequences outperform those trained with window-sampling? (ii) Calibration: Do models

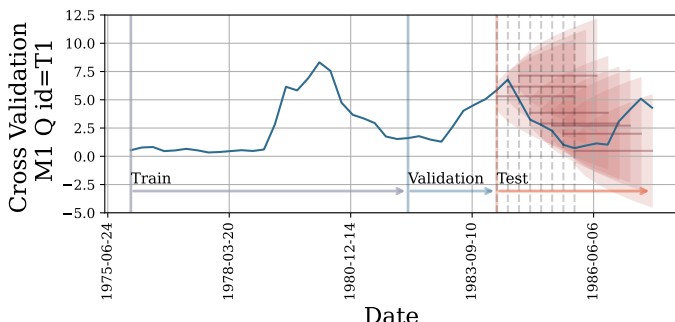

**Figure 7:** Example time series from the M1 competition dataset. The train and validation sets consist of all observations preceding the first dotted line. The cross validation test set is comprised of a set of FCDs with lengths defined in Table 6. To leverage all the FCDs within train for model estimation while preventing temporal leakage between splits, we apply masking across training and test sets.

trained with forking-sequences exhibit better calibration than those trained with window-sampling? (iii) Volatility: Do models trained with forking-sequences produce less volatile forecast revisions than those trained with window-sampling? (iii) Ensemble impact on volatility: To what extent does forking-sequences forecast ensembling reduce forecast volatility? (iv) Accuracy–Volatility Tradeoff: Can ensembling reduce volatility without degrading accuracy? (v) Architecture Interactions: How do the benefits of forking-sequences vary across different encoder architectures? We also consider ablations to assess various ensembling methods and their impact for zero-shot models: (vi) Ablations: How do alternative ensembling techniques affect accuracy and volatility, and how does forking-sequences mitigate vanishing gradients? Can we leverage forking-sequences forecast ensembling with pretrained Time Series Foundation Models (TSFMs)?

**Benchmark Datasets.** To measure the forking-sequences effects, we rely on the M-series benchmark. We use 16 large-scale time-series forecasting benchmarks, containing over 100,000 time series, drawn from well-known forecasting competitions: `M1` (Makridakis et al., 1982), `M3` (Makridakis and Hibon, 2000), `M4` (Makridakis et al., 2020), and `Tourism` (Athanasopoulos et al., 2011). We adopt the data handling and pre-processing practices established in prior work on cross-frequency transfer learning (Alexandrov et al., 2020; Olivares et al., 2025). To enable temporal cross-validation in our experiments, we expand the original test sets of a single forecast horizon to include earlier timesteps for $n$ prediction windows consecutively shifted by one FCD as depicted in Fig. 7. We adjust the validation set accordingly to equal a single forecast horizon preceding the test set with the train set preceding the validation set.

Let $h$ be the forecast horizon as specified in Table 6 and depicted in Figure 7, we define the train, validation, and test partitions, using numpy notation, of the data as follows:

$$\mathbf{Y}_{\text{train}} = \mathbf{Y}_{[\,:\,-3h+1]}, \qquad \mathbf{Y}_{\text{validation}} = \mathbf{Y}_{[-3h+1\,:\,-2h+1]}, \qquad \text{and} \qquad \mathbf{Y}_{\text{test}} = \mathbf{Y}_{[-2h+1\,:\,t]}. \qquad (13)$$

We train frequency-specialized models combining `M1`, `M3`, `M4`, and `Tourism` frequency-specific datasets. This enables us to leverage a larger training data corpus, while still focusing the forecasting tasks to frequency-specific prediction horizons (see Table 6). We default to these frequency-specific horizon values for the Tourism dataset as well for consistency across experiments.

**Forecast Baselines** For our main experiments, we evaluated a curated set of baseline models. These include the classic statistical model AutoARIMA (Hyndman and Khandakar, 2008; Hyndman et al., 2025) and several neural forecasting models designed for controlled comparisons. We fix the overall MQForecaster (Wen et al., 2017; Olivares et al., 2023) architecture and vary only the encoder and the training scheme. We select five fundamental architectures that represent the main neural backbones used throughout the time series forecasting literature: recurrent neural network (`RNN`), long-short-term memory (`LSTM`), convolutional neural network (`CNN`), `Transformer` and `State Space`. Our work employs and adapts a dilated RNN implementation (Chang et al., 2017; Olivares et al., 2022). The `Transformer` encoder uses skip connections and residual layers with dilated SelfAttention layers inspired by dilated convolutions (van den Oord et al., 2016). The `State`

`Space` encoder uses the architecture from the Structured State Space sequence model (`S4`) proposed by Gu et al. (2022). For each encoder variant of MQForecaster, we train models with forking-sequences and compare their generalization performance against models trained with window-sampling. Details on hyperparameter selection are provided in the Appendix H.

## 4.1 Accuracy and Volatility Metrics

To assess forecast accuracy, we use the *scaled Continuous Ranked Probability Score* (sCRPS, (Gneiting and Raftery, 2007)). sCRPS is a scaled version of the CRPS and is defined in Equation 14:

$$\text{sCRPS}\left(\mathbf{y}_{[b][t][h]},\ \hat{\mathbf{y}}_{[b][t][h]}\right) = \frac{\sum_{b,t,h} \text{CRPS}(y_{b,t,h},\ \hat{\mathbf{y}}_{b,t,h})}{\sum_{b,t,h} |y_{b,t,h}|}. \tag{14}$$

where CRPS is defined as an integral over quantiles:

$$\text{CRPS}(y, \hat{\mathbf{y}}) = \int_0^1 \text{QL}(y, \hat{y}^{(q)})dq \approx \frac{1}{|\mathcal{Q}|}\sum_{q \in \mathcal{Q}} \text{QL}(y, \hat{y}^{(q)}). \tag{15}$$

In practice, we approximate the integral above with a finite sum over equally spaced quantile levels, denoted by the set $\mathcal{Q}$. We use nine quantiles $q \in \{0.1, 0.2, \ldots, 0.9\}$ in our evaluation.

To asses forecast volatility, we use *Scaled Excess Volatility* (sEV) and *Symmetric Forecast Percentage Change* (sFPC). Like sCRPS, sEV is a scaled measurement that enables a comparable evaluation of the forecast volatility between datasets. The main body of the paper emphasizes sEV due to its compatibility with multi-quantile forecasting, while conventional sFPC-based point-forecast results are reported in Appendix K. sEV is defined as:

$$\text{sEV}\left(\mathbf{y}_{[b][t][h]},\ \hat{\mathbf{y}}_{[b][t][h]}\right) = \frac{\sum_{b,t,h} \text{EV}(y_{b,t,h},\ \hat{\mathbf{y}}_{b,t,h+1},\ \hat{\mathbf{y}}_{b,t+1,h})}{\sum_{b,t,h} |y_{b,t,h}|}. \tag{16}$$

We complement the main results with measurements of *Mean Absolute Error* (MAE) and *Average Coverage Error* (ACE) in Appendix K, and Appendix L.

## 4.2 Main Empirical Results

Table 2 reports mean sCRPS values, averaged over five runs, for all dataset–frequency combinations and model architectures, including statistical baselines. While all encoder variants with forking-sequences show improved sCRPS, the magnitude of improvement notably varies across encoder architectures and individual datasets as visualized in Fig. 16a, which shows per-dataset percentage changes in sCRPS for forking-sequences models relative to window-sampling models. For `LSTM` encoder models, the forking-sequences training scheme reduced sCRPS by 49.3%, on average across datasets, compared to the window-sampling scheme, as suggested by earlier observations by (Lipton et al., 2016; Dai and Le, 2015). Consistent median gains across datasets are observed for `RNN` (46.2%), and `CNN` (28.6%) encoders, while the `Transformer` (24.7%) and `State Space` (6.4%) encoders showed smaller improvement. The results of Table 2 are summarized in Figure 16a which shows the percentage change of the sCRPS metric for models with forking-sequences over that of models with window-sampling, averaged across datasets. Point forecasting results are reported in Appendix K.

Based on median sCRPS across datasets in Table 2, the top-performing models are forking-sequences variants of `CNN`, and `Transformer` encoders, in that order, all outperforming the statistical baselines `AutoETS` and `AutoARIMA`. The `S4` encoder with forking-sequences ranks fifth, following `AutoETS` but ahead of `AutoARIMA`. Among forking-sequences models, only the `RNN` and `LSTM` encoders do not outperform statistical baselines.

Table 3 presents sEV values, averaged over five runs, for all dataset–frequency combinations and model architectures, including statistical baselines. We observe that forecast volatility results vary across encoders. Median percentage improvements in sEV across datasets are observed for `CNN` (35.1%), and `S4` (2.8%) encoders with forking-sequences, while `RNN` (-43.6%), `LSTM` (-75.1%), and `Transformer` (-11.7%) encoders show greater improvements with window-sampling. Table 2 is summarized in Fig. 16e which shows the percentage change

**Table 2:** Empirical evaluation of probabilistic forecasts. Mean scaled continuous ranked probability score (sCRPS) averaged over 5 runs. Lower measurements are preferred. The methods without standard deviation have deterministic solutions. For the MQForecaster architecture we vary the type of encoder, and the training scheme between forking-sequences (FS) and window-sampling (WS). We include the forking-sequences ensemble variant (EN).

| | Freq | RNN EN | RNN FS | RNN WS | LSTM EN | LSTM FS | LSTM WS | CNN EN | CNN FS | CNN WS | Transformer EN | Transformer FS | Transformer WS | S4 EN | S4 FS | S4 WS | StatsForecast ETS | StatsForecast ARIMA |
|---|---|---|---|---|---|---|---|---|---|---|---|---|---|---|---|---|---|---|
| M1 | M | 0.6532 (0.0857) | 0.6537 (0.0855) | 0.8903 (0.092) | 0.659 (0.0755) | 0.6592 (0.0754) | 0.8632 (0.1304) | 0.1278 (0.0018) | 0.1282 (0.0019) | 0.2094 (0.0167) | 0.1418 (0.0076) | 0.1425 (0.0078) | 0.1907 (0.0145) | 0.2134 (0.0293) | 0.2168 (0.0288) | 0.2131 (0.003) | 0.1418 (-) | 0.1509 (-) |
| | Q | 0.6424 (0.051) | 0.6427 (0.0511) | 0.8897 (0.0316) | 0.675 (0.0846) | 0.6749 (0.0847) | 0.9537 (0.09) | 0.1138 (0.0071) | 0.1141 (0.0072) | 0.125 (0.0088) | 0.1186 (0.0039) | 0.1187 (0.0038) | 0.1366 (0.0196) | 0.136 (0.0112) | 0.1381 (0.0112) | 0.1289 (0.0023) | 0.1139 (-) | 0.1282 (-) |
| | Y | 0.9947 (0.0008) | 0.9947 (0.0008) | 0.9975 (0.0014) | 0.9947 (0.0011) | 0.9947 (0.0011) | 0.9973 (0.0017) | 0.1011 (0.007) | 0.1019 (0.0068) | 0.1024 (0.0096) | 0.1088 (0.0086) | 0.1097 (0.0086) | 0.1271 (0.02) | 0.1109 (0.005) | 0.1112 (0.0048) | 0.1149 (0.0075) | 0.1097 (-) | 0.1068 (-) |
| M3 | O | 0.0669 (0.0366) | 0.0669 (0.0368) | 0.2311 (0.1047) | 0.1144 (0.1355) | 0.1143 (0.1356) | 0.3307 (0.1136) | 0.038 (0.0083) | 0.0379 (0.0082) | 0.0479 (0.0071) | 0.0344 (0.0008) | 0.0344 (0.0008) | 0.0785 (0.0395) | 0.0528 (0.0038) | 0.0524 (0.0037) | 0.0417 (0.0004) | 0.0328 (-) | 0.0337 (-) |
| | M | 0.121 (0.0053) | 0.1227 (0.0058) | 0.2273 (0.0813) | 0.1162 (0.0054) | 0.1173 (0.0055) | 0.2334 (0.0803) | 0.0893 (0.0007) | 0.0895 (0.0007) | 0.1401 (0.023) | 0.0955 (0.0048) | 0.096 (0.0049) | 0.1266 (0.0097) | 0.0989 (0.004) | 0.0999 (0.0044) | 0.1326 (0.0003) | 0.105 (-) | 0.1059 (-) |
| | Q | 0.085 (0.0024) | 0.0853 (0.0026) | 0.1356 (0.0431) | 0.087 (0.0043) | 0.0872 (0.0046) | 0.2137 (0.0556) | 0.0708 (0.0046) | 0.0708 (0.0047) | 0.0935 (0.0058) | 0.071 (0.0036) | 0.0708 (0.0035) | 0.1159 (0.055) | 0.0757 (0.0031) | 0.076 (0.0035) | 0.088 (0.0007) | 0.0773 (-) | 0.0779 (-) |
| | Y | 0.1647 (0.0195) | 0.1649 (0.0193) | 0.3476 (0.1352) | 0.1764 (0.0284) | 0.1764 (0.0284) | 0.3518 (0.0883) | 0.1265 (0.0024) | 0.1264 (0.0025) | 0.1684 (0.0056) | 0.1351 (0.013) | 0.1349 (0.0126) | 0.1767 (0.0328) | 0.14 (0.0074) | 0.1396 (0.0072) | 0.1492 (0.0027) | 0.149 (-) | 0.1549 (-) |
| M4 | H | 0.3041 (0.0787) | 0.3042 (0.0787) | 0.6371 (0.0305) | 0.3151 (0.0916) | 0.3151 (0.0916) | 0.5813 (0.036) | 0.031 (0.0026) | 0.031 (0.0025) | 0.1586 (0.0062) | 0.0394 (0.0071) | 0.0396 (0.0072) | 0.1328 (0.0114) | 0.0426 (0.0074) | 0.0428 (0.0073) | 0.1554 (0.0021) | 0.0684 (-) | 0.0308 (-) |
| | D | 0.0253 (0.0009) | 0.0253 (0.0009) | 0.2076 (0.1574) | 0.0258 (0.0026) | 0.0257 (0.0026) | 0.2282 (0.1409) | 0.0214 (0.0005) | 0.0214 (0.0004) | 0.0516 (0.0284) | 0.022 (0.0003) | 0.022 (0.0003) | 0.0223 (0.0005) | 0.022 (0.0002) | 0.022 (0.0002) | 0.0222 (0.0004) | 0.0226 (-) | 0.0226 (-) |
| | W | 0.0606 (0.0034) | 0.0608 (0.0034) | 0.1965 (0.0551) | 0.0641 (0.0029) | 0.0643 (0.0029) | 0.3046 (0.0574) | 0.0425 (0.0012) | 0.0426 (0.0012) | 0.0695 (0.0127) | 0.0459 (0.0062) | 0.0461 (0.0064) | 0.0638 (0.0052) | 0.0521 (0.0038) | 0.0524 (0.0039) | 0.0603 (0.0004) | 0.0542 (-) | 0.0507 (-) |
| | M | 0.1136 (0.0059) | 0.1143 (0.0061) | 0.2992 (0.1314) | 0.1101 (0.0045) | 0.1104 (0.0045) | 0.3037 (0.1249) | 0.0875 (0.0004) | 0.0875 (0.0005) | 0.129 (0.0259) | 0.0922 (0.0031) | 0.0924 (0.0031) | 0.1187 (0.0097) | 0.0957 (0.0033) | 0.0962 (0.0033) | 0.1174 (0.0006) | 0.097 (-) | 0.0987 (-) |
| | Q | 0.092 (0.0018) | 0.0925 (0.0019) | 0.1727 (0.0384) | 0.097 (0.0066) | 0.0974 (0.0069) | 0.3349 (0.1022) | 0.0788 (0.0047) | 0.0787 (0.0048) | 0.1008 (0.0066) | 0.0806 (0.0049) | 0.0805 (0.0048) | 0.1272 (0.0653) | 0.0842 (0.0036) | 0.0847 (0.0038) | 0.0945 (0.0005) | 0.0807 (-) | 0.0849 (-) |
| | Y | 0.185 (0.0386) | 0.1851 (0.0386) | 0.4282 (0.1355) | 0.21 (0.0661) | 0.2101 (0.0662) | 0.411 (0.1189) | 0.1076 (0.002) | 0.1076 (0.0021) | 0.1383 (0.0029) | 0.1147 (0.0075) | 0.1146 (0.0072) | 0.1505 (0.0271) | 0.1206 (0.0043) | 0.1201 (0.0042) | 0.1283 (0.0016) | 0.124 (-) | 0.1322 (-) |
| Tourism | M | 0.6601 (0.0733) | 0.6605 (0.0684) | 0.838 (0.0495) | 0.6527 (0.0684) | 0.6529 (0.0684) | 0.8197 (0.0644) | 0.0956 (0.0062) | 0.096 (0.0063) | 0.275 (0.0149) | 0.1303 (0.032) | 0.1311 (0.0325) | 0.2499 (0.0206) | 0.2386 (0.0363) | 0.2408 (0.0361) | 0.2812 (0.0024) | 0.1198 (-) | 0.1479 (-) |
| | Q | 0.8072 (0.0235) | 0.8077 (0.0234) | 0.9306 (0.0083) | 0.8177 (0.0479) | 0.8182 (0.0478) | 0.9425 (0.0148) | 0.1037 (0.0124) | 0.1044 (0.0132) | 0.164 (0.008) | 0.0989 (0.0029) | 0.0988 (0.003) | 0.2264 (0.1103) | 0.1687 (0.0112) | 0.1737 (0.0112) | 0.1724 (0.0021) | 0.1042 (-) | 0.1187 (-) |
| | Y | 0.9819 (0.0022) | 0.9819 (0.0022) | 0.9901 (0.0036) | 0.9821 (0.0031) | 0.9821 (0.0031) | 0.9891 (0.0044) | 0.1631 (0.0036) | 0.1631 (0.0036) | 0.1927 (0.0047) | 0.1732 (0.0055) | 0.1724 (0.0052) | 0.2163 (0.0376) | 0.1869 (0.0031) | 0.1853 (0.003) | 0.1874 (0.0026) | 0.1413 (-) | 0.1697 (-) |

of the sEV metric for models with forking-sequences over that of models with window-sampling, averaged across datasets. We present sFPC results in Table 20.

We show that for forking-sequences models, forecast ensembling during inference can reduce forecast volatility compared to forecasts without ensembling for all encoders as shown in Fig. 15. Specifically, applying exponential smoothing ($\alpha = 0.9$) at inference to models trained with forking-sequences yields median percentage improvements in sEV across datasets of 13.2%, 13.0%, 10.9%, 10.2%, and 11.2% for RNN, LSTM, CNN, Transformer, and State Space-based architectures, respectively, while maintaining forecast accuracy. Furthermore, compared to window-sampling models, these forking-sequences models with ensembling show median percentage changes in sEV across datasets of -24.5%, -52.4%, 42.2%, 1.0%, and 13.7% for RNN, LSTM, CNN, Transformer, and S4 encoders, respectively (Fig. 16f).

Forking-sequences significantly improves upon windows-sampling calibration, measured with average coverage error (ACE) as shown in Appendix L.1. We observe median percentage improvements across datasets of 35.9%, 66.3%, 61.3%, 61.5%, 43.9%, and 17.6% for MLP, RNN, LSTM, CNN, Transformer, and S4 encoders, respectively. We note that the additional ACE improvement from ensembling during inference over forking-sequences without ensembling is relatively marginal, with median percentage improvements across datasets of 0.3%, 0.0%, 0.0%, 0.5%, -1.1%, 0.1% for MLP, RNN, LSTM, CNN, Transformer, and S4 encoders.

## 4.3 Ablation Studies

In Table 2, we observe that the performance gains from forking-sequences vary between encoder architectures. In controlled experiments where only the encoder module differs, Transformer and State Space encoder models show smaller improvements, which we hypothesized may stem from their inherently stronger gradient propagation that reduce vanishing gradient issues of recurrent architectures. Appendix C presents both empirical and theoretical exploration of how forking-sequences and window-sampling influence gradient flow in LSTMs and Transformers. By analyzing the Jacobian products that compose backpropagation-through-time, we find that window-sampling LSTM encoders suffer from vanishing gradients with respect to the series length, whereas Transformer encoders do not.

**Table 3:** Empirical evaluation of probabilistic forecasts stability. Mean *scaled Excess Volatility* (sEV) averaged over 5 runs. Lower measurements are preferred. The methods without standard deviation have deterministic solutions. For the `MQForecaster` architecture we vary the type of encoder, and the training scheme between forking-sequences (FS) and window-sampling (WS). We include the forking-sequences ensemble variant (EN).

| | Freq | RNN EN | RNN FS | RNN WS | LSTM EN | LSTM FS | LSTM WS | CNN EN | CNN FS | CNN WS | Transformer EN | Transformer FS | Transformer WS | S4 EN | S4 FS | S4 WS | StatsForecast ETS | StatsForecast ARIMA |
|---|---|---|---|---|---|---|---|---|---|---|---|---|---|---|---|---|---|---|
| M1 | M | 0.0056 (0.0013) | 0.0067 (0.0016) | 0.0027 (0.0017) | 0.0046 (0.001) | 0.0054 (0.0012) | 0.0025 (0.0016) | 0.0131 (0.0006) | 0.0149 (0.0007) | 0.0441 (0.0057) | 0.0176 (0.0022) | 0.0198 (0.0024) | 0.0163 (0.0172) | 0.035 (0.0036) | 0.0401 (0.0042) | 0.0496 (0.0012) | 0.0069 (-) | 0.0085 (-) |
| | Q | 0.0046 (0.0025) | 0.0052 (0.0028) | 0.0018 (0.0004) | 0.0028 (0.0009) | 0.0034 (0.0012) | 0.0007 (0.0001) | 0.014 (0.0013) | 0.0158 (0.0014) | 0.0185 (0.0017) | 0.0129 (0.0021) | 0.0144 (0.0025) | 0.0189 (0.0085) | 0.0236 (0.002) | 0.0268 (0.0023) | 0.0227 (0.0004) | 0.0129 (-) | 0.0236 (-) |
| | Y | 0.0001 (0.00003) | 0.0001 (0.00004) | 0.0001 (0.00001) | 0.00005 (0.00001) | 0.0001 (0.00001) | 0.0001 (0.00002) | 0.0201 (0.0015) | 0.0227 (0.0016) | 0.016 (0.0017) | 0.0207 (0.0032) | 0.0233 (0.0038) | 0.0117 (0.0065) | 0.0162 (0.0029) | 0.0181 (0.0032) | 0.0157 (0.0017) | 0.0122 (-) | 0.0166 (-) |
| M3 | O | 0.0037 (0.0008) | 0.0042 (0.0009) | 0.0039 (0.0016) | 0.002 (0.0007) | 0.0022 (0.0008) | 0.0014 (0.0006) | 0.0045 (0.0003) | 0.005 (0.0002) | 0.0033 (0.0006) | 0.0041 (0.0003) | 0.0046 (0.0004) | 0.002 (0.0016) | 0.0039 (0.0006) | 0.0043 (0.0007) | 0.0034 (0.0005) | 0.004 (-) | 0.004 (-) |
| | M | 0.0156 (0.0025) | 0.0183 (0.0029) | 0.0119 (0.0098) | 0.0126 (0.0005) | 0.0147 (0.0006) | 0.0096 (0.008) | 0.0084 (0.0003) | 0.0095 (0.0003) | 0.0222 (0.0033) | 0.0111 (0.0009) | 0.0125 (0.001) | 0.0087 (0.0085) | 0.0129 (0.0019) | 0.0147 (0.0022) | 0.0255 (0.0007) | 0.0053 (-) | 0.0061 (-) |
| | Q | 0.0096 (0.0009) | 0.011 (0.001) | 0.0155 (0.002) | 0.0087 (0.0015) | 0.0099 (0.0018) | 0.0053 (0.0033) | 0.0083 (0.0004) | 0.0093 (0.0005) | 0.013 (0.0017) | 0.0074 (0.0015) | 0.0082 (0.0017) | 0.0134 (0.006) | 0.0102 (0.0009) | 0.0116 (0.0011) | 0.0158 (0.0005) | 0.0071 (-) | 0.009 (-) |
| | Y | 0.0138 (0.0023) | 0.0159 (0.0027) | 0.0093 (0.0018) | 0.0128 (0.0024) | 0.0146 (0.0029) | 0.0081 (0.0023) | 0.0205 (0.0006) | 0.023 (0.0007) | 0.0165 (0.0023) | 0.0197 (0.0022) | 0.0219 (0.0026) | 0.0138 (0.0059) | 0.0173 (0.002) | 0.0193 (0.0021) | 0.0174 (0.0018) | 0.0126 (-) | 0.0211 (-) |
| M4 | H | 0.0022 (0.0003) | 0.0023 (0.0004) | 0.0049 (0.0024) | 0.0012 (0.0006) | 0.0013 (0.0007) | 0.0033 (0.0014) | 0.0015 (0.0002) | 0.0016 (0.0002) | 0.0117 (0.0011) | 0.0026 (0.0013) | 0.0028 (0.0014) | 0.0036 (0.0029) | 0.0044 (0.0006) | 0.0049 (0.0007) | 0.0115 (0.0003) | 0.0038 (-) | 0.0014 (-) |
| | D | 0.0019 (0.0003) | 0.0022 (0.0004) | 0.0012 (0.0005) | 0.0017 (0.0004) | 0.002 (0.0005) | 0.0005 (0.0002) | 0.0019 (0.0002) | 0.0021 (0.0002) | 0.0126 (0.0203) | 0.0016 (0.0002) | 0.0018 (0.0002) | 0.0021 (0.0001) | 0.002 (0.0001) | 0.0022 (0.0001) | 0.0021 (0.0004) | 0.0022 (-) | 0.0022 (-) |
| | W | 0.0048 (0.0005) | 0.0055 (0.0006) | 0.0052 (0.0005) | 0.0044 (0.0001) | 0.0051 (0.0002) | 0.0031 (0.001) | 0.0042 (0.0002) | 0.0047 (0.0002) | 0.0081 (0.0013) | 0.0045 (0.0013) | 0.005 (0.0016) | 0.0053 (0.0033) | 0.0065 (0.0003) | 0.0074 (0.0003) | 0.0079 (0.0001) | 0.0052 (-) | 0.004 (-) |
| | M | 0.0101 (0.0011) | 0.0117 (0.0013) | 0.0068 (0.0054) | 0.0081 (0.0003) | 0.0093 (0.0004) | 0.0055 (0.0041) | 0.0074 (0.0002) | 0.0082 (0.0002) | 0.0153 (0.002) | 0.0087 (0.0006) | 0.0097 (0.0007) | 0.0063 (0.0056) | 0.0098 (0.0011) | 0.011 (0.0012) | 0.0172 (0.0005) | 0.0075 (-) | 0.008 (-) |
| | Q | 0.0102 (0.0006) | 0.0118 (0.0007) | 0.0111 (0.0013) | 0.0092 (0.0012) | 0.0106 (0.0014) | 0.004 (0.0017) | 0.009 (0.0004) | 0.0101 (0.0005) | 0.0134 (0.0015) | 0.0085 (0.0017) | 0.0095 (0.0019) | 0.0137 (0.0063) | 0.0111 (0.0008) | 0.0126 (0.0009) | 0.0163 (0.0005) | 0.0082 (-) | 0.0091 (-) |
| | Y | 0.0097 (0.0013) | 0.011 (0.0015) | 0.0082 (0.0015) | 0.0091 (0.0007) | 0.0101 (0.0009) | 0.0072 (0.0022) | 0.017 (0.0004) | 0.0192 (0.0004) | 0.0138 (0.002) | 0.0166 (0.0026) | 0.0185 (0.0029) | 0.0114 (0.0047) | 0.0137 (0.0015) | 0.0153 (0.0016) | 0.0142 (0.0016) | 0.0109 (-) | 0.0192 (-) |
| Tourism | M | 0.0073 (0.0016) | 0.0084 (0.0018) | 0.004 (0.003) | 0.0055 (0.0011) | 0.0064 (0.0012) | 0.0027 (0.0024) | 0.011 (0.0011) | 0.0127 (0.0013) | 0.043 (0.0051) | 0.0159 (0.0047) | 0.018 (0.0051) | 0.0177 (0.0156) | 0.0285 (0.0028) | 0.0322 (0.0032) | 0.0474 (0.0011) | 0.0089 (-) | 0.0091 (-) |
| | Q | 0.0049 (0.001) | 0.0057 (0.0012) | 0.0023 (0.0003) | 0.0043 (0.001) | 0.0049 (0.0012) | 0.0007 (0.0005) | 0.0168 (0.0037) | 0.0191 (0.0045) | 0.0395 (0.008) | 0.0126 (0.0023) | 0.0141 (0.0026) | 0.0409 (0.0203) | 0.045 (0.0018) | 0.0505 (0.002) | 0.0503 (0.0007) | 0.0186 (-) | 0.0203 (-) |
| | Y | 0.0001 (0.00003) | 0.0002 (0.00003) | 0.0002 (0.0001) | 0.0002 (0.00003) | 0.0002 (0.00003) | 0.0001 (0.0001) | 0.0277 (0.0015) | 0.0308 (0.0017) | 0.0176 (0.0022) | 0.0202 (0.0024) | 0.0223 (0.0028) | 0.0136 (0.0052) | 0.0176 (0.0033) | 0.0193 (0.0036) | 0.0165 (0.0016) | 0.0226 (-) | 0.0284 (-) |

In contrast with window-sampling, forking-sequences the `LSTM` encoder does not have the vanishing gradient problem. We complement these findings with explicit measurements of gradient norms between positions in the context window (Fig. 12).

Our ablation study in Appendix I highlights the trade-off between forecast volatility and accuracy under different ensembling strategies, showing that stronger smoothing can reduce volatility at the cost of slower adaptation to new information. Exponential smoothing with $\alpha = 0.9$ emerges as an effective approach, successfully reducing forecast volatility while maintaining model sCRPS. Although lower $\alpha$ values reduce forecast volatility in terms of sEV, they do so at the cost of accuracy. This trade-off is intuitive since $\alpha$ controls the weighting of future forecasts: higher $\alpha$ values give more weight to target date predictions closer in the horizon, which are generally more accurate than predictions for the same target date farther in the horizon. As such, this weighting scheme helps preserve the accuracy of the forecast while reducing volatility.

In Appendix J, we examine whether the forking-sequences forecast ensembling benefits extend beyond architectures specifically designed for it. We evaluated `NBEATS` (Oreshkin et al., 2020) and `PatchTST` (Nie et al., 2023) (both pretrained as foundation models on synthetic data), as well as pretrained `Chronos-2` (Ansari et al., 2025), `Toto 2.0` (Khwaja et al., 2026), and `TimesFM` (Das et al., 2024). Model details are included in Section H. Our results show that these architectures can be seamlessly augmented with forecast ensembling during inference to achieve a median reduction in forecast volatility of approximately 10% across the M-series benchmark. This improvement comes without compromising their original sCRPS (less than 0.1% difference). We summarize these findings in Fig. 8.

In our final ablation study reported in Figure 9 and Appendix L.2, we investigate the effects of forecast ensembles on the probabilistic coverage of forecasts in the M1 and M3 datasets. We found no consistent calibration effect attributable to the ensemble for window sizes between one to six. On M3 quarterly dataset, increasing the ensemble's window size marginally alleviated the over-coverage for P10 and P50 while maintained P90 coverage. Conversely, on the M3 yearly dataset, increasing the window size slightly degraded P50 coverage. These mixed, dataset- and quantile-dependent results suggest the coverage impact of simple ensembling is inconclusive, highlighting the need for future research to explore more sophisticated,

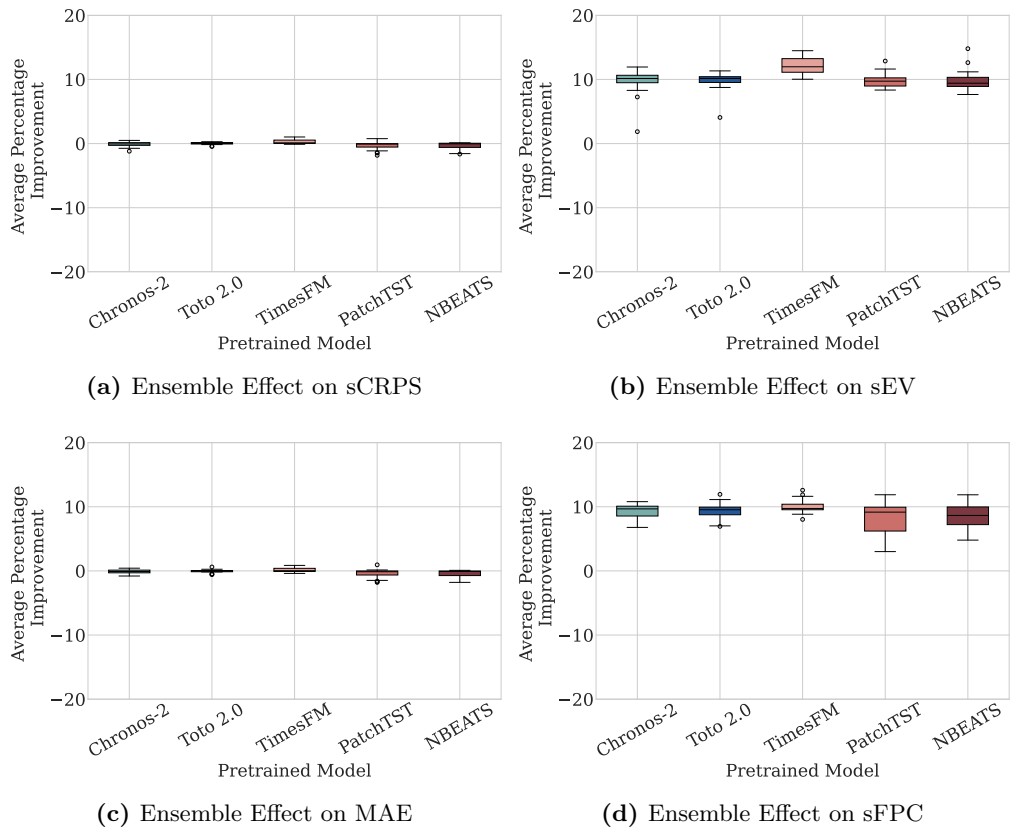

**(a)** Ensemble Effect on sCRPS

**(b)** Ensemble Effect on sEV

**(c)** Ensemble Effect on MAE

**(d)** Ensemble Effect on sFPC

**Figure 8:** Ablation study on applying forecast ensembling at inference. For each dataset, we average metric values across random seeds for ensembling (exponential smoothing $\alpha = 0.9$) versus no ensembling, calculate the percentage difference, then plot the distribution of percentage improvements across datasets for pretrained time series foundation models. Forecast ensembling can substantially reduce forecast volatility (sEV, sFPC) while maintaining forecast accuracy (sCRPS, MAE), demonstrating its utility as a general purpose technique during inference that can be leveraged for models trained with either forking-sequences or window-sampling.

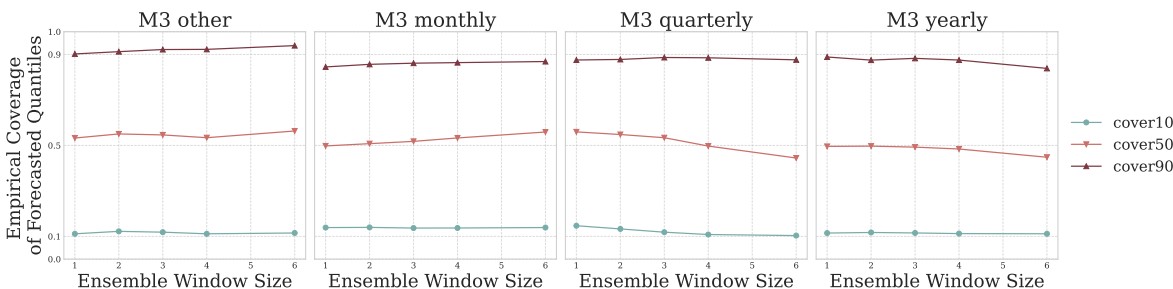

**Figure 9:** Ensemble Effect on Coverage. Up to six-length windows we do not observe clear effects of the ensemble size in the coverage of NBEATS forecasted quantiles.

conditionally-aware aggregation methods, such as those inspired by Quantile Regression Averaging (Liu et al., 2017) or weighted ensembles with quantile-specific weights (Fakoor et al., 2023).

## 5 Discussion and Conclusion

**Central Findings.** We re-visit *forking-sequences* and introduce two complementary metrics of forecast volatility; *scaled excess volatility* (sEV) and *symmetric forecast percentage change* (sFPC). We provided

extensive theoretical and empirical validation of the main benefits of forking-sequences, showing: 1) improved gradients by enhancing the signal to noise ratio and mitigating vanishing gradients, 2) an order-of-magnitude improvement in the computational efficiency of cross-validation inference compared with window-sampling, and 3) a natural ability to ensemble overlapping multi-horizon forecasts. Using the M-series benchmark; 16 datasets from the M1, M3, M4, and Tourism competitions, we demonstrated significant improvements in forecast volatility with no accuracy tradeoff, both for MQForecaster architectures and for other foundation forecasting models.

**Limitations and Research Directions.** This forking-sequences study focused only on univariate time series forecasting, but the technique can be readily extended to multivariate settings. We chose the univariate case to isolate the theoretical and empirical effects of the paradigm in the most general forecasting scenario, leaving multivariate extensions as a promising direction for future work. Additionally, we focus on simple forecast ensemble strategies (see the Appendix I), but accuracy-volatility tradeoffs could be further improved by integrating forecasting-specific inductive biases in the ensemble design or by learning the ensemble weights directly (Liu et al., 2017; Eisenach et al., 2021; Fakoor et al., 2023). We restricted our analysis to frequency-specialized models, but foundation forecasting models are typically trained in a cross-frequency transfer learning setting (Ness et al., 2023; Olivares et al., 2025). It remains an open question whether the benefits of forking-sequences extend to scenarios where series vary widely in scale and length. A foreseeable limitation is the increase in memory usage when applying the forking-sequences to high-frequency data, where time series tend to be of longer length. Our two metrics serve as complementary diagnostic tools: while sFPC measures the raw revision magnitude of point forecasts without requiring ground truth, sEV retroactively assesses the volatility of probabilistic forecasts by distinguishing accuracy-improving revisions from detrimental ones. Although sEV is a compelling metric, it remains to be determined whether it satisfies the conditions of a strictly proper scoring rule Gneiting and Raftery (2007) or if it could be artificially improved through oversmoothing at the expense of sharpness. An additional research line could explore leveraging such metrics during training to regularize learning with respect to forecast volatility.

In computer vision and natural language processing, the forking-sequences architectural design and its associated training paradigms are the default, yet they remain underexplored in the forecasting domain. The limited adoption of forking-sequences is partly explained by the dominance of simple window-sampling pipelines in major neural forecasting libraries `TSLib` (Wu et al., 2023), `Darts` (Herzen et al., 2022), `GluonTS` (Alexandrov et al., 2020), `PytorchForecasting` (Beitner, 2020), and `NeuralForecast` (Olivares et al., 2022). As mainstream neural forecast libraries grow in complexity, integrating forking-sequence–based training and inference becomes increasingly difficult. We hope that our results motivate the rediscovery of these techniques within the forecasting community and encourage library maintainers to adopt forking-sequences.

## Acknowledgments

The authors would like to thank Kevin Chen, Lee Dicker, Dean Foster, and Kari Torkola for their insightful comments and discussions. We also express our gratitude to Sunny Ruan for her invaluable assistance with the SPADEv2 architecture; her contributions enabled the integration of a forking-sequences ensemble, which reduced production forecast revision audits by 75%.

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

## A  Forecast Grid and Train Masking

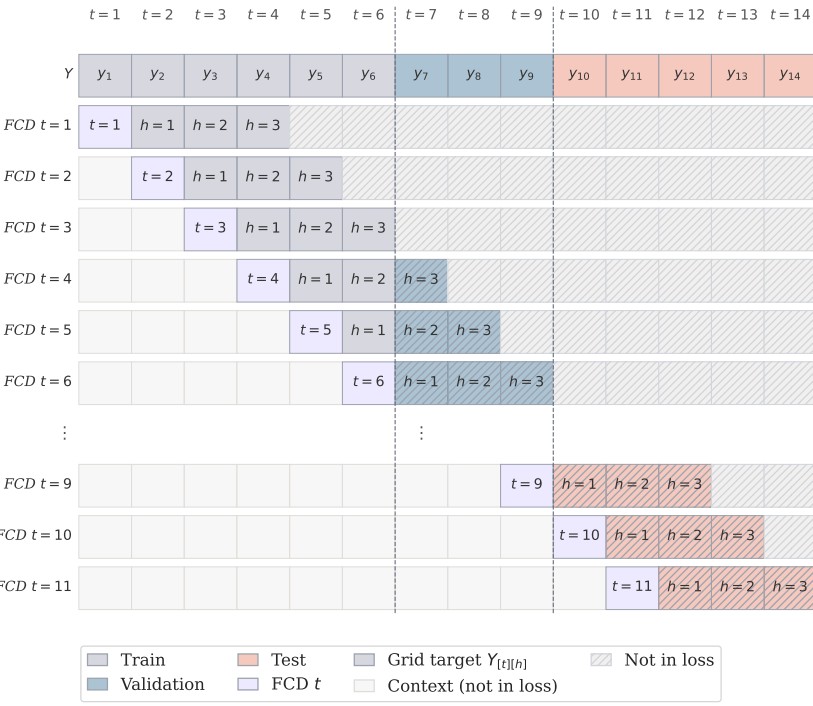

**Figure 10:** Example of a forking-sequences target grid. The validation set, marked in blue, corresponds to the data between the dotted lines, while the test set is shown in orange. Forking-sequences architectures generate forecasts for all FCDs simultaneously by reusing the encoder's computations, whereas window-sampling produces forecasts for each FCD independently. A masking strategy depicted with hatching lines prevents temporal leakge.

In this Appendix, we explain in greater detail the creation of the target grid described in Equation 2, and its implications to the models' optimization. A forking-sequence architecture design produces a target grid $\mathbf{Y}_{[t][h]} \in \mathbb{R}^{T \times H}$ for all $T$ FCD simultaneously, instead of a single FCD target $\mathbf{Y}_{t,[h]} \in \mathbb{R}^H$ as in the window-sampling approach. The implied optimization is described in Algorithm 1. To prevent temporal leakage across the train, validation, and test splits when training on multiple FCDs, forking sequence architectures mask the validation and test sets, following the strategy illustrated in Figure 10.

---

**Algorithm 1** Forking-Sequences or Windows-Sampling (SGD)

---

1: Initialize model $\hat{\boldsymbol{\theta}}_1 \in \Theta$, select learning rate $\alpha$, steps $s \leftarrow 1$
2: **repeat**
3:     Uniformly sample $b$-batch of full grid targets $Y_{[b][t][h]}$ or single targets $Y_{[b],t,[h]}$
4:     Mask training objectives based on $\mathbb{1}_{\{t+[h] \in \text{Train}\}}$
5:     Compute gradient

6:         $\text{WS} : \nabla\mathcal{L} = \frac{1}{B \times H} \sum_{b,h} \nabla\text{QL}(y_{b,t,h}, \hat{y}_{b,t,h}, \hat{\boldsymbol{\theta}}_s)$ or

7:         $\text{FS} : \nabla\mathcal{L}_T = \frac{1}{B \times T \times H} \sum_{b,t,h} \nabla\text{QL}(y_{b,t,h}, \hat{y}_{b,t,h}, \hat{\boldsymbol{\theta}}_s)$

8:     Update parameters with momentum-transformed gradient $\hat{\boldsymbol{\theta}}_{s+1} = \hat{\boldsymbol{\theta}}_s - \alpha \, \text{m}(\nabla\mathcal{L})$
9:     $s \leftarrow s + 1$
10: **until** $s \geq S_{\max}$

---

## B    Forking-Sequences Theoretical Foundations

Here we prove the gradient variance reduction guarantees for the forking-sequences training scheme. The gradient is computed with respect to the model parameters by minimizing Equation 5. We show that, under reasonable temporal correlation assumptions, the gradient variance decreases linearly with the number of FCDs ($\mathcal{O}(1/T)$), and its signal-to-noise ratio (SNR) improves linearly with the number of FCDs.

Our arguments are based on techniques and definitions from Shumway and Stoffer (2017) and Marshall and Olkin (1960). M-dependence is illustrated in Figure 11a.

For simplicity of the arguments' in the section we simplify the notation to refer to the forking-sequences gradient estimator:

$$\nabla L_T = \frac{1}{B \times T \times H} \sum_{b=1}^{B} \sum_{t=1}^{T} \sum_{h=1}^{H} \nabla \text{QL} \left( y_{b,t,h}, \ \hat{y}_{b,t,h} \right). \tag{17}$$

**Definition 1.** Let $\{\nabla L_t\} \subset \mathbb{R}^P$, with P the dimension of the network parameters $\boldsymbol{\theta}$, be a sequence of random variables, the sequence is $M$-dependent if:

- The gradient estimators are stationary $\mathbb{E}[\nabla L_t] = \mu$, with $\mu$ its expected gradient.

- with bounded stationary covariance $\Sigma = \text{Cov}(\nabla L_s, \nabla L_t) = \gamma(|s - t|) \in \mathbb{R}^{P \times P}$.

  where $\gamma(b) = \text{Cov}(\nabla L_0, \nabla L_b)$ and $\gamma(b) = 0$ for all $|b| > M$,

  that is, the gradient estimators $\{\nabla L_t\}$ are not correlated after M steps.

**Lemma 1.** Consider the forking sequences gradient estimator $\bar{\nabla} L_T = \frac{1}{T} \sum_t \nabla L_t$. If the gradient samples are M-dependent, the covariance estimator almost surely converges to 0 as $T$ grows, that is, $\text{Cov}\left(\bar{\nabla} L_T\right) \xrightarrow{a.s.} \mathbf{0}$.

We prove that with probability 1, the sequence of covariance estimator random variables converges to 0. That is $\mathbb{P}\left(\lim_{T \to \infty} \text{Cov}\left(\bar{\nabla} L_T\right) = 0\right) = 1$

*Proof.*

$$
\begin{aligned}
\text{Cov}(\bar{\nabla} L_T) &= \frac{1}{T^2} \text{Cov} \left( \sum_{t=1}^{T} \nabla L_t \right) \\
&= \frac{1}{T^2} \sum_{s=1}^{T} \sum_{t=1}^{T} \text{Cov}(\nabla L_s, \nabla L_t) \\
&= \frac{1}{T^2} \sum_{b=-M}^{M} (T - |b|) \gamma(b) \\
&= \frac{1}{T} \sum_{b=-M}^{M} \left( 1 - \frac{|b|}{T} \right) \gamma(b) \\
&\xrightarrow{a.s.} 0 \times \left( \sum_{b=-M}^{M} \gamma(b) \right) = 0 \times \Sigma
\end{aligned}
$$

Using the M-dependence assumption, we know M is finite. Since $\frac{|b|}{T} \to 0$ then the finite sum converges to the stationary covariance. Finally using the product of almost sure convergences we conclude.

$\square$

**Theorem 1.** (Forking-Sequences SNR Gains under $M$-Dependence) Consider the forking sequences gradient estimator $\bar{\nabla}L_T = \frac{1}{T}\sum_{t=1}^{T}\nabla L_t$. If gradient samples $\{\nabla L_t\}$ are stationary and $M$-dependent with covariance $\gamma(b) = \text{Cov}(\nabla L_0, \nabla L_b)$, then the Signal to Noise Ratio (SNR) grows linearly with $T$.

*Proof.* Under stationarity, the gradient estimator the signal term is

$$\|\mathbb{E}[\bar{\nabla}L_T]\|_2^2 = \|\mu\|_2^2. \tag{18}$$

From Lemma 1, and taking the trace of the covariance of the estimator

$$\text{Tr}(\text{Cov}(\bar{\nabla}L_T)) = \frac{1}{T}\sum_{b=-M}^{M}\left(1 - \frac{|b|}{T}\right)\text{Tr}(\gamma(b)). \tag{19}$$

Using the $M$-dependence assumption, we have $\gamma(b) = 0$ for $|b| > M$, and the covariance trace becomes:

$$C := \sum_{b=-M}^{M}\text{Tr}(\gamma(b)) < \infty. \tag{20}$$

Then,

$$\text{Tr}(\text{Cov}(\bar{\nabla}L_T)) = \frac{1}{T}\left(C + o(1)\right), \tag{21}$$

Combining signal and noise terms,

$$\text{SNR}_T = \frac{\|\mu\|_2^2}{\text{Tr}(\text{Cov}(\bar{\nabla}L_T))} = \mathcal{O}(T). \tag{22}$$

$\square$

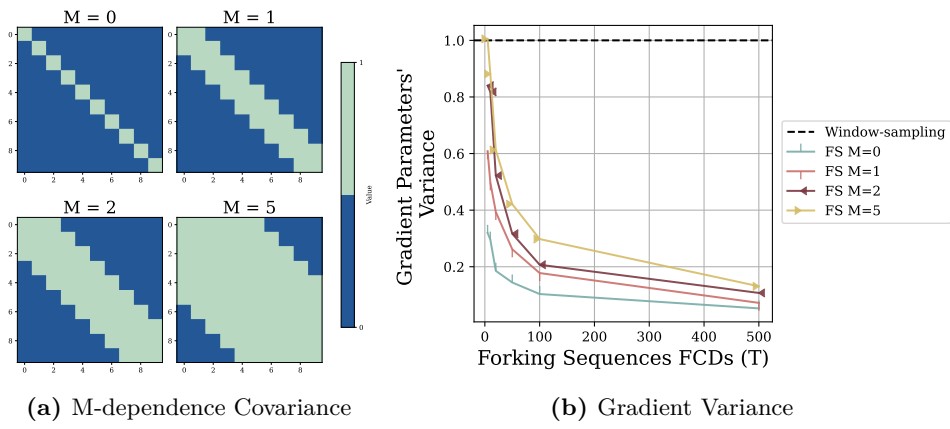

**(a)** M-dependence Covariance  **(b)** Gradient Variance

**Figure 11:** a) Visualization of the covariance of M-dependent random variables. Green sections indicate correlated samples, while blue indicate uncorrelated variables. b) Mean estimator variance reduction as a function of the samples of the forking-sequences training scheme for different levels of M-dependence.

Here we empirically validate the theoretical results from Lemma 1, with synthetic data. We control the M-dependence of a multivariate normal random variable using the covariances in Figure 11a, and then measure the variance of a mean estimator that we report in Figure 11b.

## C  Encoder Gradient Study

We observe that forking-sequences yields the largest performance improvement for the `LSTM` encoder compared to window-sampling. In contrast, forking-sequences has minimal impact on the `Transformer` encoder. We hypothesize that this difference stems from the Transformer's superior gradient flow: with window-sampling, Transformers encode entire context windows at once via self-attention, eliminating sequential information bottlenecks. LSTMs, however, suffer from vanishing gradients during backpropagation through time. Forking-sequences mitigates this issue for the `LSTM` encoder by decoding across all FCDs, which preserves stronger gradients to early timesteps (see Proposition 1). We demonstrate this effect empirically in Fig. 12 and provide theoretical justification in Propositions 1 and 2.

### C.1  Empirical Gradient Analysis

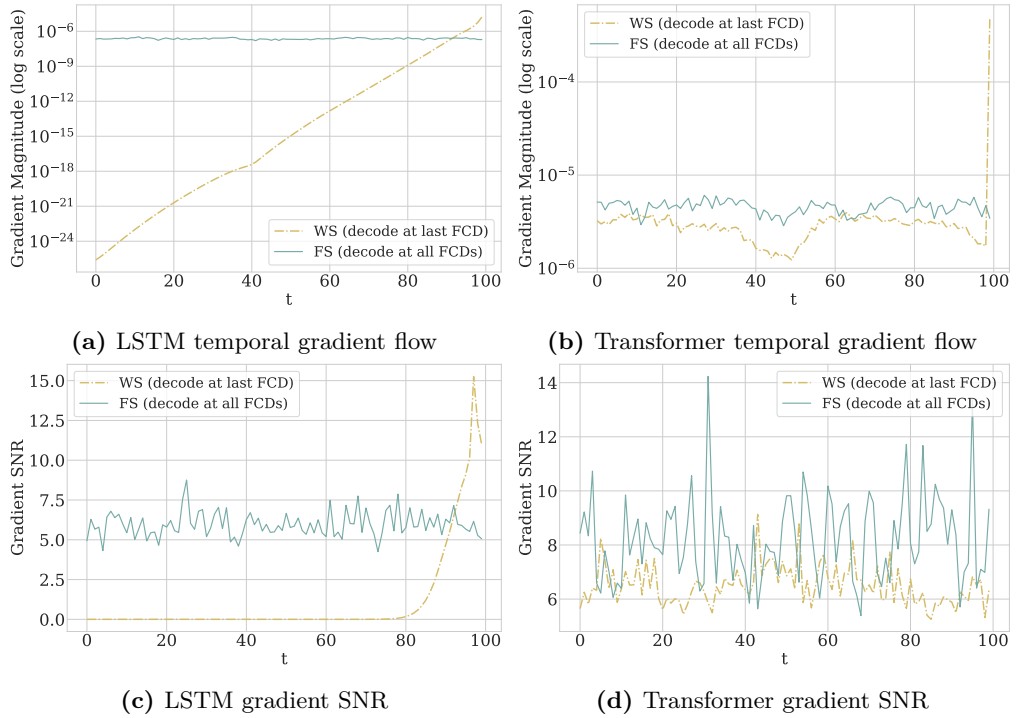

**(a)** LSTM temporal gradient flow    **(b)** Transformer temporal gradient flow

**(c)** LSTM gradient SNR    **(d)** Transformer gradient SNR

**Figure 12: (a)** For the LSTM encoder, forking-sequences (FS) maintains stable gradients across timesteps, while window-sampling (which decodes only at the final FCD) exhibits exponential gradient decay. **(b)** Forking-sequences yields higher gradient signal-to-noise ratio (SNR) for LSTMs, indicating more stable training signals. **(c)** For the Transformer encoder, when losses are averaged by a factor of $\frac{1}{T}$, forking-sequences yields gradient magnitudes comparable to window-sampling. **(d)** Transformer gradient SNR remains relatively stable for both forking-sequences and window-sampling, as Transformers do not suffer from vanishing gradients.

To isolate gradient flow effects, we generate 100 synthetic time series of length $T = 100$ following $y(t) = 0.02t + \sin(t) + 0.5\sin(2t) + \epsilon$ with $\epsilon \sim \mathcal{N}(0, 0.01)$ and $t \in [0, 4\pi]$. We forecast $h = 12$ timesteps ahead using a single-layer LSTM encoder (hidden dimension 32) and a 2-layer Transformer encoder ($d_{\mathrm{model}} = 32$, 4 heads, feedforward dimension 128), both paired with a 2-layer MLP decoder (hidden dimension 64). We measure temporal gradient flow by computing $\left\| \frac{\partial \mathcal{L}}{\partial x_t} \right\|$ at each input position $t \in \{0, \dots, T-1\}$ after backpropagating loss against synthetic random targets. To assess gradient stability, we repeat this 20 times and compute signal-to-noise ratio (SNR) as $\mathrm{SNR}_t = \mu\left( \left\| \frac{\partial \mathcal{L}}{\partial x_t} \right\| \right) / \left[ \sigma\left( \left\| \frac{\partial \mathcal{L}}{\partial x_t} \right\| \right) + \epsilon \right]$ where $\epsilon = 10^{-8}$, with higher SNR indicating more consistent gradient signals. We find that forking-sequences preserves gradient flow for the `LSTM` encoder as shown in Fig. 12. For the `Transformer` encoder, window-sampling and forking-sequences have comparable SNR, which is expected as Transformers are not affected by vanishing gradient issues.

### C.2 Theoretical Gradient Analysis

We supplement these empirical findings with theoretical analysis of gradient flow to early timesteps, specifically gradients at $h_0$, for LSTM and Transformer encoders under both forking-sequences and window-sampling. We begin with the following assumptions:

1. The number of decoded FCDs satisfies $T = 1$ for window-sampling and $T > 1$ for forking-sequences.

2. All loss functions $\mathcal{L}_t$ at FCD $t \in \{1, \dots, T\}$ have similar magnitudes: $\left\| \frac{\partial \mathcal{L}_t}{\partial h_t} \right\| \approx c$ for some constant $c > 0$.

3. For the LSTM encoder, the spectral norm of the hidden state Jacobian satisfies $\left\| \frac{\partial h_t}{\partial h_{t-1}} \right\| \leq \gamma < 1$ for all timesteps $t \in \{1, \dots, T\}$.

**Proposition 1.** (LSTM gradient preservation with forking-sequences.) For an LSTM encoder, forking-sequences with $T$ decoded FCDs provides gradient magnitudes to early timesteps that are bounded by $\sum_{t=1}^{T} \gamma^t$ compared to $\gamma^T$ for window-sampling.

*Proof.* For window-sampling,

$$\frac{\partial \mathcal{L}}{\partial h_0} = \frac{\partial \mathcal{L}}{\partial h_T} \frac{\partial h_T}{\partial h_{T-1}} \frac{\partial h_{T-1}}{\partial h_{T-2}} \cdots \frac{\partial h_1}{\partial h_0} \tag{23}$$

$$= \frac{\partial \mathcal{L}}{\partial h_T} \prod_{j=0}^{T-1} \frac{\partial h_{T-j}}{\partial h_{T-j-1}}. \tag{24}$$

Taking norms and applying the bound $\left\| \frac{\partial h_j}{\partial h_{j-1}} \right\| \leq \gamma$, the gradient magnitude is bounded by

$$\left\| \frac{\partial \mathcal{L}}{\partial h_0} \right\| \leq \gamma^T \left\| \frac{\partial \mathcal{L}}{\partial h_T} \right\|. \tag{25}$$

For forking-sequences, the total loss is

$$\mathcal{L} = \frac{1}{T} \sum_{t=1}^{T} \mathcal{L}_t, \tag{26}$$

where $\mathcal{L}_t$ is the loss from decoding at FCD $t$. We omit standard batch and hidden dimension inverse scaling factors (see Equation 10) as they are shared across forking-sequences and window-sampling.

The gradient to $h_0$ is computed via the chain rule through each decoding path:

$$\frac{\partial \mathcal{L}}{\partial h_0} = \frac{1}{T} \left( \frac{\partial \mathcal{L}_T}{\partial h_T} \frac{\partial h_T}{\partial h_{T-1}} \frac{\partial h_{T-1}}{\partial h_{T-2}} \cdots \frac{\partial h_1}{\partial h_0} \right.$$
$$+ \frac{\partial \mathcal{L}_{T-1}}{\partial h_{T-1}} \frac{\partial h_{T-1}}{\partial h_{T-2}} \frac{\partial h_{T-2}}{\partial h_{T-3}} \cdots \frac{\partial h_1}{\partial h_0}$$
$$\vdots$$
$$\left. + \frac{\partial \mathcal{L}_1}{\partial h_1} \frac{\partial h_1}{\partial h_0} \right) \tag{27}$$

$$= \frac{1}{T} \sum_{t=1}^{T} \frac{\partial \mathcal{L}_t}{\partial h_t} \prod_{j=0}^{t-1} \frac{\partial h_{t-j}}{\partial h_{t-j-1}}. \tag{28}$$

Taking norms and applying the bound $\left\| \frac{\partial h_j}{\partial h_{j-1}} \right\| \leq \gamma$, the gradient magnitude is bounded by

$$\left\| \frac{\partial \mathcal{L}}{\partial h_0} \right\| \leq \frac{1}{T} \sum_{t=1}^{T} \gamma^t \left\| \frac{\partial \mathcal{L}_t}{\partial h_t} \right\|. \tag{29}$$

$\square$

**Proposition 2.** (Transformer gradient scaling with forking-sequences.) Unlike LSTMs, Transformers do not suffer from vanishing gradients, so forking-sequences provides only a linear scaling factor of the number of FCDs $T$ rather than the exponential improvement observed in LSTMs. When losses are averaged by a factor of $\frac{1}{T}$, forking-sequences yields gradient magnitudes comparable to window-sampling.

*Proof.* In Transformer, each token attends to all tokens with self-attention. Let us assume each timestep $t$ is an individual token. The hidden state at $t$ is computed as

$$h_t = \text{Attention}(h_0, h_1, \ldots, h_T). \tag{30}$$

This attention mechanism ensures that $\frac{\partial h_i}{\partial h_j}$ does not decay with distance $|i - j|$.

For window-sampling that decodes at a single FCD $T$,

$$\frac{\partial \mathcal{L}}{\partial h_0} = \frac{\partial \mathcal{L}}{\partial h_T} \frac{\partial h_T}{\partial h_0}, \tag{31}$$

$$\tag{32}$$

and so the gradient magnitude is bounded by

$$\left\| \frac{\partial \mathcal{L}}{\partial h_0} \right\| \leq \left\| \frac{\partial \mathcal{L}}{\partial h_T} \right\| \left\| \frac{\partial h_T}{\partial h_0} \right\|. \tag{33}$$

For forking-sequences, the total loss is

$$\mathcal{L} = \frac{1}{T} \sum_{t=1}^{T} \mathcal{L}_t, \tag{34}$$

where $\mathcal{L}_t$ is the loss from decoding at FCD $t$. We omit standard batch and hidden dimension inverse scaling factors (see Equation 10) as they are shared across forking-sequences and window-sampling methods.

The gradient to $h_0$ receives contributions from all decoded FCDs:

$$\frac{\partial \mathcal{L}}{\partial h_0} = \frac{1}{T} \left( \frac{\partial \mathcal{L}_T}{\partial h_T} \frac{\partial h_T}{\partial h_0} + \frac{\partial \mathcal{L}_{T-1}}{\partial h_{T-1}} \frac{\partial h_{T-1}}{\partial h_0} + \cdots + \frac{\partial \mathcal{L}_1}{\partial h_1} \frac{\partial h_1}{\partial h_0} \right) \tag{35}$$

$$= \frac{1}{T} \sum_{t=1}^{T} \frac{\partial \mathcal{L}_t}{\partial h_t} \frac{\partial h_t}{\partial h_0}. \tag{36}$$

$$\tag{37}$$

The gradient magnitude is bounded by

$$\left\| \frac{\partial \mathcal{L}}{\partial h_0} \right\| \leq \frac{1}{T} \sum_{t=1}^{T} \left\| \frac{\partial \mathcal{L}_t}{\partial h_t} \right\| \left\| \frac{\partial h_t}{\partial h_0} \right\|. \tag{38}$$

$\square$

Unlike LSTMs, where forking-sequences provides exponential improvement over window-sampling by mitigating vanishing gradients, Transformers do not suffer from vanishing gradients. Averaging the losses by $\frac{1}{T}$ for forking-sequences compensates for the linear accumulation from multiple gradient paths, resulting in similar gradient magnitudes at early timesteps for both Transformers with forking-sequences and window-sampling. This result is demonstrated empirically in Fig. 12.

# D Forking-Sequences Convergence Speed Ablation Study

## D.1 Linear Model Experiment

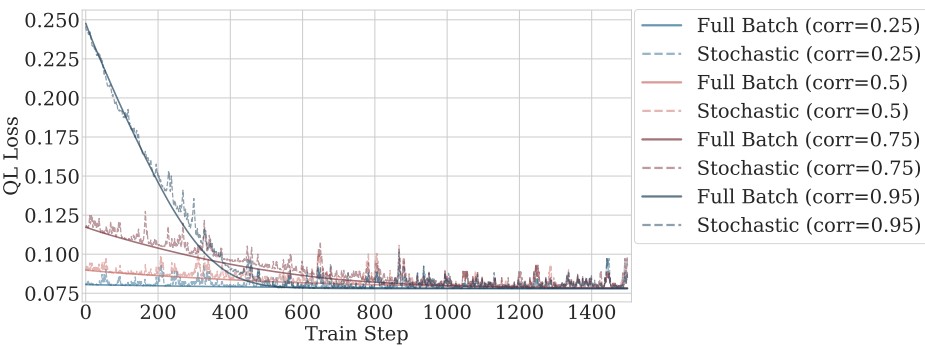

**Figure 13:** Training dynamics comparison between full-batch and SGD approaches across different autocorrelation levels. Full-batch training exhibits faster and more stable convergence compared to SGD across all scenarios. Models trained on highly autocorrelated time series ($\rho = 0.95$) show higher initial training loss but demonstrate steeper convergence rates, compared to those with lower autocorrelation ($\rho = 0.25$). This behavior illustrates how temporal dependencies influence optimization dynamics, with implications for batch processing strategies in time series forecasting.

We conducted an ablation study comparing full-batch gradient descent (GD) versus stochastic gradient descent (SGD) training approaches in time series forecasting. This comparison provides insights relevant to forking sequences implementations, as the key distinction between batch sizes during training (all samples vs. single sample) parallels the training schemes of forking-sequences and window-sampling, respectively.

The experimental setup involved four scenarios with varying temporal dependency levels (autocorrelation coefficients $\rho \in 0.25, 0.5, 0.75, 0.95$) following the data-generating process:

$$X_t = \rho X_{t-1} + \varepsilon_t \quad \text{where} \quad \varepsilon_t \sim \mathcal{N}(0, \sigma^2). \tag{39}$$

Higher autocorrelation values ($\rho$) indicate stronger temporal dependencies between consecutive time steps. Training was conducted over 1,500 iterations, with full-batch gradient descent processing all 1,000 samples simultaneously, while SGD processed one sample per iteration. This experimental design helps understand the impact of batch size on training dynamics across different temporal dependency structures. Experimental parameters are included in Table 4.

**Table 4:** Parameters for Linear Autoregressive Model.

| | |
|---|---|
| Horizon | 1 |
| Input size (lag) | 1 |
| Batch size (number of time series) | 1 |
| Random seed | 42 |
| Maximum Train Steps | 1,500 |
| Loss quantiles | 0.5 |
| Learning rate | 0.01 |
| Sample Size | T (Full) or 1 (SGD) |
| Number of samples (T) | 1,000 |
| Noise level | 0.1 |
| Autocorrelation levels ($\rho$) | {0.25, 0.5, 0.75, 0.95} |
| Loss function | MAE |

## D.2 Deep Learning Model Experiments

The use of forking-sequences leads to faster training convergence across all models and datasets compared to window-sampling, as observed in Figure 14.

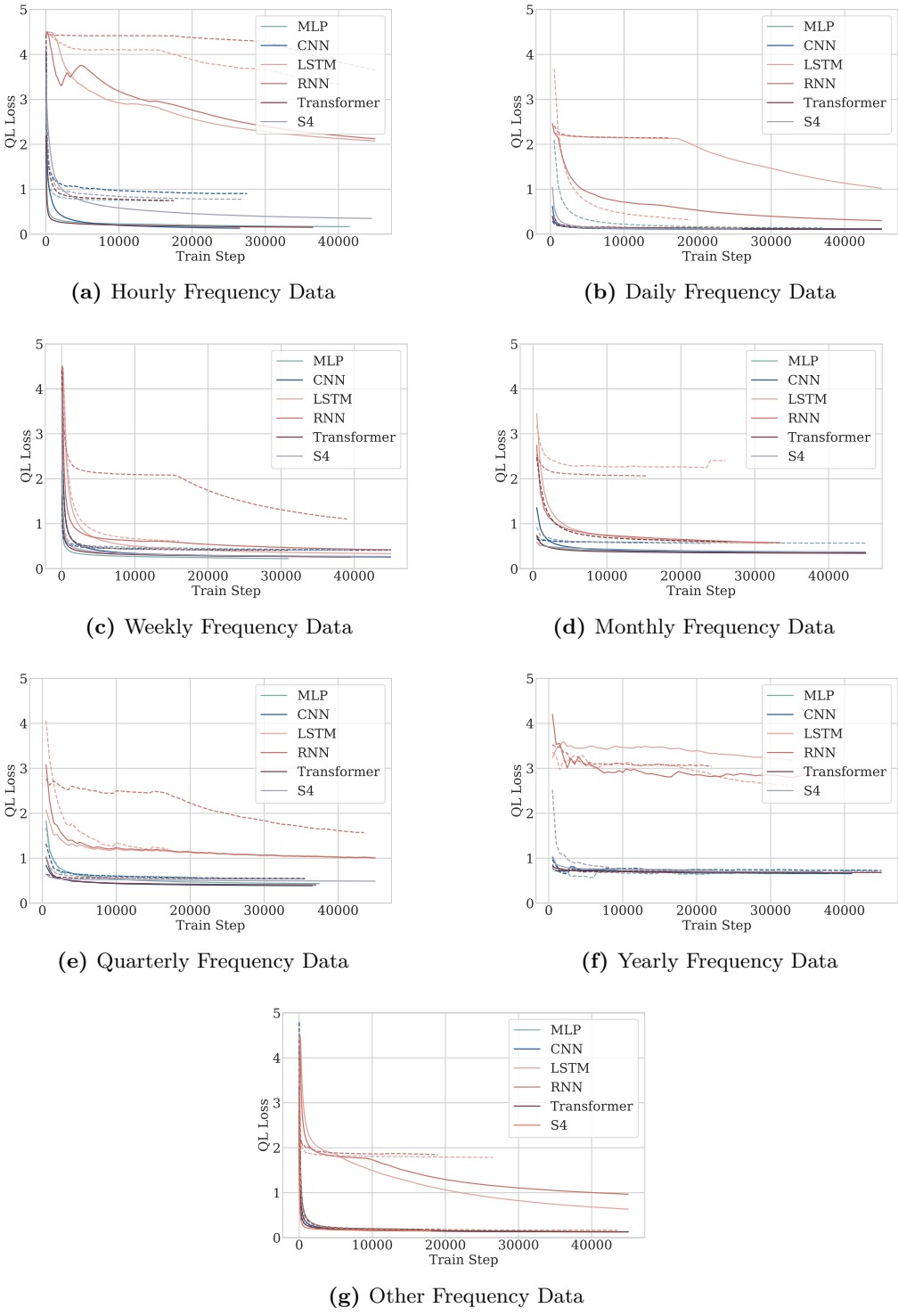

**(a)** Hourly Frequency Data

**(b)** Daily Frequency Data

**(c)** Weekly Frequency Data

**(d)** Monthly Frequency Data

**(e)** Quarterly Frequency Data

**(f)** Yearly Frequency Data

**(g)** Other Frequency Data

**Figure 14:** Deep Learning optimization convergence using either *forking-sequences* (solid) or *window-sampling* (dashed) techniques. Figures show training quantile loss for the train set as a function of train step.

# E Theoretical Encoder Computational Complexity

We provide propositions and supporting proofs for various encoder types to supplement the results presented in Table 1. We begin with the following assumptions:

1. The input is a univariate time series $\mathbf{y} \in \mathbb{R}$.

2. Hidden sizes ($d$ for RNNs, $d$ for CNN output channels, $d_1$, $d_2$ for MLP layers) and kernel sizes $k$ are treated as fixed constants, independent of $T$.

3. Window-sampling (WS)-restricted uses a fixed window size $L < T$. WS-full uses $T$ timesteps of history.

4. For forking-sequences (FS), the encoder processes the entire sequence of input length $T$.

## E.1 Multi-Layer Perceptrons (MLPs)

**Proposition 3** (Complexity of MLP Encoders). For an `MLP` encoder with fixed hidden dimensions and window size $L$, the computational complexity scales as follows: FS: $\mathcal{O}(T)$, WS-restricted: $\mathcal{O}(TL)$, and WS-full: $\mathcal{O}(T^2)$.

*Proof.* Given a 2-layer `MLP`,

$$\mathbf{h}_{t,[h]} = \sigma(W_2\sigma(W_1\mathbf{y}_{t-L+1:t} + b_1) + b_2) \tag{40}$$

where $\mathbf{y}_{t-L+1:t} \in \mathbb{R}^L$ is a window of $L$ consecutive observations, $W_1 \in \mathbb{R}^{d_1 \times L}$ flattens and projects the window to $d_1$ dimensions, $W_2 \in \mathbb{R}^{d_2 \times d_1}$ projects to $d_2$ dimensions, and $\mathbf{h}_{t,[h]} \in \mathbb{R}$ is the encoder output (hidden state) at forecast creation date $t$ for forecast horizon $h$.

For a single encoder computation on a window of size $L$, the complexity is $\mathcal{O}(d_1L + d_2d_1)$. Processing $T$ FCDs yields total complexity $\mathcal{O}(T(d_1L + d_2d_1))$.

Treating $L$, $d_1$, and $d_2$ as fixed architectural constants, the complexity for FS is $\mathcal{O}(T)$. Similarly, for WS-restricted, since each encoder computation processes a window of size $L$, the complexity is $\mathcal{O}(TL)$.

For WS-full, each encoder computation processes a window of size up to $T$, where $W_1 \in \mathbb{R}^{d_1 \times T}$. The cost per computation becomes $\mathcal{O}(d_1T)$, and with $T$ encoder computations, the total complexity is $\mathcal{O}(T \cdot d_1T) = \mathcal{O}(T^2)$. □

## E.2 Convolutional Encoders

**Proposition 4** (Complexity of Convolutional Encoders). For a convolutional encoder with fixed kernel size $k$ and channel dimension, the computational complexity scales as follows: FS: $\mathcal{O}(T)$, WS-restricted: $\mathcal{O}(TL)$, WS-full: $\mathcal{O}(T^2)$

*Proof.* Given a 1D `CNN` encoder with kernel size $k$ and a single output channel, a single convolution operation at timestep $t$ produces the encoder output,

$$\mathbf{h}_{t,[h]} = \sigma\left(W * \mathbf{y}_{t-k+1} + b\right), \tag{41}$$

where $W \in \mathbb{R}^{1 \times k}$ is the convolutional kernel, $b \in \mathbb{R}$ is the bias, $*$ denotes 1D convolution, $\sigma(\cdot)$ is a nonlinearity, and $\mathbf{h}_{t,[h]} \in \mathbb{R}$ is the encoder output (hidden state) at forecast creation date $t$ for forecast horizon $h$.

Processing the entire sequence of length $T$ requires applying the convolution $T$ times with kernel size $k$, yielding total complexity $\mathcal{O}(T \cdot k \cdot d) = \mathcal{O}(T)$, treating $k$ and $d$ as fixed constants.

For WS-restricted, each encoder evaluation sees a fixed window of length $L$, so the convolution is applied $L$ times per evaluation. With $T$ evaluations, the total complexity is $\mathcal{O}(T \cdot L \cdot k \cdot d) = \mathcal{O}(TL)$.

For WS-full, each encoder process processes a window of size up to $T$, giving complexity of $\mathcal{O}(T \cdot k \cdot d)$. With $T$ processes of the encoder, the overall complexity is $\mathcal{O}(T \cdot T \cdot k \cdot d) = \mathcal{O}(T^2)$. □

### E.3 Recurrent Encoders

**Proposition 5** (Complexity of RNN Encoders). *For a recurrent encoder with hidden size $d$ treated as constant, the complexity is FS: $\mathcal{O}(T)$, WS-restricted: $\mathcal{O}(TL)$, and WS-full: $\mathcal{O}(T^2)$.*

*Proof.* Considering an `LSTM` encoder, which processes input sequentially via:

$$\mathbf{i}_t = \sigma(W_{ii}y_t + \mathbf{b}_{ii} + W_{hi}\mathbf{h}_{t-1} + \mathbf{b}_{hi}) \quad \text{(input gate)} \tag{42}$$

$$\mathbf{f}_t = \sigma(W_{if}y_t + \mathbf{b}_{if} + W_{hf}\mathbf{h}_{t-1} + \mathbf{b}_{hf}) \quad \text{(forget gate)} \tag{43}$$

$$\mathbf{g}_t = \tanh(W_{ig}y_t + \mathbf{b}_{ig} + W_{hg}\mathbf{h}_{t-1} + \mathbf{b}_{hg}) \quad \text{(candidate cell state)} \tag{44}$$

$$\mathbf{o}_t = \sigma(W_{io}y_t + \mathbf{b}_{io} + W_{ho}\mathbf{h}_{t-1} + \mathbf{b}_{ho}) \quad \text{(output gate)} \tag{45}$$

$$\mathbf{c}_t = \mathbf{f}_t \odot \mathbf{c}_{t-1} + \mathbf{i}_t \odot \mathbf{g}_t \quad \text{(cell state update)} \tag{46}$$

$$\mathbf{h}_{t,[h]} = \mathbf{o}_t \odot \tanh(\mathbf{c}_t) \quad \text{(hidden state)}, \tag{47}$$

where $y_t \in \mathbb{R}$, $\mathbf{h}_{t,[h]} \in \mathbb{R}^d$, $\mathbf{c}_t \in \mathbb{R}^d$, $\mathbf{h}_{t-1}$ is the hidden state of the layer at time t-1 or the initial hidden state at time 0, $\sigma$ is the sigmoid function, and $\odot$ is the Hadamard product.

At each timestep, the LSTM computes four gates, each requiring matrix-vector multiplications costing $\mathcal{O}(d^2)$ operations. Treating hidden dimension $d$ as constant, processing each timestep costs $\mathcal{O}(1)$. □

With forking-sequence, the recurrent encoder consumes the sequence sequentially in a single forward pass, resulting in $\mathcal{O}(T)$ total computation. However, WS-restricted requires recomputing encoder states over length-$L$ sequences at each timestep, incurring $\mathcal{O}(TL)$ total complexity. Similarly, WS-full recomputes over the entire history $T$, leading to $\mathcal{O}(T^2)$

### E.4 Self-Attention (Transformers)

**Proposition 6** (Complexity of Transformer Encoders). *For a self-attention encoder with hidden dimension $d$, the computational complexity scales as FS: $\mathcal{O}(T^2)$, WS-restricted: $\mathcal{O}(TL^2)$, and WS-full: $\mathcal{O}(T^3)$.*

*Proof.* Self-attention computes attention scores between all pairs of positions:

$$\mathbf{h}_{t,[h]} = \text{Attention}(Q, K, V) = \text{SoftMax}\left(\frac{QK^T}{\sqrt{d_k}}\right) V \tag{48}$$

where $Q, K, V \in \mathbb{R}^{T \times d}$ are the query, key, and value matrices, and $\mathbf{h}_{t,[h]} \in \mathbb{R}^d$ is the encoder output (hidden state) at forecast creation date $t$ for forecast horizon $h$. The dominant operation is computing $QK^T$, which costs $\mathcal{O}(T^2 d)$. Since $d$ is constant, the overall complexity of self-attention for FS is $\mathcal{O}(T^2)$.

For WS-restricted, self-attention is applied within each window of size $L$, costing $\mathcal{O}(L^2)$ per window. With $T$ encoder processes, the total complexity is $\mathcal{O}(TL^2)$. The computation of attention matrices of $\mathcal{O}(T^2)$ or $\mathcal{O}(L^2)$ can be recycled with the use of causal masks without additional computation.

If implemented in a window-based sampling manner, each subsequence would require its own self-attention pass. In the optimistic case—where attention is applied only to each progressively shorter subsequence; the total cost is $T^2 + (T-1)^2 + \cdots + 1^2 = \mathcal{O}(T^3)$. In practice, shorter windows are typically padded to a common length, meaning each of the $T$ windows incurs a full $T^2$ attention cost, again yielding an overall complexity of $\mathcal{O}(T^3)$. In other words, WS-full requires $T$ encoder computations over windows of average size $\mathcal{O}(T)$, and because self-attention on a window of length $T$ is $\mathcal{O}(T^2)$, the total encoder cost remains cubic in $T$.

□

### E.5 Structured State Space (S4)

**Proposition 7** (Complexity of S4 Encoders). *For a structured state space encoder with state dimension $N$ and hidden dimension $d$, the computational complexity scales as FS: $\mathcal{O}(T \log T)$, WS-restricted: $\mathcal{O}(TL \log L)$, and WS-full: $\mathcal{O}(T^2 \log T)$.*

*Proof.* The S4 model processes input through a continuous-time state space:

$$x'(t) = Ax(t) + Bu(t) \tag{49}$$
$$y(t) = Cx(t) + Du(t) \tag{50}$$

which is discretized with step size $\Delta$.

S4 computes the output as a convolution $\mathbf{h}_{t,[h]} = (\bar{K} * u) + Du$, where $\mathbf{h}_{t,[h]} \in \mathbb{R}^d$ is the encoder output (hidden state) at forecast creation date $t$ for forecast horizon $h$, and the kernel $\bar{K} \in \mathbb{R}^T$ is:

$$\bar{K}_t = C \cdot \frac{e^{\Delta A} - 1}{A} \cdot e^{t\Delta A} \tag{51}$$

Computing the kernel for $T$ timesteps requires $\mathcal{O}(TN)$ operations. The convolution is computed via FFT in $\mathcal{O}(T \log T)$ time. Treating $N$ as constant, the FFT dominates, giving $\mathcal{O}(T \log T)$ complexity.

For FS, the encoder processes the entire sequence of length $T$ once, so the kernel computation $\mathcal{O}(T)$ plus FFT-based convolution $\mathcal{O}(T \log T)$ yields total complexity $\mathcal{O}(T \log T)$.

For WS-restricted, each encoder computation processes a fixed window of size $L$, costing $\mathcal{O}(L \log L)$ per window. With $T$ encoder computations to generate forecasts at each FCD, the total complexity is $\mathcal{O}(TL \log L)$.

For WS-full, each encoder computation processes a window of size up to $T$, costing $\mathcal{O}(T \log T)$ per computation. With $T$ encoder computations, the total complexity is $\mathcal{O}(T^2 \log T)$. □

# F   Excess Volatility Metric

Recall the *Excess Volatility* (EV) definition in Equation 12 from Section 3.6, here we provide the proof of its convenient properties.

$$\text{EV}(y, \ \hat{\mathbf{y}}_1, \ \hat{\mathbf{y}}_2) = \text{QL}(\hat{\mathbf{y}}_2, \hat{\mathbf{y}}_1) - (\text{QL}(y, \hat{\mathbf{y}}_1) - \text{QL}(y, \hat{\mathbf{y}}_2)). \tag{52}$$

**Theorem 2.** (*Excess Volatility Metric Properties*) Consider ground truth $y$, and two consecutive forecasts $\hat{\mathbf{y}}_1, \hat{\mathbf{y}}_2$, we prove in Appendix F the EV metric has the following properties:

- **Positivity:** $\text{EV}(y, \ \hat{\mathbf{y}}_1, \ \hat{\mathbf{y}}_2) \geq 0$

- **Zero-Penalty for Improving Revisions:** When the revision from $\hat{\mathbf{y}}_1$ to $\hat{\mathbf{y}}_2$ represents a direct, proportional improvement, and $\hat{\mathbf{y}}_2$ lies on the linear path from $y$ to $\hat{\mathbf{y}}_1$ then $\text{EV}(y, \ \hat{\mathbf{y}}_1, \ \hat{\mathbf{y}}_2) = 0$. An improved revision example is shown in Fig. 6a.

- **Maximum penalty for Deteriorating Revisions:** When $\hat{\mathbf{y}}_1$ lies on the direct path between $y$ and $\hat{\mathbf{y}}_2$; that is, when the revision from $\hat{\mathbf{y}}_1$ to $\hat{\mathbf{y}}_2$ moves linearly away from the truth, then $\text{EV}(y, \ \hat{\mathbf{y}}_1, \ \hat{\mathbf{y}}_2) = \text{QL}(y, \hat{\mathbf{y}}_2) - \text{QL}(y, \hat{\mathbf{y}}_1)$. In this fully misdirected case, EV assigns the maximum penalty, equal to the entire accuracy degradation. A deteriorating revision example is shown in Fig. 6b.

- **Overshoot-Revision Penalty Property:** When $y$ lies on the line segment between $\hat{\mathbf{y}}_1$ and $\hat{\mathbf{y}}_2$—that is, when $\hat{\mathbf{y}}_2$ revises $\hat{\mathbf{y}}_1$ in the correct direction but overshoots—then $\text{EV} = \text{QL}(y, \hat{\mathbf{y}}_2)$. In this excess-revision case, EV penalizes only the overshoot. An overshoot revision example is shown in Fig. 6c.

*Proof.* All the above properties can be proved by cases, as shown in Table 5 below:

| Case | Sub-case | $L_q(y_2,y_1)$ | $L_q(y,y_1)$ | $L_q(y,y_2)$ | EV |
|---|---|---|---|---|---|
| $\|y - y_1\| = \|y - y_2\| + \|y_2 - y_1\|$ | $y \leq y_2 \leq y_1$ | $(1-q)(y_1 - y_2)$ | $(1-q)(y_1 - y)$ | $(1-q)(y_2 - y)$ | $0$ |
| | $y_1 \leq y_2 \leq y$ | $q(y_2 - y_1)$ | $q(y - y_1)$ | $q(y - y_2)$ | $0$ |
| $\|y - y_2\| = \|y - y_1\| + \|y_1 - y_2\|$ | $y \leq y_1 \leq y_2$ | $q(y_2 - y_1)$ | $(1-q)(y_1 - y)$ | $(1-q)(y_2 - y)$ | $y_2 - y_1$ |
| | $y_2 \leq y_1 \leq y$ | $(1-q)(y_1 - y_2)$ | $q(y - y_1)$ | $q(y - y_2)$ | $y_1 - y_2$ |
| $\|y_1 - y_2\| = \|y_1 - y\| + \|y - y_2\|$ | $y_1 \leq y \leq y_2$ | $q(y_2 - y_1)$ | $q(y - y_1)$ | $(1-q)(y_2 - y)$ | $y_2 - y$ |
| | $y_2 \leq y \leq y_1$ | $(1-q)(y_1 - y_2)$ | $(1-q)(y_1 - y)$ | $q(y - y_2)$ | $y - y_2$ |

**Table 5:** Proof by cases.

$\square$

## G   Dataset Details

In this Appendix, we describe the datasets used in our analysis.

**Table 6:** Summary of forecasting datasets used in our empirical study.

|         | Frequency | Seasonality | Horizon | Series | Min Length | Max Length | % Erratic |
|---------|-----------|-------------|---------|--------|------------|------------|-----------|
| M1      | Monthly   | 12          | 18      | 617    | 48         | 150        | 0         |
|         | Quarterly | 4           | 8       | 203    | 18         | 114        | 0         |
|         | Yearly    | 1           | 6       | 181    | 15         | 58         | 0         |
| M3      | Other     | 4           | 8       | 174    | 71         | 104        | 0         |
|         | Monthly   | 12          | 18      | 1428   | 66         | 144        | 2         |
|         | Quarterly | 4           | 8       | 756    | 24         | 72         | 1         |
|         | Yearly    | 1           | 6       | 645    | 20         | 47         | 10        |
| M4      | Hourly    | 24          | 48      | 414    | 748        | 1008       | 17        |
|         | Daily     | 1           | 14      | 4,227  | 107        | 9933       | 2         |
|         | Weekly    | 1           | 13      | 359    | 93         | 2610       | 16        |
|         | Monthly   | 12          | 18      | 48,000 | 60         | 2812       | 6         |
|         | Quarterly | 4           | 8       | 24,000 | 24         | 874        | 11        |
|         | Yearly    | 1           | 6       | 23,000 | 19         | 841        | 18        |
| Tourism | Monthly   | 12          | 18      | 366    | 91         | 333        | 51        |
|         | Quarterly | 4           | 8       | 427    | 30         | 130        | 39        |
|         | Yearly    | 1           | 6       | 518    | 11         | 47         | 23        |

**M1 Dataset Details.**   The early M1 competition (Makridakis et al., 1982) focused on 1,001 time series drawn from demography, industry, and economics, with lengths ranging from 9 to 132 observations and varying in frequency (monthly, quarterly, and yearly). A key empirical finding of this competition was that simple forecasting methods, such as ETS (Holt, 1957), often outperformed more complex approaches. These results had a lasting impact on the field, initiating a research legacy that emphasized accurate forecasting, model automation, and caution against overfitting. The competition also marked a conceptual shift, helping to distinguish time-series forecasting from traditional time series analysis.

**M3 Dataset Details.**   The M3 competition (Makridakis and Hibon, 2000), held two decades after the M1competition, featured a dataset of 3,003 time series spanning business, demography, finance and economics. These series ranged from 14 to 126 observations and included monthly, quarterly, and yearly frequencies. All series had positive values, with only a small proportion displaying erratic behavior and none exhibiting intermittency (Syntetos et al., 2005). The M3competition reinforced the trend of simple forecasting methods outperforming more complex alternatives, with the `Theta` method (Hyndman and Billah, 2003) emerging as the best performing approach.

**M4 Dataset Details.**   The M4 competition marked a substantial increase in both the size and diversity of the M competition datasets, comprising 100,000 time series across six frequencies: hourly, daily, weekly, monthly, quarterly, and annual. These series covered a wide range of domains, including demography, finance, industry, and both micro- and macroeconomic indicators. The competition also introduced the evaluation of prediction intervals in addition to point forecasts, broadening the assessment criteria. M4's proportion of non-smooth or erratic time series increased to 18 percent (Syntetos et al., 2005). For the first time, a neural forecasting model - ESRNN(Smyl, 2019) - outperformed traditional methods.

**Tourism Dataset Details.**   The Tourism dataset (Athanasopoulos et al., 2011) was designed to evaluate forecasting methods applied to tourism demand data across multiple temporal frequencies. It comprises 1,311 time series at monthly, quarterly, and yearly frequencies. This competition introduced the Mean Absolute Scaled Error (MASE) as an alternative metric to evaluate scaled point forecasts, alongside the evaluation of forecast intervals. Notably, 36% of the series were classified as erratic or intermittent. Due to this high proportion of irregular data, the Naïve1 method proved particularly difficult to outperform at the yearly frequency.

# H   Training Methodology and Hyperparameters

In this Appendix, we expand on the training methodology outlined in Section 4. We conducted all neural network experiments using a single AWS p4d.24xlarge with 1152 GiB of RAM and 96 vCPUs.

## H.1   Trained Models

We train multi-quantile loss (MQ) models varying only the encoder type to include MLP, RNN, LSTM, CNN, Transformer, and S4 modules. We train models with hyperparameter tuning using Optuna. Using 5 trials, we train individual models and select hyperparameters that minimize validation loss using the multi-quantile loss function defined in Eqn. 5. Training times mostly depend on the architecture; however, we restrict the SGD training steps to 45K per architecture. Tables 7 and 8–13 list the shared and encoder-specific hyperparameters, respectively.

**Table 7:** Shared hyperparameters across encoder types

| HYPERPARAMETER | CONSIDERED VALUES |
|---|---|
| Single GPU SGD Batch Size[*] | $\{4, 8, 16\}$ |
| Initial learning rate | loguniform(1e-4, 5e-2) |
| Maximum Training steps $S_{max}$ | 45,000 |
| Learning rate decay | 0.1 |
| Learning rate step size | 15,000 |
| Scaler type | Standard |
| H-Agnostic Decoder Dimension | 100 |
| H-Specific Decoder Dimension | 20 |

**Table 8:** MLP

| HYPERPARAMETER | CONSIDERED VALUES |
|---|---|
| Main Activation Function | ReLU |
| Encoder Dimension | $\{128, 256\}$ |
| Number of Layers | $\{3, 4\}$ |
| Input size | $\{1, 8, 32, 96\}$ |

**Table 9:** RNN

| HYPERPARAMETER | CONSIDERED VALUES |
|---|---|
| Main Activation Function | tanh |
| Encoder Dimension | $\{128, 256\}$ |
| Dilations | $\{[[1,2], [4,8]],$ $[[1,2], [4,8], [16,32]]\}$ |

**Table 10:** LSTM

| HYPERPARAMETER | CONSIDERED VALUES |
|---|---|
| Main Activation Functions | Sigmoid, tanh |
| Encoder Dimension | $\{128, 256\}$ |
| Dilations | $\{[[1,2], [4,8]],$ $[[1,2], [4,8], [16,32]]\}$ |

**Table 11:** CNN

| HYPERPARAMETER | CONSIDERED VALUES |
|---|---|
| Main Activation Function | ReLU |
| Encoder Dimension | $\{128, 256\}$ |
| Temporal Convolution Kernel Size | $\{2, 5\}$ |
| Temporal Convolution Dilations | $\{[1, 2, 4, 8],$ $[1, 2, 4, 8, 16, 32]\}$ |

**Table 12:** Transformer

| HYPERPARAMETER | CONSIDERED VALUES |
|---|---|
| Main Activation Functions | Sigmoid, tanh |
| Encoder Dimension | $\{128, 256\}$ |
| Projection Dimension | $\{128, 256\}$ |
| Input Patch Lengths | $\{1, 8, 32, 96\}$ |
| Attention Dilations | $\{[2, 6, 8], [2, 4, 8, 16]\}$ |
| Attention Dropout | 0.1 |
| Number of Attention Heads | 4 |

**Table 13:** State Space S4

| HYPERPARAMETER | VALUES |
|---|---|
| Main Activation Function | GELU |
| Encoder Dimension | $\{128, 256\}$ |
| Projection Hidden Size | $\{128, 256\}$ |

## H.2 Pretrained Models

For our implementations of `NBEATS` (Oreshkin et al., 2020) and `PatchTST` (Nie et al., 2023) pre-trained on a synthetic dataset, model parameters were set to the defaults of the original implementations in the NeuralForecast library (Olivares et al., 2022). Open-source model checkpoints were used for `Chronos-2`, `TimesFM`, and `Toto 2.0`. All models were set to output quantile predictions with quantiles: $\{0.1, 0.2, 0.3, 0.4, 0.5, 0.6, 0.7, 0.8, 0.9\}$.

**Table 14:** `Toto 2.0`

| HYPERPARAMETER | VALUES |
| --- | --- |
| Checkpoint | Datadog/Toto-2.0-313m |
| Input size | 512 |

**Table 15:** `Chronos-2`

| HYPERPARAMETER | VALUES |
| --- | --- |
| Checkpoint | amazon/chronos-2 |
| Input size | 512 |

**Table 16:** `TimesFM`

| HYPERPARAMETER | VALUES |
| --- | --- |
| Checkpoint | google/timesfm-2.5-200m |
| Max input size | 512 |
| Max horizon | 128 |
| Per-Core batch size | 256 |
| Normalize inputs | True |
| Continuous quantile head | True |
| Force flip invariance | False |
| Infer is positive | False |
| Fix quantile crossing | False |
| Return backcast | False |

**Table 17:** `NBEATS`

| HYPERPARAMETER | VALUES |
| --- | --- |
| Max steps | 100000 |
| Validation check steps | 300 |
| Random seed | 42 |
| Learning rate | 0.001 |
| Learning rate decay | 0.9 |
| Learning rate step size | 5,000 |
| Scaler Type | Absolute max |
| Input Size | 4096 |
| Max horizon | 512 |
| Layer Width | 1024 |
| Number of Blocks | 10 |
| Number of Layers | 3 |

**Table 18:** `PatchTST`

| HYPERPARAMETER | VALUES |
| --- | --- |
| Max steps | 100000 |
| Validation check steps | 300 |
| Random seed | 42 |
| Learning rate | 0.001 |
| Learning rate decay | 0.9 |
| Learning rate step size | 5,000 |
| Scaler Type | Absolute max |
| Input Size | 4096 |
| Max horizon | 512 |
| Activation | GELU |
| Encoder layers | 6 |
| Hidden size | 512 |
| Learn positional encoding | True |
| Linear hidden size | 1024 |
| Number of heads | 8 |
| Normalization | LayerNorm |
| Patch length | 64 |
| Positional encoding | zeros |
| Residual attention | True |
| RevIN | True |
| RevIN affine | True |
| Stride | 32 |

# I  Ensembling Ablation Studies

We propose the use of ensembling techniques, which aggregate predictions for each target date across FCDs to reduce forecast variance, for model inference with the forking-sequences scheme.

## I.1  Forecast Ensembling Impact on Trained Model Predictions

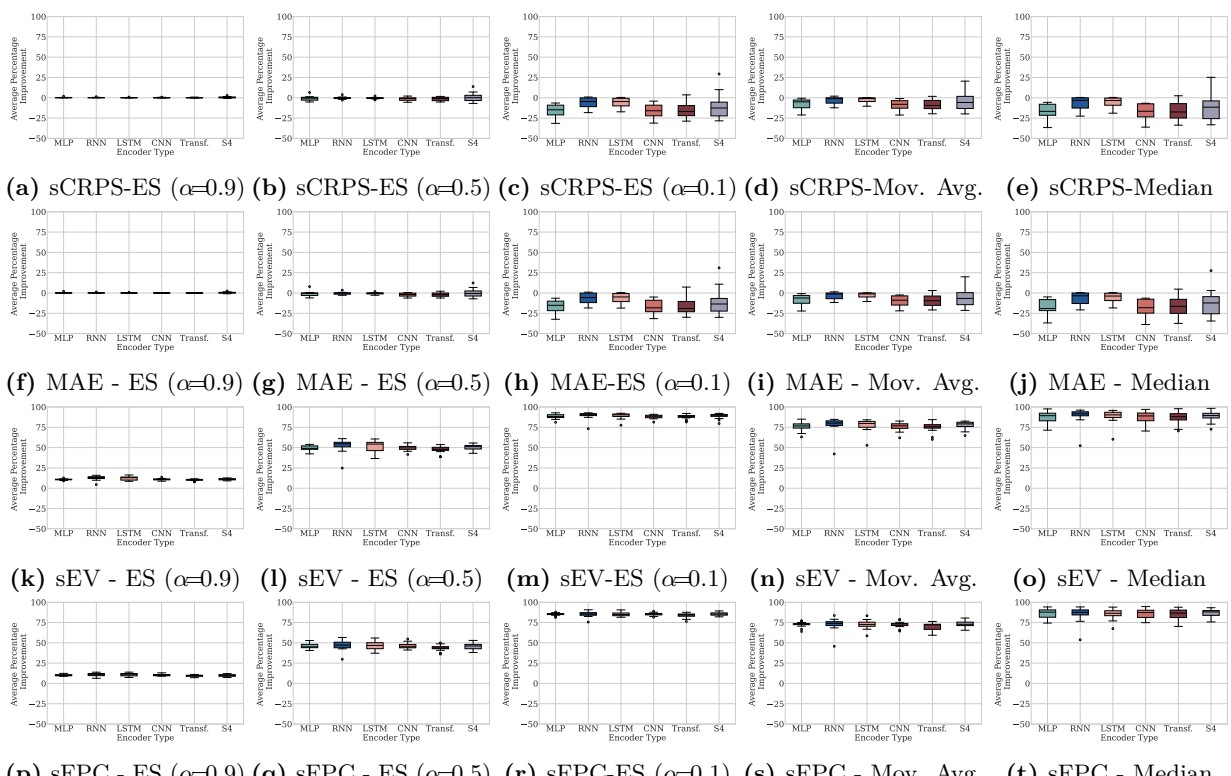

**(a)** sCRPS-ES ($\alpha$=0.9)  **(b)** sCRPS-ES ($\alpha$=0.5)  **(c)** sCRPS-ES ($\alpha$=0.1)  **(d)** sCRPS-Mov. Avg.  **(e)** sCRPS-Median

**(f)** MAE - ES ($\alpha$=0.9)  **(g)** MAE - ES ($\alpha$=0.5)  **(h)** MAE-ES ($\alpha$=0.1)  **(i)** MAE - Mov. Avg.  **(j)** MAE - Median

**(k)** sEV - ES ($\alpha$=0.9)  **(l)** sEV - ES ($\alpha$=0.5)  **(m)** sEV-ES ($\alpha$=0.1)  **(n)** sEV - Mov. Avg.  **(o)** sEV - Median

**(p)** sFPC - ES ($\alpha$=0.9)  **(q)** sFPC - ES ($\alpha$=0.5)  **(r)** sFPC-ES ($\alpha$=0.1)  **(s)** sFPC - Mov. Avg.  **(t)** sFPC - Median

**Figure 15:** Comparative performance gains achieved through diverse ensembling techniques applied to MLP, RNN, LSTM, CNN, Transformer (Transf.), and State Space (S4) models trained with forking-sequences, expressed as percentage improvements over baseline (no ensembling). Results are aggregated across all evaluated datasets. The ensembling strategies include moving average (Mov. Avg.), moving median functions, and exponential smoothing (ES) with varying smoothing parameters ($\alpha$). Larger $\alpha$ parameter values assigns higher weights to predictions where the target date appears earlier in the forecast horizon.

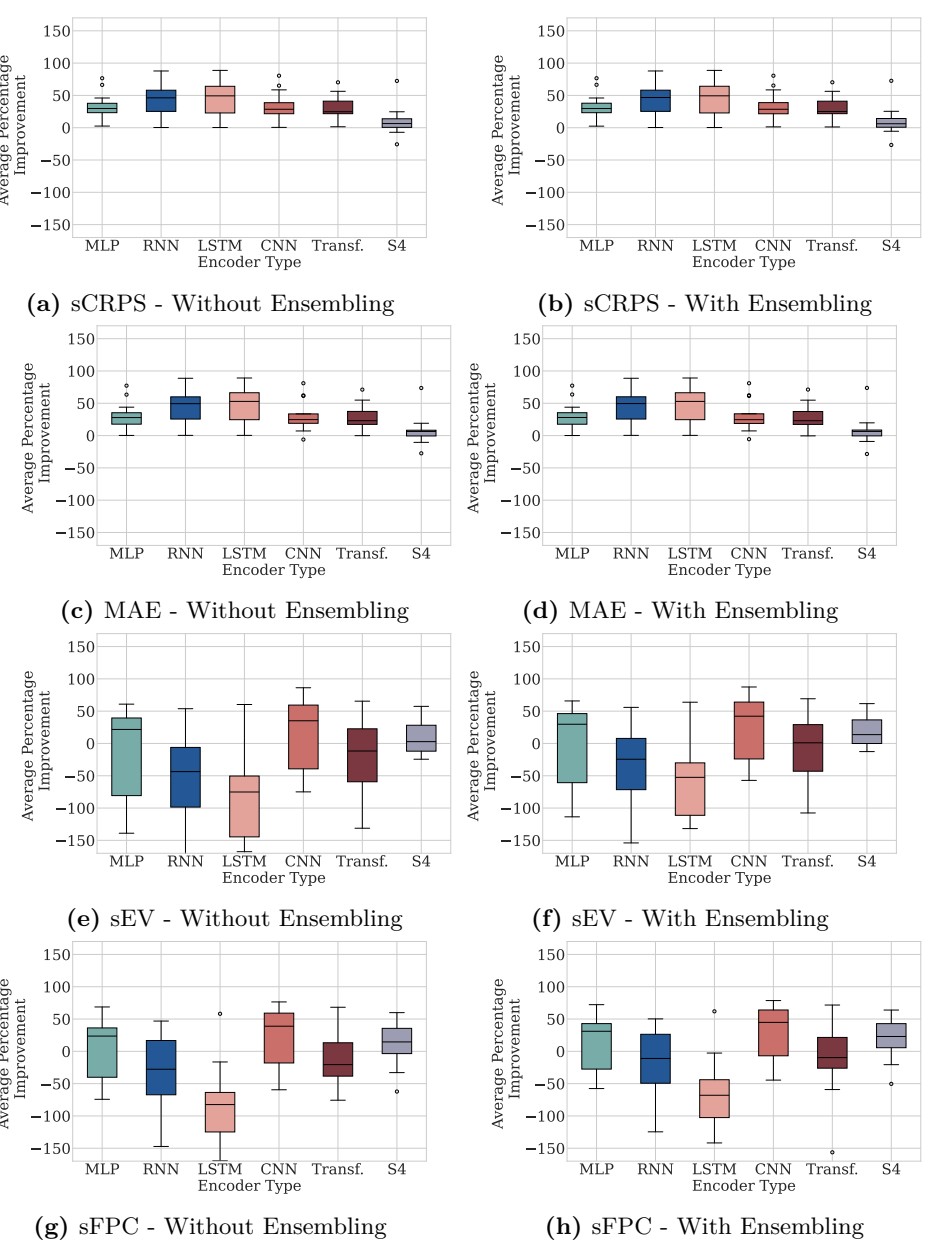

**(a)** sCRPS - Without Ensembling

**(b)** sCRPS - With Ensembling

**(c)** MAE - Without Ensembling

**(d)** MAE - With Ensembling

**(e)** sEV - Without Ensembling

**(f)** sEV - With Ensembling

**(g)** sFPC - Without Ensembling

**(h)** sFPC - With Ensembling

**Figure 16:** Percentage improvement in sCRPS, MAE, sEV, and sFPC metrics for models with forking-sequences compared with the window-sampling scheme both without ensembling during inference, averaged across datasets **(a, c, e, g)**. Percentage improvement in sCRPS, MAE, sEV, and sFPC metrics for models with the *forking-sequences* scheme and exponential smoothing ($\alpha = 0.9$) ensembling during inference compared with the *window-sampling* scheme, averaged across datasets **(b, d, f, h)**. Results greater than zero indicate lower metric values using forking-sequences. Forking-sequences with ensembling during inference demonstrates significant improvements in sCRPS and MAE values over windows-sampling across all encoders. While forecast ensembling during inference reduces forecast volatility compared to no ensembling for models with forking-sequences (see Fig. 15), it does not improve sEV over window-sampling for `RNN` and `LSTM` encoders **(f)**.

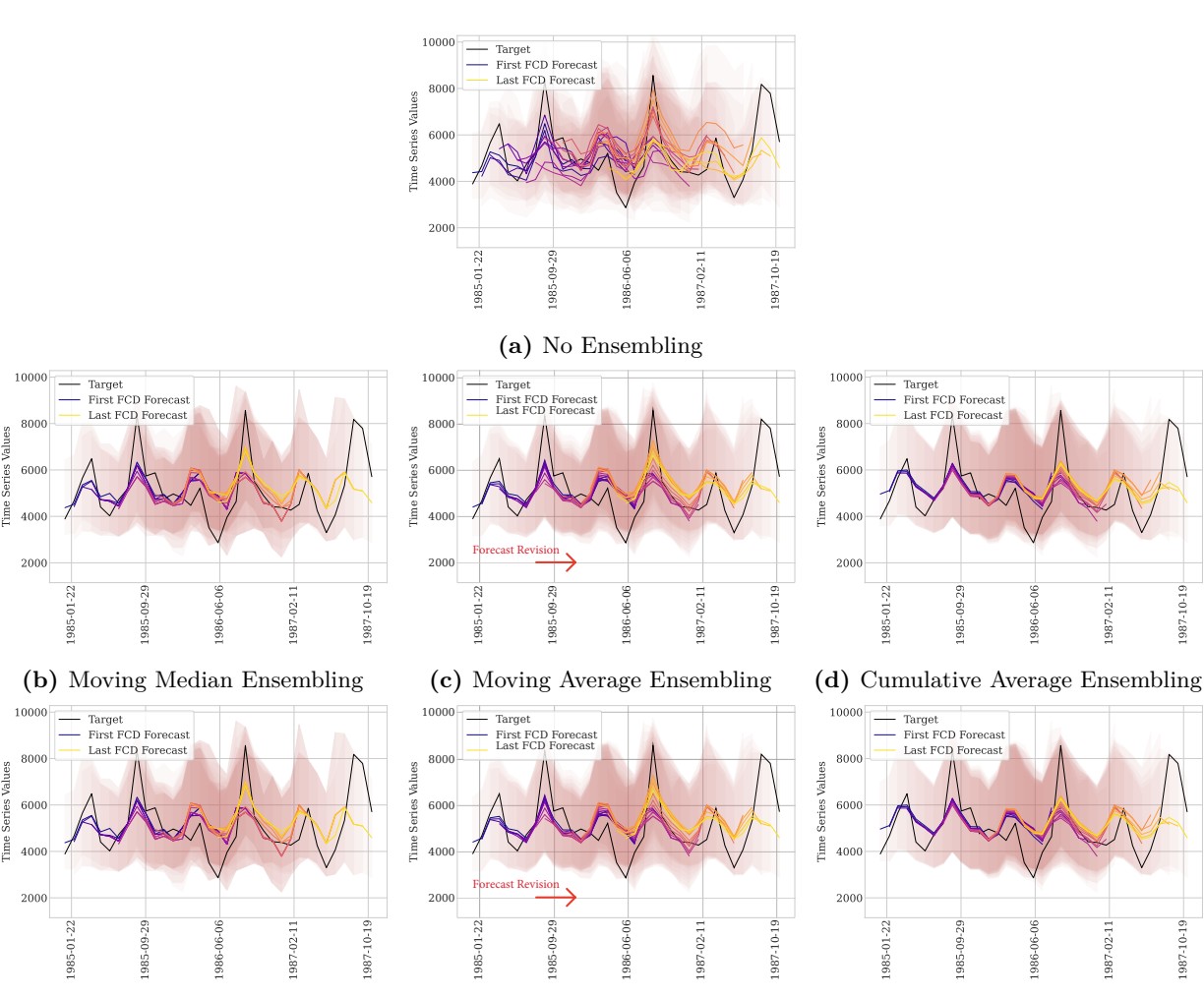

**(a)** No Ensembling

**(b)** Moving Median Ensembling     **(c)** Moving Average Ensembling     **(d)** Cumulative Average Ensembling

**(e)** Exponential Smoothing ($\alpha$=0.1)     **(f)** Exponential Smoothing ($\alpha$=0.5)     **(g)** Exponential Smoothing ($\alpha$=0.9)

**Figure 17:** Forecast outputs for **a)** no ensembling technique and various ensembling techniques applied during inference, including **b)** moving median, **c)** moving average, and **d)** cumulative average, and **e,f,g)** exponential smoothing with various levels of smoothing $\alpha = \{0.1, 0.5, 0.9\}$.

## J Forecast Ensembling Impact on Foundation Forecasting Models

To demonstrate the benefits of forecast ensembling during inference, we reimplement `PatchTST` (Nie et al., 2023) and `NBEATS` (Oreshkin et al., 2020) using the forking-sequences architectural design, while maintaining their original window-sampling for pre-training.

These models are trained on a synthetic dataset and evaluated in the cross-frequency transfer learning task, achieving performance comparable to their original implementations. We evaluate several ensembling functions as shown in Fig 18. Our analysis reveals a trade-off between forecast volatility and accuracy when applying different ensembling functions. Notably, exponential smoothing with $\alpha \in \{0.5, 0.9\}$ emerges as an effective approach, successfully reducing forecast volatility during inference while maintaining model accuracy in terms of sCRPS and MAE. In contrast, functions like median and exponential smoothing with $\alpha = 0.1$, while achieving larger reductions in forecast revisions, do so at the cost of significant degradation in forecast accuracy. Exponential smoothing with a larger $\alpha$ parameter value assigns higher weights to predictions where the target date appears earlier in the forecast horizon. The smaller degradation in accuracy with larger $\alpha$ values aligns with the general expectation that models will make more accurate predictions for near-term versus distant forecast horizons.

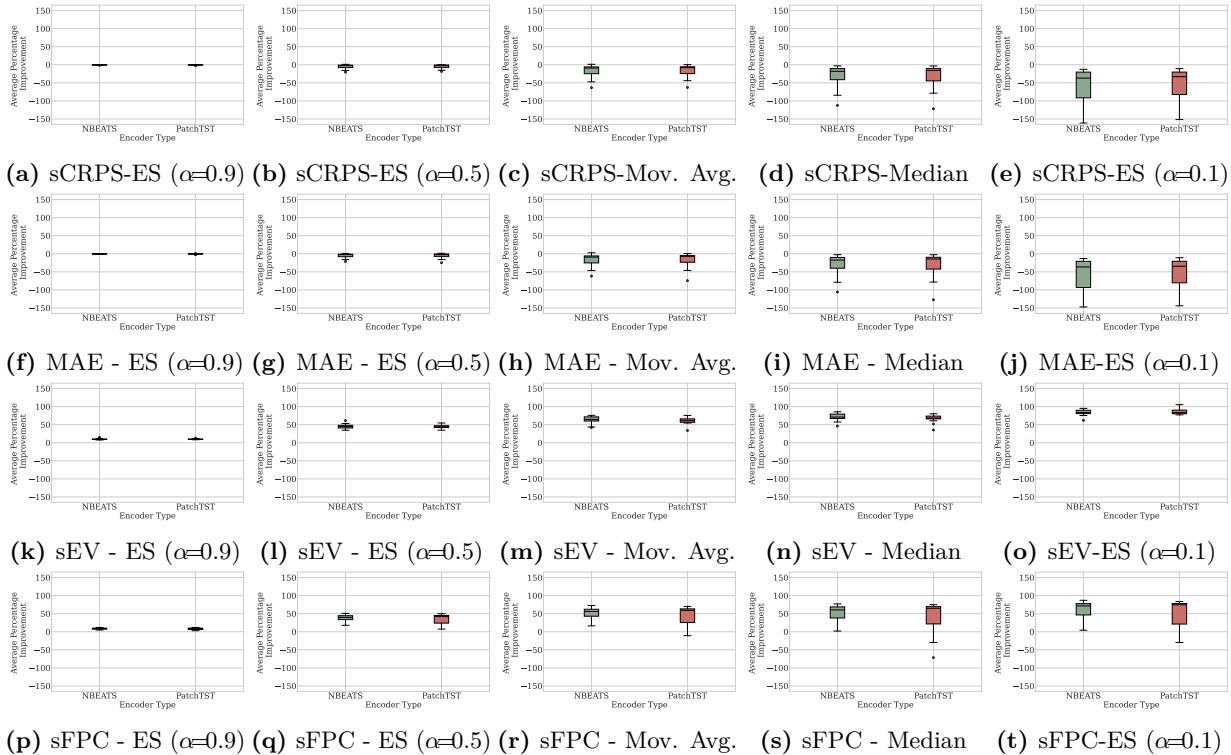

**Figure 18:** Comparative performance gains achieved through diverse ensembling techniques applied to pre-trained `NBEATS` and `PatchTST` models, expressed as percentage improvements over baseline (no ensembling). Results are aggregated across all evaluated datasets. The ensembling strategies include moving average (Mov. Avg.), moving median functions, and exponential smoothing (ES) with varying smoothing parameters ($\alpha$). Larger $\alpha$ parameter values assigns higher weights to predictions where the target date appears earlier in the forecast horizon.

# K  Additional Point Forecasting Results

**Table 19:** Empirical evaluation of probabilistic forecasts. Mean Absolute Error (MAE) averaged over 5 runs. Lower measurements are preferred. The methods without standard deviation have deterministic solutions. For the `MQForecaster` architecture we vary the type of encoder, and the training scheme between *forking-sequences* (FS) and *window-sampling* (WS). We include the forking-sequences ensemble variant (EN).

| | Freq | RNN EN | FS | WS | LSTM EN | FS | WS | CNN EN | FS | WS | Transformer EN | FS | WS | S4 EN | FS | WS | StatsForecast ETS | ARIMA |
|---|---|---|---|---|---|---|---|---|---|---|---|---|---|---|---|---|---|---|
| M1 | M | 7151.2 | 7156.5 | 9926.1 | 7205.8 | 7208.6 | 9689.3 | 1764.2 | 1768.5 | 2617.6 | 1943.5 | 1951.4 | 2370.3 | 2873.7 | 2911.1 | 2634.8 | 1898.7 | 1987.1 |
| | | (884.4) | (882.8) | (1204.7) | (780.9) | (780.0) | (1668.9) | (23.1) | (23.7) | (253.8) | (104.3) | (107.2) | (168.6) | (561.3) | (549.4) | (29.6) | (-) | (-) |
| | Q | 11319.2 | 11323.1 | 15452.0 | 11841.1 | 11839.2 | 16863.0 | 2468.8 | 2471.0 | 2661.3 | 2614.7 | 2611.7 | 2821.4 | 2862.5 | 2906.1 | 2687.2 | 2465.6 | 2745.7 |
| | | (852.4) | (854.3) | (591.6) | (1408.7) | (1411.4) | (1811.4) | (118.7) | (121.4) | (151.2) | (44.4) | (44.1) | (364.2) | (215.8) | (219.4) | (47.3) | (-) | (-) |
| | Y | 734926.8 | 734929.5 | 737291.0 | 734925.4 | 734928.0 | 737230.4 | 95608.8 | 96092.8 | 90540.5 | 99471.0 | 100019.8 | 116972.4 | 98865.1 | 99182.7 | 107798.3 | 100188.5 | 96408.3 |
| | | (689.3) | (689.6) | (1190.1) | (873.5) | (874.5) | (1401.8) | (7503.3) | (7279.3) | (10321.3) | (9534.3) | (9379.2) | (17944.2) | (7587.9) | (6952.0) | (11464.0) | (-) | (-) |
| M3 | O | 393.0 | 392.8 | 1381.4 | 700.7 | 699.6 | 2057.5 | 231.1 | 230.3 | 284.7 | 206.3 | 205.9 | 456.7 | 314.7 | 312.2 | 244.9 | 203.0 | 206.6 |
| | | (230.2) | (231.3) | (691.4) | (860.7) | (861.3) | (722.6) | (48.5) | (47.9) | (57.2) | (3.7) | (3.5) | (223.1) | (27.4) | (27.1) | (1.9) | (-) | (-) |
| | M | 733.4 | 742.9 | 1565.6 | 703.8 | 709.7 | 1601.3 | 595.7 | 597.1 | 859.3 | 631.3 | 633.8 | 772.2 | 643.9 | 649.1 | 802.1 | 686.9 | 691.3 |
| | | (34.4) | (37.8) | (671.3) | (35.4) | (35.6) | (626.8) | (4.7) | (4.8) | (166.2) | (32.0) | (32.3) | (60.3) | (19.6) | (21.2) | (5.2) | (-) | (-) |
| | Q | 565.9 | 567.4 | 934.9 | 582.3 | 583.3 | 1567.4 | 492.6 | 491.8 | 626.7 | 491.2 | 489.5 | 769.9 | 520.0 | 521.4 | 574.4 | 536.1 | 535.3 |
| | | (20.8) | (21.8) | (289.5) | (38.2) | (40.6) | (405.0) | (19.3) | (19.7) | (70.1) | (15.7) | (15.0) | (368.6) | (17.7) | (18.3) | (2.5) | (-) | (-) |
| | Y | 1183.3 | 1184.5 | 2826.1 | 1257.2 | 1256.3 | 2832.9 | 927.9 | 925.4 | 1195.2 | 965.9 | 962.9 | 1264.8 | 984.5 | 980.3 | 1053.2 | 1050.8 | 1066.8 |
| | | (173.3) | (172.2) | (1210.8) | (228.6) | (229.1) | (899.9) | (14.3) | (14.8) | (54.3) | (73.2) | (70.4) | (241.6) | (53.1) | (51.2) | (43.8) | (-) | (-) |
| M4 | H | 2581.4 | 2582.0 | 5233.0 | 2600.9 | 2601.0 | 4803.5 | 294.4 | 294.3 | 1551.1 | 371.5 | 372.8 | 1292.5 | 406.7 | 409.1 | 1550.8 | 635.6 | 290.8 |
| | | (532.5) | (532.6) | (239.8) | (693.1) | (693.2) | (285.7) | (21.1) | (20.9) | (62.5) | (62.0) | (62.8) | (97.7) | (73.6) | (73.4) | (15.5) | (-) | (-) |
| | D | 193.0 | 193.2 | 1711.8 | 196.3 | 196.2 | 1806.3 | 169.5 | 169.1 | 438.5 | 173.5 | 172.9 | 172.7 | 171.3 | 171.2 | 175.1 | 168.4 | 170.0 |
| | | (7.4) | (7.7) | (1426.1) | (18.4) | (18.4) | (1364.8) | (0.9) | (0.9) | (297.9) | (1.4) | (1.2) | (5.1) | (3.3) | (3.2) | (9.9) | (-) | (-) |
| | W | 367.3 | 368.5 | 1232.0 | 393.3 | 394.3 | 1941.8 | 285.5 | 285.9 | 430.4 | 297.5 | 298.1 | 383.3 | 338.4 | 340.3 | 353.7 | 336.6 | 325.4 |
| | | (21.2) | (21.2) | (370.9) | (18.4) | (18.5) | (311.7) | (7.0) | (7.0) | (107.1) | (26.8) | (28.1) | (42.3) | (10.7) | (11.3) | (2.5) | (-) | (-) |
| | M | 620.0 | 623.5 | 1814.3 | 601.1 | 602.8 | 1846.2 | 523.3 | 523.5 | 715.7 | 547.1 | 547.8 | 654.7 | 558.9 | 561.0 | 642.5 | 560.6 | 567.6 |
| | | (32.6) | (34.1) | (862.8) | (27.1) | (27.3) | (821.5) | (2.7) | (2.7) | (173.9) | (16.6) | (16.8) | (57.8) | (11.2) | (11.9) | (7.3) | (-) | (-) |
| | Q | 624.4 | 627.3 | 1210.7 | 667.2 | 669.4 | 2505.5 | 565.7 | 565.3 | 695.4 | 573.9 | 572.8 | 862.9 | 594.8 | 597.2 | 634.6 | 568.4 | 597.1 |
| | | (14.3) | (14.9) | (265.7) | (52.5) | (54.8) | (864.9) | (16.3) | (16.8) | (79.3) | (19.4) | (18.5) | (446.9) | (16.2) | (16.9) | (2.3) | (-) | (-) |
| | Y | 1307.9 | 1308.7 | 3105.4 | 1493.3 | 1493.5 | 2997.4 | 775.2 | 774.4 | 967.5 | 807.7 | 806.0 | 1064.0 | 837.8 | 833.4 | 901.1 | 849.0 | 889.3 |
| | | (317.2) | (317.2) | (999.9) | (510.3) | (511.3) | (887.8) | (10.9) | (11.4) | (32.0) | (38.2) | (36.0) | (190.2) | (30.0) | (29.0) | (22.4) | (-) | (-) |
| Tourism | M | 12639.1 | 12647.1 | 16232.7 | 12487.8 | 12492.2 | 15925.6 | 2215.0 | 2225.7 | 5883.0 | 3132.1 | 3143.6 | 5269.2 | 5469.6 | 5514.1 | 5922.6 | 2624.5 | 3125.8 |
| | | (1314.2) | (1314.7) | (1205.7) | (1234.5) | (1234.7) | (1503.2) | (151.4) | (153.9) | (401.7) | (752.6) | (760.2) | (430.7) | (1054.1) | (1034.8) | (15.6) | (-) | (-) |
| | Q | 74188.2 | 74238.3 | 84967.3 | 75109.5 | 75149.3 | 86636.3 | 11600.5 | 11642.5 | 17520.6 | 11164.2 | 11137.1 | 23072.6 | 17985.8 | 18446.6 | 17921.3 | 11375.6 | 12567.5 |
| | | (1972.5) | (1966.1) | (774.4) | (4157.5) | (4150.4) | (1703.6) | (1019.4) | (1102.9) | (886.8) | (510.9) | (520.2) | (10670.8) | (948.6) | (956.5) | (271.0) | (-) | (-) |
| | Y | 372645.8 | 372644.4 | 376107.2 | 372760.6 | 372762.5 | 375857.6 | 76585.0 | 76514.3 | 88681.1 | 81122.7 | 80648.5 | 98405.9 | 86618.8 | 85851.7 | 86030.8 | 74574.0 | 86727.3 |
| | | (969.2) | (970.1) | (1518.4) | (1300.6) | (1302.6) | (1753.5) | (1209.5) | (1269.9) | (1184.2) | (1445.6) | (1368.7) | (14645.7) | (1912.7) | (1919.1) | (1696.4) | (-) | (-) |

To complement the probabilistic results in Section 4.2, we evaluate median forecasts denoted by $\hat{\mathbf{y}}_{[b][t][h]}$ through the *mean absolute error* (MAE), as described by

$$\text{MAE}\left(\mathbf{y}_{[b][t][h]},\ \hat{\mathbf{y}}_{[b][t][h]}\right) = \frac{1}{B \times T \times H} \sum_{b,t,h} |y_{b,t,h} - \hat{y}_{b,t,h}|. \tag{53}$$

The forking-sequences training scheme improved median MAE across datasets by 27.9%, 49.4%, 52.9%, 24.7%, 23.0%, and 6.4% for `MLP`, `RNN`, `LSTM`, `CNN`, `Transformer`, and `S4` encoders, respectively. These changes can be attributed to improved training convergence, induced by better gradient flow with the forking-sequences training scheme. The results are generally consistent with sCRPS outcomes in Table 2.

**Table 20:** Empirical evaluation of probabilistic forecasts. Mean *Symmetric Forecast Percentage Change* (sFPC) averaged over 5 runs. Lower measurements are preferred. The methods without standard deviation have deterministic solutions. For the `MQForecaster` architecture we vary the type of encoder, and the training scheme between *forking-sequences* (FS) and *window-sampling* (WS). We include the forking-sequences ensemble variant (EN).

| | Freq | EN | RNN FS | WS | EN | LSTM FS | WS | EN | CNN FS | WS | EN | Transformer FS | WS | EN | S4 FS | WS | StatsForecast ETS | ARIMA |
|---|---|---|---|---|---|---|---|---|---|---|---|---|---|---|---|---|---|---|
| M1 | M | 3.4924 | 3.9411 | 4.538 | 2.8962 | 3.2843 | 2.8205 | 1.9414 | 2.1806 | 4.4672 | 3.4843 | 3.8668 | 2.7681 | 5.2026 | 5.6956 | 6.7165 | 2.0154 | 2.5845 |
| | | (0.5102) | (0.5173) | (3.136) | (0.4261) | (0.4558) | (2.6973) | (0.0964) | (0.1116) | (0.3984) | (1.0391) | (1.1464) | (2.1328) | (1.4705) | (1.4706) | (1.4489) | (-) | (-) |
| | Q | 6.421 | 7.2619 | 5.1985 | 3.2528 | 3.6152 | 1.7921 | 2.9598 | 3.2764 | 3.7791 | 4.1717 | 4.6037 | 4.8285 | 6.9782 | 7.6894 | 5.7839 | 3.6355 | 3.8982 |
| | | (5.2246) | (6.0403) | (1.8765) | (0.5765) | (0.6424) | (1.1461) | (0.619) | (0.6916) | (0.9215) | (0.8179) | (0.8722) | (2.2595) | (1.5255) | (1.5869) | (1.5694) | (-) | (-) |
| | Y | 7.4046 | 8.1509 | 3.2968 | 7.573 | 8.4845 | 3.4459 | 5.0958 | 5.6231 | 3.5255 | 7.4488 | 8.0148 | 6.7425 | 7.4841 | 8.1555 | 9.2269 | 2.8426 | 3.4499 |
| | | (3.5761) | (3.9804) | (1.6987) | (4.1538) | (4.9628) | (3.0855) | (0.7678) | (0.7927) | (0.1156) | (2.5393) | (2.6697) | (1.8123) | (1.5233) | (1.7793) | (1.1255) | (-) | (-) |
| M3 | O | 0.9031 | 0.9995 | 1.6348 | 0.6483 | 0.6996 | 0.3693 | 1.0737 | 1.1783 | 0.9602 | 0.9726 | 1.0734 | 0.6111 | 1.4273 | 1.5399 | 0.9491 | 0.8979 | 0.8581 |
| | | (0.055) | (0.0752) | (1.3581) | (0.2506) | (0.2742) | (0.1861) | (0.131) | (0.1223) | (0.0281) | (0.1196) | (0.1454) | (0.3303) | (0.2212) | (0.2503) | (0.0151) | (-) | (-) |
| | M | 3.8699 | 4.4908 | 3.2095 | 3.2437 | 3.7658 | 2.2956 | 1.8599 | 2.091 | 5.1675 | 2.3857 | 2.6766 | 2.1897 | 2.8336 | 3.22 | 6.2859 | 1.3749 | 1.7204 |
| | | (0.5919) | (0.6782) | (2.6824) | (0.1635) | (0.1948) | (1.8574) | (0.0645) | (0.0775) | (0.6651) | (0.193) | (0.2194) | (2.1382) | (0.3866) | (0.444) | (0.0731) | (-) | (-) |
| | Q | 2.4254 | 2.732 | 4.1033 | 2.2332 | 2.5034 | 1.5421 | 1.9246 | 2.1346 | 3.1639 | 1.7045 | 1.8696 | 3.0422 | 2.2908 | 2.5586 | 3.7077 | 1.8586 | 2.1937 |
| | | (0.1911) | (0.2185) | (0.6063) | (0.3236) | (0.3837) | (1.412) | (0.1389) | (0.169) | (0.4793) | (0.1674) | (0.1958) | (1.3185) | (0.1804) | (0.2093) | (0.0525) | (-) | (-) |
| | Y | 4.1709 | 4.6288 | 2.3418 | 3.8628 | 4.2636 | 1.9358 | 4.4447 | 4.9079 | 4.2122 | 4.4675 | 4.8987 | 3.5971 | 4.2741 | 4.6956 | 4.5213 | 3.4889 | 5.0013 |
| | | (0.6868) | (0.7607) | (0.764) | (0.934) | (1.0314) | (0.6738) | (0.1472) | (0.159) | (0.2383) | (0.279) | (0.3258) | (1.2377) | (0.0792) | (0.096) | (0.0726) | (-) | (-) |
| M4 | H | 2.8868 | 3.0818 | 5.7989 | 1.5155 | 1.6658 | 3.9801 | 1.739 | 1.9606 | 5.0825 | 2.6945 | 2.9935 | 1.0511 | 1.8989 | 2.1092 | 5.2594 | 2.4881 | 0.7861 |
| | | (0.7011) | (0.7081) | (1.6801) | (0.3828) | (0.4082) | (1.962) | (0.2744) | (0.3176) | (0.601) | (1.5328) | (1.7179) | (0.9523) | (0.0513) | (0.0577) | (0.3391) | (-) | (-) |
| | D | 0.4537 | 0.5107 | 0.3125 | 0.4416 | 0.4918 | 0.1826 | 0.4159 | 0.4565 | 1.9417 | 0.3598 | 0.3879 | 0.4764 | 0.4314 | 0.4754 | 0.4771 | 0.497 | 0.4992 |
| | | (0.0553) | (0.0649) | (0.1397) | (0.0514) | (0.0647) | (0.087) | (0.0187) | (0.023) | (2.8271) | (0.0386) | (0.048) | (0.0004) | (0.0007) | (0.0007) | (0.001) | (-) | (-) |
| | W | 1.5329 | 1.7458 | 2.359 | 1.3984 | 1.594 | 0.9472 | 1.1896 | 1.3295 | 2.5746 | 1.3185 | 1.4691 | 1.6595 | 1.8769 | 2.1104 | 2.4523 | 1.4943 | 1.1399 |
| | | (0.1394) | (0.1738) | (0.4512) | (0.0658) | (0.0761) | (0.3152) | (0.0875) | (0.0997) | (0.5053) | (0.4269) | (0.491) | (0.9969) | (0.2082) | (0.238) | (0.0028) | (-) | (-) |
| | M | 2.3649 | 2.7051 | 1.8339 | 1.9748 | 2.2485 | 1.3231 | 1.6072 | 1.7764 | 3.2445 | 1.8045 | 1.9916 | 1.4646 | 2.1085 | 2.3616 | 3.8059 | 1.8584 | 2.2324 |
| | | (0.2915) | (0.3302) | (1.5209) | (0.0736) | (0.0918) | (0.9638) | (0.0286) | (0.0332) | (0.3738) | (0.0945) | (0.1058) | (1.2334) | (0.1798) | (0.2053) | (0.0492) | (-) | (-) |
| | Q | 2.5847 | 2.9166 | 3.3789 | 2.4478 | 2.7578 | 1.1567 | 2.0464 | 2.2708 | 3.1419 | 1.96 | 2.1564 | 3.0392 | 2.539 | 2.8373 | 3.6894 | 2.0469 | 2.2461 |
| | | (0.1264) | (0.1541) | (0.3135) | (0.2539) | (0.2713) | (0.9771) | (0.1392) | (0.1636) | (0.4541) | (0.1817) | (0.2122) | (1.2977) | (0.1688) | (0.1917) | (0.0538) | (-) | (-) |
| | Y | 3.4186 | 3.7695 | 2.1096 | 3.0068 | 3.2865 | 1.8734 | 3.803 | 4.1943 | 3.4233 | 3.7699 | 4.1268 | 2.9492 | 3.5015 | 3.8325 | 3.7027 | 3.082 | 4.5323 |
| | | (0.3673) | (0.4114) | (0.4923) | (0.3657) | (0.4118) | (0.6477) | (0.0418) | (0.0436) | (0.2137) | (0.301) | (0.3249) | (0.9133) | (0.0807) | (0.0952) | (0.0534) | (-) | (-) |
| Tourism | M | 7.3248 | 8.3547 | 7.2186 | 5.8403 | 6.7001 | 4.9197 | 3.3065 | 3.8021 | 12.8501 | 4.6389 | 5.2159 | 5.2677 | 7.2549 | 8.2547 | 15.4095 | 2.1456 | 1.7748 |
| | | (0.8437) | (0.8877) | (5.7788) | (0.2804) | (0.3252) | (4.4183) | (0.3017) | (0.3691) | (1.251) | (1.1698) | (1.2896) | (4.9948) | (1.021) | (1.1437) | (0.6283) | (-) | (-) |
| | Q | 7.8873 | 9.1285 | 10.2508 | 6.9251 | 8.0001 | 2.9944 | 4.5001 | 5.1328 | 12.5189 | 3.7441 | 4.1992 | 13.2162 | 9.464 | 10.6922 | 16.3992 | 3.1549 | 2.7744 |
| | | (0.7101) | (0.7232) | (1.6287) | (0.7941) | (0.9042) | (2.1642) | (1.8249) | (2.2049) | (2.981) | (0.2548) | (0.3116) | (6.2977) | (1.5825) | (1.7716) | (0.0464) | (-) | (-) |
| | Y | 4.3769 | 4.8294 | 2.5744 | 4.1175 | 4.5017 | 2.3038 | 7.2886 | 8.0661 | 5.6373 | 7.8023 | 8.4955 | 6.1573 | 7.1197 | 7.8125 | 7.5449 | 9.7853 | 12.016 |
| | | (1.2373) | (1.4316) | (0.9831) | (1.0605) | (1.1246) | (0.9455) | (0.2171) | (0.2155) | (0.1494) | (1.2193) | (1.3445) | (1.5354) | (0.4173) | (0.3978) | (0.6747) | (-) | (-) |

To complement the probabilistic forecast revision results, we evaluate median forecasts with the *Symmetric Forecast Percentage Change* (sFPC), that we define in Equation 54. sFPC measures the relative change in predicted quantiles across consecutive forecast creation dates, providing a quantitative view of temporal volatility or forecast revision rates. Inspired by the sMAPE metric, sFPC uses a symmetric denominator, based on both current and previous forecasts, to mitigate issues of numerical instability (Hyndman and Koehler, 2006). This design ensures robustness when dealing with small predicted values and avoids the division-by-zero problems common in traditional percentage-based metrics.

$$\text{sFPC}_q\left(\hat{\mathbf{Y}}^{(q)}_{[b][t][h]}\right) = \frac{200}{B \times T \times H} \sum_{b,t,h} \frac{|\hat{Y}^{(q)}_{b,t+1,h} - \hat{Y}^{(q)}_{b,t,h+1}|}{|\hat{Y}^{(q)}_{b,t+1,h}| + |\hat{Y}^{(q)}_{b,t,h+1}|}. \tag{54}$$

Because uncertainty decreases across FCDs it is expected that forecasted quantiles above P50 decrease while below P50 increase, to avoid complications, we measure the revisions for the median (q=.50).

Table 3 presents sFPC values, averaged over five runs, for all dataset–frequency combinations and model architectures, including statistical baselines. For `LSTM` encoder models, the forking-sequences scheme reduces forecast revisions by %, on average across datasets, compared to window-sampling. Consistent gains are observed for `MLP` (28.8%), `RNN` (28.8%), and `CNN` (31.3%) encoders, while the `Transformer`-based encoder has relatively less improvement (8.8%). Table 2 is summarized in Fig. 16h which shows the percentage change of the sFPC metric for models with the *forking-sequences* training scheme over that of models with the *window-sampling* scheme, averaged across datasets. All encoder variants with forking-sequences have improved sFPC.

# L   Additional Quantile Coverage Results

## L.1   Forking-sequences Effects on Average Coverage Error

**Table 21:** Empirical evaluation of probabilistic forecasts. Average Coverage Error (ACE) averaged over 5 runs. Lower measurements are preferred. The methods without standard deviation have deterministic solutions. For the `MQForecaster` architecture we vary the type of encoder, and the training scheme between *forking-sequences* (FS) and *window-sampling* (WS). We include the forking-sequences ensemble variant (EN).

| | Freq | EN | RNN FS | WS | EN | LSTM FS | WS | EN | CNN FS | WS | EN | Transformer FS | WS | EN | S4 FS | WS | StatsForecast ETS | ARIMA |
|---|---|---|---|---|---|---|---|---|---|---|---|---|---|---|---|---|---|---|
| M1 | M | 0.1192 | 0.1192 | 0.1882 | 0.1223 | 0.1222 | lstmws | 0.12 | 0.1207 | 0.1682 | 0.0538 | 0.0534 | 0.1481 | 0.0383 | 0.0374 | 0.1604 | 0.0503 | 0.0454 |
| | | (0.071) | (0.071) | (0.0519) | (0.0596) | (0.0599) | (0.0287) | (0.0186) | (0.0186) | (0.0883) | (0.0184) | (0.0189) | (0.0547) | (0.0136) | (0.0138) | (0.0518) | (-) | (-) |
| | Q | 0.1159 | 0.1136 | 0.2589 | 0.1008 | 0.1005 | lstmws | 0.1548 | 0.1554 | 0.2086 | 0.109 | 0.1078 | 0.0827 | 0.0656 | 0.0645 | 0.1461 | 0.0944 | 0.0964 |
| | | (0.0445) | (0.0446) | (0.0687) | (0.0536) | (0.053) | (0.057) | (0.0842) | (0.0838) | (0.0497) | (0.0737) | (0.0738) | (0.0352) | (0.0276) | (0.0287) | (0.0771) | (-) | (-) |
| | Y | 0.1564 | 0.1571 | 0.1714 | 0.1533 | 0.1544 | lstmws | 0.064 | 0.0668 | 0.1014 | 0.1209 | 0.1196 | 0.1094 | 0.0842 | 0.0826 | 0.0829 | 0.2595 | 0.2031 |
| | | (0.0437) | (0.0449) | (0.0088) | (0.0217) | (0.0213) | (0.0095) | (0.0212) | (0.0212) | (0.0324) | (0.0253) | (0.0244) | (0.0423) | (0.0513) | (0.0511) | (0.0287) | (-) | (-) |
| M3 | O | 0.1203 | 0.1199 | 0.1581 | 0.1242 | 0.124 | lstmws | 0.047 | 0.0466 | 0.1257 | 0.0196 | 0.0215 | 0.1402 | 0.1609 | 0.1572 | 0.055 | 0.0202 | 0.0175 |
| | | (0.0701) | (0.0706) | (0.1175) | (0.0592) | (0.0591) | (0.0091) | (0.0315) | (0.0309) | (0.0911) | (0.0075) | (0.0067) | (0.0689) | (0.0435) | (0.0443) | (0.013) | (-) | (-) |
| | M | 0.0521 | 0.0533 | 0.1527 | 0.0512 | 0.0517 | lstmws | 0.0197 | 0.0192 | 0.0866 | 0.0199 | 0.0192 | 0.0451 | 0.0285 | 0.03 | 0.0466 | 0.0392 | 0.0229 |
| | | (0.0055) | (0.0054) | (0.0678) | (0.0056) | (0.0056) | (0.0661) | (0.0044) | (0.0047) | (0.0676) | (0.0067) | (0.0067) | (0.0067) | (0.0111) | (0.0111) | (0.0162) | (-) | (-) |
| | Q | 0.0619 | 0.0612 | 0.2075 | 0.042 | 0.0418 | lstmws | 0.0427 | 0.0424 | 0.0811 | 0.0549 | 0.0541 | 0.0797 | 0.0629 | 0.0617 | 0.0439 | 0.0511 | 0.0575 |
| | | (0.0186) | (0.0183) | (0.1107) | (0.0258) | (0.0255) | (0.0605) | (0.0175) | (0.0174) | (0.0298) | (0.0187) | (0.0186) | (0.0304) | (0.0197) | (0.0197) | (0.0086) | (-) | (-) |
| | Y | 0.0465 | 0.0464 | 0.2575 | 0.0418 | 0.0416 | lstmws | 0.0201 | 0.0216 | 0.1262 | 0.0397 | 0.0391 | 0.0626 | 0.0626 | 0.0621 | 0.0554 | 0.0964 | 0.087 |
| | | (0.0334) | (0.0336) | (0.1167) | (0.0284) | (0.0284) | (0.1059) | (0.0129) | (0.0139) | (0.0264) | (0.0332) | (0.0323) | (0.0271) | (0.0286) | (0.0286) | (0.0198) | (-) | (-) |
| M4 | H | 0.0435 | 0.0435 | 0.2161 | 0.0383 | 0.0383 | lstmws | 0.0468 | 0.047 | 0.0868 | 0.1025 | 0.1021 | 0.0623 | 0.0452 | 0.0447 | 0.13 | 0.0128 | 0.0326 |
| | | (0.0092) | (0.0093) | (0.1366) | (0.0181) | (0.0181) | (0.0503) | (0.0212) | (0.021) | (0.0436) | (0.0501) | (0.05) | (0.0212) | (0.0203) | (0.0205) | (0.0458) | (-) | (-) |
| | D | 0.0463 | 0.0464 | 0.1891 | 0.0412 | 0.0414 | lstmws | 0.0291 | 0.0294 | 0.256 | 0.0379 | 0.0381 | 0.0502 | 0.0351 | 0.0351 | 0.043 | 0.0473 | 0.0437 |
| | | (0.0293) | (0.0295) | (0.1101) | (0.0218) | (0.0216) | (0.1124) | (0.0048) | (0.0048) | (0.136) | (0.0117) | (0.0119) | (0.0284) | (0.0205) | (0.0209) | (0.0216) | (-) | (-) |
| | W | 0.0312 | 0.0312 | 0.2709 | 0.0333 | 0.0332 | lstmws | 0.0276 | 0.0281 | 0.1136 | 0.0236 | 0.024 | 0.05 | 0.0371 | 0.038 | 0.043 | 0.0493 | 0.0404 |
| | | (0.0095) | (0.0094) | (0.1059) | (0.0045) | (0.0044) | (0.0545) | (0.0082) | (0.0077) | (0.0593) | (0.0091) | (0.0093) | (0.0197) | (0.016) | (0.0156) | (0.0134) | (-) | (-) |
| | M | 0.0412 | 0.0411 | 0.1265 | 0.0325 | 0.0325 | lstmws | 0.0083 | 0.0092 | 0.0987 | 0.0183 | 0.018 | 0.0337 | 0.0363 | 0.0374 | 0.0446 | 0.023 | 0.0142 |
| | | (0.0119) | (0.0117) | (0.1047) | (0.0076) | (0.0076) | (0.0709) | (0.0023) | (0.0023) | (0.0816) | (0.0038) | (0.0038) | (0.0139) | (0.0061) | (0.006) | (0.0208) | (-) | (-) |
| | Q | 0.0232 | 0.022 | 0.2433 | 0.0282 | 0.0279 | lstmws | 0.0328 | 0.0323 | 0.0807 | 0.0331 | 0.0323 | 0.0749 | 0.0351 | 0.0357 | 0.0358 | 0.0479 | 0.0368 |
| | | (0.0031) | (0.0031) | (0.108) | (0.0211) | (0.0218) | (0.1155) | (0.0112) | (0.0122) | (0.0293) | (0.0077) | (0.0076) | (0.0385) | (0.007) | (0.0078) | (0.0061) | (-) | (-) |
| | Y | 0.0392 | 0.0392 | 0.1886 | 0.0605 | 0.0607 | lstmws | 0.0348 | 0.0379 | 0.1296 | 0.0478 | 0.0481 | 0.0586 | 0.0666 | 0.0667 | 0.0813 | 0.1071 | 0.092 |
| | | (0.0223) | (0.0224) | (0.0825) | (0.0338) | (0.0336) | (0.0716) | (0.0245) | (0.0247) | (0.0224) | (0.0394) | (0.0389) | (0.0373) | (0.033) | (0.0336) | (0.0239) | (-) | (-) |
| Tourism | M | 0.1607 | 0.1615 | 0.1507 | 0.1641 | 0.1645 | lstmws | 0.031 | 0.0303 | 0.1477 | 0.017 | 0.0162 | 0.1433 | 0.0423 | 0.0441 | 0.1483 | 0.0743 | 0.054 |
| | | (0.0101) | (0.0102) | (0.0322) | (0.0107) | (0.0107) | (0.0851) | (0.0022) | (0.0023) | (0.009) | (0.0042) | (0.0036) | (0.0064) | (0.0126) | (0.0132) | (0.0062) | (-) | (-) |
| | Q | 0.1485 | 0.1494 | 0.2646 | 0.16 | 0.1611 | lstmws | 0.0674 | 0.0661 | 0.1209 | 0.0497 | 0.0487 | 0.1574 | 0.0522 | 0.0535 | 0.1241 | 0.0931 | 0.0817 |
| | | (0.0051) | (0.0053) | (0.0592) | (0.0129) | (0.0134) | (0.0938) | (0.0342) | (0.035) | (0.0242) | (0.0338) | (0.0342) | (0.0795) | (0.0267) | (0.0278) | (0.0026) | (-) | (-) |
| | Y | 0.2423 | 0.2427 | 0.2679 | 0.2428 | 0.243 | lstmws | 0.0478 | 0.0496 | 0.0893 | 0.066 | 0.0668 | 0.1136 | 0.0742 | 0.0745 | 0.0901 | 0.0875 | 0.0889 |
| | | (0.0106) | (0.0106) | (0.0213) | (0.0177) | (0.0176) | (0.0123) | (0.0122) | (0.0129) | (0.0335) | (0.0231) | (0.0227) | (0.0438) | (0.0189) | (0.0185) | (0.0163) | (-) | (-) |

We assess the calibration of all forecast quantiles using the *Average Coverage Error* (ACE), defined as:

$$\text{ACE}\left(\mathbf{y}_{[b][t][h]},\ \hat{\mathbf{Y}}_{[b][t][h]}\right) = \frac{1}{Q}\sum_q \left| \frac{1}{B \times T \times H}\sum_{b,t,h}\mathbb{1}_{\{y_{b,t,h} \leq \hat{Y}_{b,t,h}^{(q)}\}} - q \right|. \tag{55}$$

The forking-sequences training scheme improves ACE over window-sampling with median percentage improvements across datasets of 35.9%, 66.3%, 61.3%, 61.5%, 43.9%, and 17.6% for `MLP`, `RNN`, `LSTM`, `CNN`, `Transformer`, and `S4` encoders, respectively. These results highlight that forking-sequences can substantially improve model calibration across prediction quantiles over window-sampling.

We note that the additional ACE improvement from ensembling during inference over forking-sequences without ensembling is relatively marginal, with median percentage improvements across datasets of 0.3%, 0.0%, 0.0%, 0.5%, -1.1%, 0.1% for `MLP`, `RNN`, `LSTM`, `CNN`, `Transformer`, and `S4` encoders, respectively, where only slight degradation is observed for the `Transformer` encoder. Thus, forking-sequences with ensembling during inference does not substantially degrade, but rather can further improve, ACE over window-sampling, with median percentage improvements across datasets of 36.3%, 66.7%, 61.5%, 61.0%, 43.8%, and 18.2% for `MLP`, `RNN`, `LSTM`, `CNN`, `Transformer`, and `S4` encoders, respectively. These improvements over window-sampling differ from those of forking-sequences without ensembling by only 0.5% and 0.1% for `CNN` and `Transformer`, indicating that ensembling introduces negligible change across all encoders.

## L.2 Ensemble Effects on Quantile Coverage

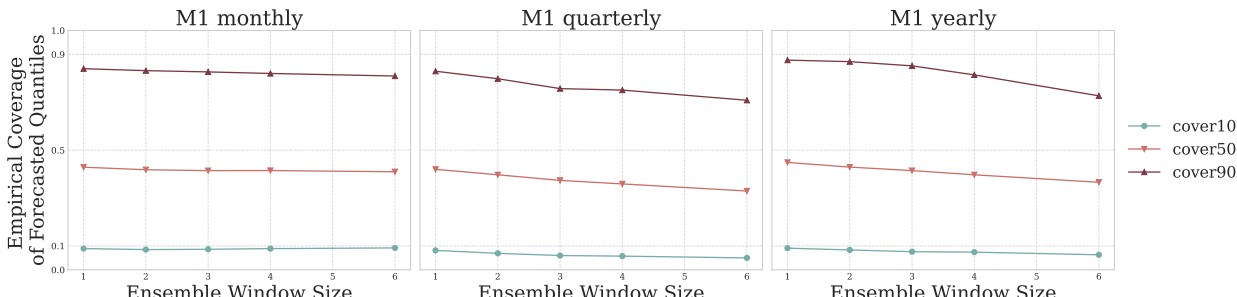

**Figure 19:** Empirical coverage for `NBEATS` model with forking-sequence moving-average ensembling during inference. (a) On the M1 monthly dataset, ensembling slightly does not change coverage. On the M3 quarterly and yearly datasets, ensembling shows a tendency to reduce coverage of the forecasted quantiles. Overall, the influence of the ensemble window on quantile coverage is not consistent across datasets or quantiles. However there is a tendency of the ensemble to narrow down forecast intervals when increasing the windows included in the ensemble.

In this experiment, we asses the effect of forking-sequences ensemble on the empirical coverage of the forecasted quantiles:

$$\text{Coverage}\left(\mathbf{y}_{[b][t][h]},\ \hat{\mathbf{Y}}^{(q)}_{[b][t][h]}\right) = \frac{1}{B \times T \times H} \sum_{b,t,h} \mathbb{1}_{\{y_{b,t,h} \leq \hat{Y}^{(q)}_{b,t,h}\}}. \tag{56}$$

We apply the ensemble technique to a pre-trained `NBEATS` model using the benchmark M1 and M3 datasets. The `NBEATS` model is employed in a zero-shot capacity, with the only variation being the length of the ensemble considered for the forking sequences. In our controlled experiment, we vary the windows used in the forking-sequences ensemble and measure their impact on the coverage of the estimated quantiles.

For the M1 dataset (Figure 19), we observe a general under-coverage across quantiles. P50 and P90 start at roughly 10% under-coverage with a window size of 1 and gradually move toward about 14% under-coverage as the window increases. P10, in contrast, shows a minimal changes. While these trends suggest that the ensemble window can influence calibration, the direction and magnitude of the impact differ across quantiles. Results on the M3 quarterly dataset (Figure 9) display a different pattern. The model begins with slight over-coverage; approximately 6% for P50, 2% for P10; increasing the ensemble window reduces this over-coverage and eventually shifts P10 and P50 toward slight under-coverage, while the effect on P90 remains ambiguous.

The M1 and M3 reveal no consistent calibration effect attributable to the ensemble window. The influence varies by dataset and quantile, and in several cases the shifts are small or in opposing directions. These mixed results indicate that the coverage impact of forking-sequence ensembling is inconclusive and point to the need for a deeper understanding of how ensembling interacts with quantile estimation.

Building on these observations, future research can explore more sophisticated methods for aggregating quantile forecasts, moving beyond simple averaging techniques. Specifically, inspiration can be drawn from the Quantile Regression Averaging (QRA) method (Liu et al., 2017). QRA uses quantile linear regression on the output of individual base models to generate more robust probabilistic forecasts, a method that has proven effective in various forecasting competitions. The varying impact of ensembling on different quantiles observed here suggests a potential benefit in employing the more flexible aggregation strategies described in the paper by Fakoor et al. (2023). Their approach considers weighted ensembles where weights can vary not only across individual models but also across quantile levels, which could help address the inconsistent coverage effects noted in our experiments.

