# OpenReview forum: "Forking-Sequences: Statistically and Computationally Efficient Multi-Horizon Forecasting with Reduced Volatility"
_TMLR — Accepted by TMLR_

### Review · Reviewer_3s2u · 2026-02-17

**Summary Of Contributions:**

The authors compare window-based training (reduced, independent context) with forking sequence training (full context interlinking predictions).
The main contributions of the authors are:
* They present theorems ensuring forking sequence training guarantees linear variance reduction with processed time steps while having lower computational complexity.
* They propose to use forking sequences to generate ensembles by averaging predictions made at different time steps and step ahead for the same time point
* They propose a new volatility metric to for assessing forecast stability

However, considering that:
* Window-based sampling (the standard in forecast literature) has also a linear variance reduction as a function of batch size, as stated by the authors
* Window-based sampling can produce identical model averaging as forking, as stated several times by the authors
* The computational complexity reduction is a trivial fact (window-based training increases it by factor L)
* Forecast stability is desirable but shouldn’t be optimised for, as forecasts naturally degrade as a function of step ahead, as stated by the authors

the main theoretical contribution of the article seems to be the theorem showing that averaging decreases error variance, which seems trivial. The interesting point of incorporating forecast ensembling in training is deferred to future work.
The main empirical contribution comes from the experiment, but I have some concerns, expressed in the following comments.

**Audience:**

Yes

**Audience Explanation:**

The topic is of interest to the forecasting community, and could provide evidence that forking sequences are a better training paradigm of current standard (window sampling).  In this view, it's pivotal for empirical results to be convincing.

**Claims And Evidence:**

No

**Claims Explanation:**

Sensational results. “forking-sequences training scheme reduced sCRPS by 49.3%, on average across datasets, compared to the window-sampling scheme confirming earlier observations by...” The three cited works show marginal improvements, nowhere near 50% or close to the improvements reported in the abstract. It’s true they didn’t report the same KPI, but such an impressive result should be explained in detailed especially when compared to previous work contradicting it.

**Requested Changes:**

The following points should be addressed before acceptance:
* I strongly suggest to publish the code to reproduce the experiments to increase the impact of the publication. I think that's the most straightforward way to provide evidence for the superiority of forking sequence training, and could encourage library maintainers to adopt forking-sequences.
* I suggest to include reliability /coverage in the probabilistic evaluation also for the results in the main text, as CRPS alone is not enough to evaluate the goodness of the probabilistic forecasts.

Minor points
* I suggest to sharpen and increase a bit the specificity of the title with another couple of words, like “A theoretical and empirical study on forking-sequences for forecasting”.
* Gradient variance: the authors are referring to the gradient of loss w.r.t. model’s parameters at training time. I suggest to make it explicit once.
* “median forecast accuracy across datasets improves by up to 49.3%” please rephrase it, here and elsewhere, specifying you are referring to CRPS. Accuracy here could be intended as MAE.
* “rather than recognizing its connections with cross-validation and its simpler and more natural role in reducing variance during training”. What is reducing variance during training when using CV? Are you referring to ex-post variances of errors?
* healthcare care
* Notation like [h] = [1,...,h] is slightly confusing, suggest to replace with something like \mathcal{H} = [1,…,h]
* Notation for (5, 9, 10) is not clear. In (9) the index t is not mediated, loss L was never defined but if this is the argument of equation (5) no time averages are included, even if they should (there should be a Y[t][h] in (5)).  In (9)  L is both in the RHS and LHS.
* I think Fig 2 is not very helpful. Take fig 2a, as I understood from the explanation of window sampling, each y_t produces  an hidden state h_t (not h_[t, [h]]) and then only the last h_t (of hidden dimension d) is decoded into the forecast horizon  \hat{Y[t][h]}. Conversely, if I understood your distinction for window sampling and forking seq training, in the latter you pass non-independent, consecutive time steps of Yt during training. In fig.2 b it seems like you still divide in windows of size L.
* The manuscript is confusing on forecasts averaging: from figure 2 it seems that using the forking sequence approach the prediction is the average prediction made at different time steps (an average of “revisions”). But from loss 10 it doesn’t seem the case. Are the forecasts produced at different times for the same target timestamp averaged before entering the loss? If yes, I suggest an additional ablation study in which they’re not. Moreover, paragraph “forecast revision” seems to allude to using revisions to regularise learning, or at least, connections with previous and later paragraphs are not clear.
* “enriching the number of effective training signals extracted from each series”. Both approaches use a T*H number of targets (Y[t][h] per equation (2)) at the end if I understand correctly (total number of steps times total number of steps ahead). I.e. I assume window sampling uses an Hankel matrix. “while expanding the number of training losses by a facto of T” for the same reason seems incorrect.
* (17) uses the form of the loss for window sampling (9) and not (10). I don’t understand why.
* “We show that this architectural design yields a computational speedup factor of T” only with full window training, otherwise L
* It is not totally clear if results in table 2 and 3 for forking sequence refer to just the training mechanism or if these include also averaging.

---

> ### Author Response · Authors · 2026-03-09
>
> We appreciate the reviewer’s valuable comments and feedback. We have incorporated the suggestions in the revised paper and marked the changes in the paper in $\color{red}{\text{red}}$. We also would like to address and clarify each of the four consideration points mentioned in the reviewer’s summary:
>
> > (1) Window-based sampling (the standard in forecast literature) has also a linear variance reduction as a function of batch size, as stated by the authors
>
> > (2) Window-based sampling can produce identical model averaging as forking, as stated several times by the authors
>
> > (3) The computational complexity reduction is a trivial fact (window-based training increases it by factor L)
>
> > (4) Forecast stability is desirable but shouldn’t be optimised for, as forecasts naturally degrade as a function of step ahead, as stated by the authors
>
> Forking-sequences is analogous to teacher forcing, which is the standard training approach in NLP. Both methods reuse the model's internal computations to create dense supervision signals across the entire sequence at once. Despite being much more efficient than traditional window-sampling, forking-sequences remain relatively unknown in the forecasting field. Our work aims to bridge this gap by applying these efficient training principles to the specific needs of multi-horizon time-series forecasting.
>
> 1. **Variance reduction under window-based sampling.**
> We agree that window-based sampling achieves linear variance reduction as a function of batch size B. Our claim is not that forking-sequences design monopolizes this statistical property, but that it significantly improves computational–statistical trade-off. Reusing encoder computations to generate forecasts for all T forecast creation dates (FCDs) in a single forward pass, effectively increasing the supervision signal by a factor of T with negligible additional cost.
>
> 2. **Ensembling under window-sampling.**
> Window-based models can be ensembled. The distinction is again computational: forking-sequences architectures produce the full grid of forecasts in one forward pass, which makes temporal cross-validation and ensemble aggregation essentially free at inference time.
>
> 3. **On the perceived triviality of computational complexity improvements.**
> We apologize for the confusion about the computational gains. We improve Section 3.4 and Appendix E explanations. The improvements are massive from O(T^2) to O(T), and not O(T*L).
>
> 4. **Forecast stability desirability.**
> We agree that stability should not be optimized in isolation, however conditional on comparable accuracy, practitioners prefer forecasts that revise meaningfully rather than fluctuate erratically. Our goal is not to prioritize smoothness over accuracy, but to characterize the trade-off. As such one of our main contributions, the Excessive Volatility metric explicitly distinguishes beneficial forecast updates from detrimental revisions.

---

> > ### Author Response · Authors · 2026-03-09
> >
> > > Sensational results. “forking-sequences training scheme reduced sCRPS by 49.3%, on average across datasets, compared > to the window-sampling scheme confirming earlier observations by...” The three cited works show marginal improvements, > nowhere near 50% or close to the improvements reported in the abstract. It’s true they didn’t report the same KPI, but such > an impressive result should be explained in detailed especially when compared to previous work contradicting it.
> >
> > We appreciate the reviewer’s request for clarification regarding the magnitude of our results. In response, we have updated the manuscript to provide a more detailed description of the experimental outcomes, and explanation of the improvements.
> >
> > The main changes include:
> > -  We modified the results description in Section 4.2 differentiating the improvements by encoders:
> > “Consistent median gains across datasets are observed for RNN (46.2%), and CNN (28.6%) encoders, while the Transformer (14.7%) and State Space (6.4%) encoders showed smaller improvement.”
> > - We added a pointer in Section 4.2 to Figure 15a that visualizes the sCRPS accuracy improvements by encoder.
> > - We include Appendix B, where we compare gradient propagation across encoder types to explain the difference in performance gains. We find that forking-sequences alleviates the vanishing gradient issues that hinder LSTM encoders from learning effectively under window-sampling.
> >
> > We wish to clarify that prior research focusing on improved gradient propagation for LSTMs also reports substantial performance gains. For instance, "target replication" research (Lipton et al., 2016) reported macro F1 improvements of up to 17%. Furthermore, Dai & Le (2015) demonstrated that architectural innovations in LSTMs (SA-LSTM) could reduce test errors on datasets like DBpedia from 13.1% to 2.3%; this represents a relative error reduction of over 80%. These precedents confirm that substantial accuracy improvements are indeed attainable for LSTM architectures through modifications similar to the forking-sequences scheme. (https://arxiv.org/abs/1511.03677, https://arxiv.org/abs/1511.01432).
> >
> > ----
> >
> > **Requested Changes:**
> > > I strongly suggest to publish the code to reproduce the experiments to increase the impact of the publication. I think that's the most straightforward way to provide evidence for the superiority of forking sequence training, and could encourage library maintainers to adopt forking-sequences.
> >
> > Thank you for the recommendation. In the camera-ready version, we will provide links to the preprocessing, wrangling of public benchmark datasets, and baseline methods, ensembling techniques, sFPC and sEV metrics, and forecast pipelines to facilitate other researchers interested in the topic.
> >
> > We understand the importance of open access and are doing our best to make as much of the project available as possible.  However, we cannot commit to sharing all the project as it contains proprietary code and that would be considered illegal at this stage. The TMLR publication would certainly support the code sharing approval internal processes.
> >
> > ----
> >
> > > I suggest to include reliability /coverage in the probabilistic evaluation also for the results in the main text, as CRPS alone is not enough to evaluate the goodness of the probabilistic forecasts.
> >
> > We thank the reviewer for the suggestion. While CRPS is widely used for evaluating probabilistic forecasts, since it jointly captures calibration and sharpness, we agree that explicit coverage metrics provide additional insight into reliability. Accordingly, we have expanded the analysis of quantile coverage from Appendix L and added main results to the main text (Section 4.3, Figure 8).
> >
> > ----
> >
> > > I suggest to sharpen and increase a bit the specificity of the title with another couple of words, like “A theoretical and empirical study on forking-sequences for forecasting”.
> >
> > We have updated the title to reflect this recommendation, “Forking-Sequences for Time Series Forecasting”
> >
> > ----
> >
> > > Gradient variance: the authors are referring to the gradient of loss w.r.t. model’s parameters at training time. I suggest to make it explicit once.
> >
> > We appreciate the reviewer's suggestion. In the submission, in Appendix A we note that, “The gradient is computed with respect to the model parameters by minimizing Equation 5.” In the revision, we have updated section 3.3 to make this explicit, clarifying that this refers to gradients computed with respect to the model parameters during training.
> >
> > ----
> >
> > > “median forecast accuracy across datasets improves by up to 49.3%” please rephrase it, here and elsewhere, specifying you are referring to CRPS. Accuracy here could be intended as MAE.
> >
> > We have updated this sentence as follows, “We demonstrate that for MLP, RNN, LSTM, CNN, and Transformer-based architectures, median sCRPS across datasets improves by up to 49.3%.” as well as other instances in the paper where accuracy refers to sCRPS.

---

> > > ### Author Response · Authors · 2026-03-09
> > >
> > > **Requested Changes (continued)**
> > >
> > > > “rather than recognizing its connections with cross-validation and its simpler and more natural role in reducing variance during training”. What is reducing variance during training when using CV? Are you referring to ex-post variances of errors?
> > >
> > > We note that this sentence is unclear and the variance reduction refers to forking-sequences not CV. To clarify, we have revised the sentence this sentence as follows: “In some cases, its theoretical motivation was attributed to the martingale properties of forecast revisions (Chen et al., 2022; Foster and Stine, 2021), rather than its simpler and more natural role in reducing gradient variance during training.
> > >
> > > ----
> > >
> > > > healthcare care
> > >
> > > Thank you for noting this typo. We have removed the extra ‘care’ in this sentence.
> > >
> > > ----
> > >
> > > > Notation like [h] = [1,...,h] is slightly confusing, suggest to replace with something like \mathcal{H} = [1,…,h]
> > >
> > > We agree this notation should be made clearer and have updated the notation to reflect H as the horizon size: [h] = [1, …, H].
> > >
> > > ----
> > >
> > > > Notation for (5, 9, 10) is not clear. In (9) the index t is not mediated, loss L was never defined but if this is the argument of equation (5) no time averages are included, even if they should (there should be a Y[t][h] in (5)). In (9) L is both in the RHS and LHS.
> > >
> > > We appreciate the reviewer’s comment. We have removed the ‘t’ index from Equation 9 as this refers to a single set of multi-horizon predictions generated at a single FCD ‘t’. We have also changed L on the RHS to QL for Quantile Loss.
> > >
> > > ----
> > >
> > > > I think Fig 2 is not very helpful. Take fig 2a, as I understood from the explanation of window sampling, each y_t produces an hidden state h_t (not h_[t, [h]]) and then only the last h_t (of hidden dimension d) is decoded into the forecast horizon \hat{Y[t][h]}. Conversely, if I understood your distinction for window sampling and forking seq training, in the latter you pass non-independent, consecutive time steps of Yt during training. In fig.2 b it seems like you still divide in windows of size L.
> > >
> > > We thank the reviewer for this comment. To further clarify the diagram, we have explained the simplified example problem with T = 4, to prevent confusion that windows of size L are used in the figure. Regarding the [h] notation in  h_{t, [h]}, the ‘Encoder’ in equations 7 and 8 outputs a hidden state for each position in the forecast horizon, with the full hidden state tensor having shape B x T x H x D, where D denote the embedding dimension. We use [h] = [1, … H] to index the embedding corresponding to each horizon step h. We have expanded section 3.2 with this clarification.
> > >
> > > ----
> > >
> > > > The manuscript is confusing on forecasts averaging: from figure 2 it seems that using the forking sequence approach the prediction is the average prediction made at different time steps (an average of “revisions”). But from loss 10 it doesn’t seem the case. Are the forecasts produced at different times for the same target timestamp averaged before entering the loss? If yes, I suggest an additional ablation study in which they’re not. Moreover, paragraph “forecast revision” seems to allude to using revisions to regularise learning, or at least, connections with previous and later paragraphs are not clear.
> > >
> > > No, the T sets of multi-horizon forecasts produced using forking-sequences are not averaged before entering the loss. The factor T appears as a divisor (analogous to B and H) to ensure the training loss magnitude does not scale with the number of FCDs, keeping it comparable in magnitude to the window-sampling loss, which does not produce T sets of multi-horizon predictions.
> > >
> > > We do not have a paragraph titled 'forecast revision' and take this as a reference to section 3.6, but if we have misinterpreted this, we would be happy to clarify further. In section 3.6 we do not discuss using revisions to regularize learning. While we use this metric to assess forecast stability for the held-out test set, future work could leverage such metrics during training to regularize learning with respect to stability. We have expanded the discussion section to include this statement.

---

> > > > ### Author Response · Authors · 2026-03-09
> > > >
> > > > **Requested Changes (continued)**
> > > >
> > > > > “enriching the number of effective training signals extracted from each series”. Both approaches use a T*H number of targets (Y[t][h] per equation (2)) at the end if I understand correctly (total number of steps times total number of steps ahead). I.e. I assume window sampling uses an Hankel matrix. “while expanding the number of training losses by a facto of T” for the same reason seems incorrect.
> > > >
> > > > The reviewer is correct that forking-sequences produce T×H targets. However, the expansion we refer to is in the number of loss computations relative to window-sampling with fixed batch size B and horizon H. As shown by comparing equations 9 and 10, the forking-sequences gradient estimator differs from window-sampling by a factor of T. We have clarified this in the paper with the following revision, “Holding B and H fixed, forking-sequences, expands the number of train loss computations by a factor of T.”
> > > >
> > > > ----
> > > >
> > > > > (17) uses the form of the loss for window sampling (9) and not (10). I don’t understand why.
> > > >
> > > > Thank you for raising this typo. Yes equation 17 should reflect equation 10 and not 9. We have updated this in the paper.
> > > >
> > > > ----
> > > >
> > > > > “We show that this architectural design yields a computational speedup factor of T” only with full window training, otherwise L
> > > >
> > > > We would like to clarify that the forking-sequences architecture yields a computational speedup factor of T for window-sampling (both full and restricted variants), as shown in Table 1. T represents the number of FCDs in the entire series; to generate forecast outputs for each FCD using window-sampling, computation increases by a factor of T.
> > > >
> > > > ----
> > > >
> > > > > It is not totally clear if results in table 2 and 3 for forking sequence refer to just the training mechanism or if these include also averaging.
> > > >
> > > > The results in Tables 2 and 3 refer to the training mechanism: forking-sequences vs. window-sampling without ensembling during inference. To clarify this, we have revised Tables 2 and 3 to include forking-sequence ensembling results (FS-EN) in addition to forking-sequences (FS) and window-sampling (WS) results. We have also updated section 4 as follows:
> > > >
> > > > “We now evaluate the forking-sequences architecture in two separate but related studies: we first assess how forking-sequences impact model performance in comparison to window-sampling. We then assess how ensembling techniques impact forecast accuracy and stability. We present results for forking-sequences and window-sampling without ensembling as well as forking-sequences with ensembling in individual columns in Tables 2 and 3. Our main study aims to answer the following research questions: (i) Accuracy: Do models trained with forking-sequences outperform those trained with window-sampling? (ii) Stability: Do models trained with forking-sequences produce more stable forecasts than those trained with window-sampling? (iii) Ensemble impact on stability: To what extent do forking-sequences ensembles improve forecast stability? (iv) Accuracy–Stability Tradeoff: Can ensembling enhance stability without degrading accuracy? (v) Architecture Interactions: How do the benefits of forking-sequences vary across different encoder architectures? We also consider ablations to assess various ensembling methods and their impact for zero-shot models: (vi) Ablations: How do alternative ensembling techniques affect accuracy and stability, and how does forking-sequences mitigate vanishing gradients? Can we extend forking-sequences ensembling to work with established architectures such as NBEATS and PatchTST?"

---

> > > > > ### Comment · Reviewer_3s2u · 2026-03-10
> > > > >
> > > > > Dear authors, thank you very much for your replies. I’ve carefully read them as well as the reviewed paper. I would kindly ask to consider the following points.
> > > > >
> > > > > * I’m sorry to insist about the clarity of the training process, but if no code for the training loop is going to be published I may raise another question on pseudocode Algorithm 1 and eq 9 and 10. By comparing eq. 9 and eq 10, one would conclude that the forking sequence loss over one batch already processes the full dataset. If it is the case, that would (partially) explain figure 3 losses decaying much faster for forking sequences. Without access to the training code this would remain unclear. At the same time, in algorithm 1, The notation “Y[b][t][h] or single targets Y[b],t,[h]” seems to indicate that for forking sequences, each batch b contains the full target set, Yb[t][h]. Can you clarify if it is the case?
> > > > >
> > > > > * Coverage comparison: you added figure 8, but this investigates how coverage changes w.r.t. window size. The requested comparison was coverage change with or without forking sequences, ideally as in table 2 and 3.
> > > > >
> > > > > I kindly ask you to address these points

---

> > > > > > ### Author Response · Authors · 2026-04-01
> > > > > >
> > > > > > Dear Reviewer, thank you for reviewing our responses and for the follow-up clarification questions, which have helped us improve the quality of the manuscript. We have addressed each question below.
> > > > > >
> > > > > > ---
> > > > > >
> > > > > > > I’m sorry to insist about the clarity of the training process, but if no code for the training loop is going to be published I may raise another question on pseudocode Algorithm 1 and eq 9 and 10. By comparing eq. 9 and eq 10, one would conclude that the forking sequence loss over one batch already processes the full dataset. If it is the case, that would (partially) explain figure 3 losses decaying much faster for forking sequences. Without access to the training code this would remain unclear. At the same time, in algorithm 1, The notation “Y[b][t][h] or single targets Y[b],t,[h]” seems to indicate that for forking sequences, each batch b contains the full target set, Yb[t][h]. Can you clarify if it is the case?
> > > > > >
> > > > > > We thank the reviewer for recommending the inclusion of training code. In the camera-ready version, we will release a training pipeline alongside the preprocessing and dataset wrangling scripts for the public benchmark datasets, baseline methods, ensembling techniques, sQPC and sEV metric implementations, and forecast pipelines described in our first response.
> > > > > >
> > > > > > Yes, the forking-sequence loss over a single batch processes the full series, producing predictions for all T forecast creation dates (FCDs) at once versus just 1 FCD per batch under window-sampling.  We use the notation $Y_t$ to denote targets for a single FCD and $Y_{[t]}$ to denote targets for all FCDs with $[t]=[1, …, T]$. For forking-sequences, each gradient update is therefore informed by loss contributions from all T multi-horizon predictions (normalized by $T$), compared to a single multi-horizon prediction under window-sampling
> > > > > >
> > > > > >
> > > > > > ---
> > > > > >
> > > > > > > Coverage comparison: you added figure 8, but this investigates how coverage changes w.r.t. window size. The requested comparison was coverage change with or without forking sequences, ideally as in table 2 and 3.
> > > > > >
> > > > > > Thank you for clarifying. We have added a comparison of the effect of forking-sequences and window-sampling on coverage, measured with average coverage error, in Appendix L.1. We have also added a summary of these results in our main paper (section 4.2),
> > > > > >
> > > > > > “Forking-sequences significantly improves upon window-sampling calibration, measured with average coverage error (ACE) as shown in Appendix L.1. We observe median percentage improvements across datasets of 35.9%, 66.3%, 61.3%, 61.5%, 43.9%, and 17.6% for MLP, RNN, LSTM, CNN, Transformer, and S4 encoders, respectively. We note that the additional ACE improvement from ensembling during inference over forking-sequences without ensembling is relatively marginal, with median percentage improvements across datasets of 0.3%, 0.0%, 0.0%, 0.5%, -1.1%, 0.1% for MLP, RNN, LSTM, CNN, Transformer, and S4 encoders.”

---

> > > > > > > ### Comment · Reviewer_3s2u · 2026-04-02
> > > > > > >
> > > > > > > Dear authors,
> > > > > > > thank you for your reply.
> > > > > > > Publishing the training pipeline would be of great help in reproducibility studies.
> > > > > > > I find the other replies convincing, so you have the green light from my side.

---

### Review · Reviewer_Bg4t · 2026-02-20

**Summary Of Contributions:**

This paper studies forecast stability across forecast creation dates (FCDs) as an important yet underexplored objective in multi-horizon time series forecasting. The authors argue that, in many real-world applications, decision makers rely on sequences of forecast revisions rather than single forecast snapshots, and therefore stability across updates is a critical property alongside predictive accuracy.
The core technical contribution is the formalization and systematic study of the forking-sequences paradigm, an encoder-agnostic architectural and training framework in which a model jointly processes all forecast creation dates in a single forward pass, producing an entire multi-horizon forecast grid. This contrasts with conventional window-sampling approaches, which treat each forecast date independently and require repeated encoder computation. The paper provides theoretical and empirical motivation for this paradigm, which has previously been used heuristically in some neural forecasting systems.

**Key strengths**
- The paper addresses a practically relevant and underexplored problem in forecasting: the trade-off between accuracy and stability across forecast revisions.
- It provides a well-rounded treatment combining theory, architecture, metrics, and experiments.
- The proposed EV metric is conceptually appealing and operationally meaningful.
- The empirical evaluation is broad and systematic, and the appendices provide useful theoretical and implementation details.

**Key weaknesses**
- Some empirical findings are more nuanced than the main claims, as stability improvements vary by encoder architecture and baseline comparisons.
- The main stability metric depends on ground truth, which limits interpretability in real-time decision settings where the truth is unknown at revision time.
- The empirical focus is largely on controlled architectures and univariate settings; broader validation (e.g., multivariate and foundation models) remains an open direction.

**Audience:**

Yes

**Audience Explanation:**

This work appears to fit squarely within the kinds of contributions TMLR explicitly solicits. Particularly: (i) “new algorithms with sound empirical validation,” (ii) “experimental and/or theoretical studies yielding new insight into the design and behavior of learning in intelligent systems,” and (iii) “formalization of new learning tasks and of methods for assessing performance on those tasks".

**Broader Impact Concerns:**

The work is primarily methodological, and there are no ethical concerns.

**Claims And Evidence:**

Yes

**Claims Explanation:**

Yes. The paper’s main claims are generally supported by a strong combination of theoretical analysis and empirical validation. The authors provide formal arguments and appendix proofs showing that averaging gradients across multiple forecast creation dates reduces variance under mild dependence assumptions, and they corroborate this with synthetic experiments demonstrating the expected variance decay. The claim that recurrent encoders benefit more from the approach is backed by both analytical comparisons of gradient propagation and empirical studies of gradient norms and signal-to-noise ratios across architectures. Computational efficiency is supported through explicit complexity analysis and wall-clock experiments comparing full and restricted window-sampling with the proposed method. The new stability metric (EV) is rigorously defined and accompanied by proofs of its key properties, which strengthens its credibility beyond a purely heuristic proposal. In addition, the empirical results are extensive, covering multiple large benchmark datasets with repeated runs, uncertainty reporting, and ablation studies that reveal meaningful accuracy–stability trade-offs rather than overstating gains. While some findings, such as stability improvements across all encoder families and the role of increased supervision, would benefit from clearer interpretation and more tightly controlled comparisons, the overall evidentiary support is solid and meets TMLR’s standards

**Requested Changes:**

**Critical changes**

The paper should more explicitly disambiguate (and, where needed, temper) stability claims by clearly separating three comparisons: (a) forking-sequences vs window-sampling without ensembling, (b) forking-sequences vs window-sampling with ensembling, and (c) within-model improvement from adding ensembling during inference. Some reported stability outcomes vary by encoder family and comparison baseline; the main text should surface these nuances prominently (rather than leaving a reader to infer them from appendix figures/tables).

The methods section would benefit from a clearer, more implementation-oriented exposition of (i) the exact construction of the forecast grid, (ii) the masking strategy used to prevent leakage across train/validation/test when training on many FCDs, and (iii) how loss terms are sampled/weighted across time for both window-sampling and forking-sequences. This is important because the paradigm changes the effective number of supervised targets per series, which affects both optimization and generalization. A short pseudocode block (or concise algorithm description) would likely resolve ambiguity.

A “matched supervision / matched compute” ablation is strongly recommended. One concrete option: compare (1) forking-sequences training to (2) a window-sampling variant that processes multiple windows per series per step (or uses gradient accumulation) so that both approaches see a comparable number of loss terms, then report accuracy/stability vs compute. This would sharpen the causal claim that the gains stem from the architectural/training structure (and encoder reuse), not only from increased supervision density.

Reproducibility would be improved by explicitly stating whether code (or at least a fully specified evaluation pipeline) will be released. The appendix provides hyperparameter search ranges and compute environment details, which is helpful, but readers would still benefit from (i) exact dataset preprocessing scripts/splits, (ii) random seed conventions, and (iii) a precise definition of what constitutes an FCD set for each dataset frequency in the expanded cross-validation evaluation. If code cannot be released, the paper should compensate with enough procedural detail to enable independent reproduction.

**Changes that would strengthen the work**

The paper should better motivate how EV/sEV relates to stakeholder notions of “stability.” While EV’s ground-truth dependence is a feature for retrospective evaluation (rewarding improving revisions), practitioners may also care about revision magnitude irrespective of eventual truth, or about stability under distribution shifts. A brief discussion of what EV captures well vs what it intentionally does not would preempt misinterpretation and position sQPC/sEV as complementary rather than competing constructs.

Given that ensembling/smoothing can reduce responsiveness to real regime changes, it would strengthen the paper to include a targeted “shock” or “structural break” study (even synthetic) evaluating whether increased stability comes at the cost of delayed adaptation, especially if the intended deployment domains include high-stakes planning contexts.

---

> ### Author Response · Authors · 2026-03-09
>
> > Key strengths
> > - The paper addresses a practically relevant and underexplored problem in forecasting: the trade-off between accuracy and stability across forecast revisions.
> > - It provides a well-rounded treatment combining theory, architecture, metrics, and experiments.
> > - The proposed EV metric is conceptually appealing and operationally meaningful.
> > - The empirical evaluation is broad and systematic, and the appendices provide useful theoretical and implementation details.
>
> We are pleased the reviewer found our work addresses a practically relevant and underexplored problem, with a well-rounded treatment across theory, architecture, metrics, and experiments, and that the EV metric and empirical evaluation were found to be meaningful and systematic. We have incorporated the suggestions in the revised paper and marked the changes in the paper in $\color{red}{\text{red}}$.
>
> > Key weaknesses
> > - Some empirical findings are more nuanced than the main claims, as stability improvements vary by encoder architecture and baseline comparisons.
> > - The main stability metric depends on ground truth, which limits interpretability in real-time decision settings where the truth is unknown at revision time.
> > - The empirical focus is largely on controlled architectures and univariate settings; broader validation (e.g., multivariate and foundation models) remains an open direction.
>
> We appreciate the reviewer’s feedback and have addressed each point as follows:
> - While we discuss the variation in performance gains between encoder architectures in section 4.3, and the difference between Transformer and LSTM encoders in a dedicated gradient-based study (Appendix B), we agree that the same nuance should be applied to the stability results. We have updated the introduction contributions and results section 4.2 to clarify that both performance and stability improvements vary by encoder architecture and baseline comparisons.
> - We thank the reviewer for the insightful comment on the ground truth limitation for the computation of the EV metric.  Our work preventively includes a second metric, Scaled Quantile Percentage Change (sQPC), to assess forecast stability without requiring ground truth, making it suitable for real-time settings. We renamed sQPC into sFPC, since the metric is a point forecast metric and have expanded section 5 with the following to clarify sEV must be used retroactively unlike sFPC, “While sFPC measures the raw revision magnitude of point forecasts without requiring ground truth, sEV retroactively assesses the stability of probabilistic forecasts by distinguishing accuracy-improving revisions from detrimental ones.”
> - We deliberately dedicate this work analysis to the univariate setting in order to better control for, and more clearly understand, the behavior of the forking sequences technique. The univariate results in this paper have already been used as a guiding principle for the design of other neural forecast architectures (MQCNN, SPADE, MQT), however the theoretical and empirical analysis of the extension towards the multivariate setting would merit its own dedicated study.
>
> ----
>
> **Requested Changes**
>
> > The paper should more explicitly disambiguate (and, where needed, temper) stability claims by clearly separating three comparisons: (a) forking-sequences vs window-sampling without ensembling, (b) forking-sequences vs window-sampling with ensembling, and (c) within-model improvement from adding ensembling during inference. Some reported stability outcomes vary by encoder family and comparison baseline; the main text should surface these nuances prominently (rather than leaving a reader to infer them from appendix figures/tables).
>
> We appreciate the comments. We updated Section 4, to identify with greater clarity the effects of forking-sequences, and the ensemble. In particular we updated the main results in Table 2 and Table 3 with sCRPS and EV metrics, to also include a column with the forking-sequences ensemble results (EN).
>
> The improvements to Table 2, and Table 3 allow for readers to observe directly (a) the accuracy/stability effects of forking-sequences vs window-sampling without ensembling, (b/c) the accuracy/stability effects of ensemble on forking-sequences architectures. For the effects of the ensemble on window-sampling architectures we improved the description of Section 4.3 with the ensemble ablation study for NBEATS and PatchTST windows-sampling architectures.
>
> Moreover, throughout the paper we also modified the description of our main and ablation study results to highlight the nuances of the improvements, in particular we acknowledged the difference of accuracy and stability improvements across different encoders.

---

> > ### Author Response · Authors · 2026-03-09
> >
> > **Requested Changes (continued)**
> >
> > > The methods section would benefit from a clearer, more implementation-oriented exposition of (i) the exact construction of the forecast grid, (ii) the masking strategy used to prevent leakage across train/validation/test when training on many FCDs, and (iii) how loss terms are sampled/weighted across time for both window-sampling and forking-sequences. This is important because the paradigm changes the effective number of supervised targets per series, which affects both optimization and generalization. A short pseudocode block (or concise algorithm description) would likely resolve ambiguity.
> >
> > We thank the reviewer for this helpful suggestion. We agree that a more implementation-oriented exposition clarifies how the proposed paradigm affects both optimization and generalization.
> >
> > To address this, we have added a dedicated Appendix A, where we: (i) Provide a precise description of the construction of the forecast grid; (ii) Detail the masking strategy used to prevent temporal leakage across train/validation/test splits when training on multiple FCDs; (iii) Explicitly describe how loss terms are sampled and weighted across time under both the window-sampling and forking-sequences paradigms.
> >
> > In addition, Appendix A now includes Algorithm 1, a concise pseudocode-style description of the training procedures. The algorithm explicitly distinguishes the number of supervised targets per series contributing to each gradient update under the two architectural designs.
> >
> > ----
> >
> > > A “matched supervision / matched compute” ablation is strongly recommended. One concrete option: compare (1) forking-sequences training to (2) a window-sampling variant that processes multiple windows per series per step (or uses gradient accumulation) so that both approaches see a comparable number of loss terms, then report accuracy/stability vs compute. This would sharpen the causal claim that the gains stem from the architectural/training structure (and encoder reuse), not only from increased supervision density.
> >
> > We appreciate the comment. We agree that, in principle, a window-sampling approach could also reduce SGD variance by increasing the number of sampled windows per series (e.g., via multiple windows per step or gradient accumulation), thereby matching the effective number of loss terms. Our claim is not that variance reduction is unique to forking-sequences, but rather that it achieves this effect more efficiently.
> >
> > As discussed in Section 3.4 and Appendix E, the computational advantage of forking-sequences stems from encoder reuse: all FCDs associated with a series share the same encoder forward pass, so supervision density increases without proportionally increasing encoder computation. Moreover, no intermediate information needs to be stored across forward passes. In contrast, matching supervision density with window-sampling requires additional forward passes (or accumulated gradients), leading to higher computational cost. In summary the architectural benefits come from encoder computation reuse, rather than supervision density.
> >
> > ----
> >
> > > Reproducibility would be improved by explicitly stating whether code (or at least a fully specified evaluation pipeline) will be released. The appendix provides hyperparameter search ranges and compute environment details, which is helpful, but readers would still benefit from (i) exact dataset preprocessing scripts/splits, (ii) random seed conventions, and (iii) a precise definition of what constitutes an FCD set for each dataset frequency in the expanded cross-validation evaluation. If code cannot be released, the paper should compensate with enough procedural detail to enable independent reproduction.
> >
> > Thank you for the recommendation. In the camera-ready version, we will provide links to the preprocessing, wrangling of public benchmark datasets, and baseline methods, ensembling techniques, sQPC and sEV metrics, and forecast pipelines to facilitate other researchers interested in the topic.
> >
> > We understand the importance of open access and are doing our best to make as much of the project available as possible.  However, we cannot commit to sharing all the project as it contains proprietary  code and that would be considered illegal at this stage. The TMLR publication would certainly support the code sharing approval internal processes.

---

> > > ### Author Response · Authors · 2026-03-09
> > >
> > > **Requested Changes (continued)**
> > >
> > > > The paper should better motivate how EV/sEV relates to stakeholder notions of “stability.” While EV’s ground-truth dependence is a feature for retrospective evaluation (rewarding improving revisions), practitioners may also care about revision magnitude irrespective of eventual truth, or about stability under distribution shifts. A brief discussion of what EV captures well vs what it intentionally does not would preempt misinterpretation and position sQPC/sEV as complementary rather than competing constructs.
> > >
> > > We appreciate the suggestion. We renamed sQPC into sFPC, since the metric is a point forecast metric. While EV is a probabilistic forecast metric. The metrics serve a different and complementary purpose. We improved the contribution paragraph (iv) that now reads: "We introduce two complementary metrics for assessing forecast stability across FCDs: the Symmetric Forecast Percentage Change (sFPC), which measures raw point forecasts' revision magnitude independently of ground truth, andScaled Excess Volatility (sEV), that retroactively measures stability of probabilistic forecasts, distinguishing accuracy-improving revisions from detrimental ones."
> > >
> > > Additionally we added the following discussion paragraph on limitations of sEV in Section 5: "Our two metrics serve as complementary diagnostic tools: while sFPC measures the raw revision magnitude of point forecasts without requiring ground truth, sEV retroactively assesses the stability of probabilistic forecasts by distinguishing accuracy-improving revisions from detrimental ones. Although sEV is a compelling metric, it remains to be determined whether it satisfies the conditions of a strictly proper scoring rule~\cite{gneiting2007strictly} or if it could be artificially improved through oversmoothing at the expense of sharpness."
> > >
> > > ----
> > >
> > > > Given that ensembling/smoothing can reduce responsiveness to real regime changes, it would strengthen the paper to include a targeted “shock” or “structural break” study (even synthetic) evaluating whether increased stability comes at the cost of delayed adaptation, especially if the intended deployment domains include high-stakes planning contexts.
> > >
> > > We thank the reviewer for this thoughtful suggestion. We agree that there is an inherent tension between forecast stability and adaptability, particularly in the presence of regime changes or structural breaks. In the paper, we already investigate this trade-off through an ablation study of different ensembling strategies, showing that excessive smoothing can hinder the model’s ability to incorporate recent information and may therefore reduce accuracy.
> > >
> > > While we do not include a dedicated synthetic “shock” or structural break experiment, our results demonstrate that increasing stability through stronger ensembling can indeed slow adaptation. To make this clearer, we have revised the introduction to the ablation subsection (Section 4.3) to explicitly highlight this trade-off. The revised text now reads: “Our ablation study in Appendix I highlights the trade-off between forecast stability and accuracy under different ensembling strategies, showing that stronger ensemble smoothing can improve stability at the cost of slower adaptation to new information.”

---

### Review · Reviewer_eHht · 2026-04-27

**Summary Of Contributions:**

This manuscript introduces the Forking Sequences approach, where multiple horizons are considered simultaneously, and a new penalization term entitled "Excess Volatility" is introduced to reduce variability in forecasts when models are updated.  The Excess Volatility term has some nice properties that make it appealing as elaborated on Theorem 2.  Empirical results suggest that the proposed technique improves on empirical real-world data problems.

**Audience:**

Yes

**Audience Explanation:**

Time-series forecasting is a large problem, and often risk is impacted by probabilistic forecasts.  This proposed approach helps reduce stochasticity in the forecasts while also improving uncertainty measures.  The techniques are straightforward and could be included in broader problems.

**Broader Impact Concerns:**

N/A.

**Claims And Evidence:**

No

**Claims Explanation:**

Most claims are proven and empirical evidence is documented; minor issues persist as described below.

**Requested Changes:**

While this revision has addressed a lot of potential issues from prior reviewers, I have two main remaining issues.

First, the computational efficiency claims in seems to slightly mischaracterize the literature.  It is rare to use the WS (Full) approach on long sequences, which makes the claim of "orders-of-magnitude" improvement imprecise. Instead, the claim is more accurately comparing FS and WS (Restricted).  However, even there, the results seemed to be overclaimed.  It is common in time-series modeling to process and share weights during learning, such has been common in autoregressive models for decades.  While undoubtedly there are approaches that use WS (Restricted) and don't share information, it is tough to consider this as a real contribution from this manuscript.  I would suggest scaling back the claims here, and instead focusing more on your current contributions of (i) and (iii), which are clearer.

For my other issue, Theorem 1's real impact is difficult to follow in the context of the contribution.  In particular, there is the remark that this reduces variance by O(1/T); that is true for (10) under mild assumptions. However, it hides the real impact as it is simply reducing variance by the division by T, and what really matters for the effective of the optimization is the improvement on the effective signal to noise.  Additionally, I was surprised by the claim that you're converging to the true gradient from a minibatch; however, looking through the theorem, it appears that the mean is going to zero because of this division term over T, and thus the convergence happens because they both converge to zero.  As such, the details on the use of Theorem 1 needs to be put into context and the claims reduced.  If my understanding is not accurate, I would encourage you to write a more complete and detailed proof of the theorem.

---

> ### Author Response · Authors · 2026-05-10
>
> We are pleased the reviewer found our work addresses a relevant problem within forecasting, with proven claims and documented empirical evidence, and that our work has potential to address broader problems. We are also glad that the reviewer found our revision addressed most potential issues, with only two remaining concerns. We have addressed these points below, with corresponding manuscript revisions in blue.
>
> ---
>
> >First, the computational efficiency claims in seems to slightly mischaracterize the literature. It is rare to use the WS (Full) approach on long sequences, which makes the claim of "orders-of-magnitude" improvement imprecise. Instead, the claim is more accurately comparing FS and WS (Restricted). However, even there, the results seemed to be overclaimed. It is common in time-series modeling to process and share weights during learning, such has been common in autoregressive models for decades. While undoubtedly there are approaches that use WS (Restricted) and don't share information, it is tough to consider this as a real contribution from this manuscript. I would suggest scaling back the claims here, and instead focusing more on your current contributions of (i) and (iii), which are clearer.
>
> We thank the reviewer for such careful observation and feedback and we clarified/scaled the comparison between FS, WS-restricted and WS-full in our efficiency claims.
>
> The contribution paragraph in the introduction now reads:
> “We show that forking-sequences enable a much more efficient computational regime for encoder–decoder architectures by supporting shared inference across FCDs in a single forward pass. Unlike standard approaches that recompute the encoder for each FCD, forking-sequences reuse encoder outputs across all FCDs, thereby eliminating redundant computation. In the context restricted setting, where each FCD uses an L-length context, complexity improves by a factor of L (O(LT) to O(T)). In the context-unrestricted setting, complexity reduces from quadratic to linear (O(T^2) to O(T)).”
>
> ---
>
> > For my other issue, Theorem 1's real impact is difficult to follow in the context of the contribution. In particular, there is the remark that this reduces variance by O(1/T); that is true for (10) under mild assumptions. However, it hides the real impact as it is simply reducing variance by the division by T, and what really matters for the effective of the optimization is the improvement on the effective signal to noise. Additionally, I was surprised by the claim that you're converging to the true gradient from a minibatch; however, looking through the theorem, it appears that the mean is going to zero because of this division term over T, and thus the convergence happens because they both converge to zero. As such, the details on the use of Theorem 1 needs to be put into context and the claims reduced. If my understanding is not accurate, I would encourage you to write a more complete and detailed proof of the theorem.
>
> First we greatly appreciate this suggestion! We agree that the O(1/T) variance reduction in Theorem 1 follows from standard averaging arguments (e.g., Law of Large Numbers), and by itself does not capture FS practical impact. FS impact is better measured in the effective signal-to-noise ratio (SNR) of the gradient estimator: averaging T samples preserves the expected gradient while reducing noise variance by 1/T, yielding an SNR improvement that scales linearly in T. We improved Theorem 1, from being a Chebyshev probability bound for the gradient errors, to be a Theorem about the SNR estimation quality. Which much better aligns with the results in our paper.
>
> Second we apologize for any confusions from Theorem 1. The division by $T$ in the gradient estimator from Equation 10 does not cause the gradient to converge to zero. Under our stationarity assumption, each FCD gradient sample satisfies $\mathbb{E}[\nabla L_t] = \mu$, so the estimator $\bar{\nabla}L_T = \frac{1}{T}\sum_{t=1}^T \nabla L_t$ remains **centered around the gradient** $\mu$ for all FCDs $T$, i.e. $\mathbb{E}[\bar{\nabla}L_T] = \mu$.
>
> Our result from Lemma1, states that the estimator’s variance reduces by a $1/T$ factor, the 1/T factor does not affect the mean. As $T$ increases, the noise around $\mu$ diminishes (at rate $\mathcal{O}(1/T)$ under M-dependence), which leads to convergence in probability to $\mu$, and the SNR improving linearly.  We revised the text to make this distinction clearer.

---

### Author Response · Authors · 2026-03-09

Dear Editor,

We are pleased that the Reviewers believe our work addresses a practically relevant and underexplored problem in forecasting and that our findings would be of interest to TMLR’s audience. We have addressed all points raised in the reviews and updated the paper with changes in $\color{red}{\text{red}}$. We hope that you find the current version ready to be published.

In response to the feedback, we have taken the opportunity to improve the paper’s discussion and presentation of the main experiments and results. The key changes in the updated manuscript are:

1. We provided a clearer presentation of results among the three encoder-agnostic model variants: forking-sequences and window-sampling without ensembling and forking-sequences with ensembling. We extended Tables 2 and 3 to include forking-sequences (with ensembling) results in addition to forking-sequences and windows sampling (without ensembling).

2. We more explicitly report percentage changes in both accuracy and stability metrics for each encoder type as well as reference distribution plots of performance across datasets to better emphasize nuanced performance differences across encoder types and datasets (Sections 4, 4.2 and 4.3, Figures 11, 14, 15).

3. We added Appendix A to outline our forking-sequences implementation more clearly, including the exact construction of the forecast grid, the masking strategy used to prevent leakage across train/validation/test sets (illustrated in a supporting figure), and how loss terms are computed (with a supporting algorithm).

4. We expanded the discussion of variance reduction with regard to alternative approaches such as window-sampling with several forward passes or increasing batch size, to clarify that this effect is not unique to forking-sequences, but rather that forking-sequences achieves variance reduction more efficiently (Section 3.3).

5. We expanded the coverage analysis from Appendix L and added key results to the main text (Section 4.3, Figure 8) to support additional evaluations of probabilistic forecasts in addition to sCRPS and sEV metrics.

6. We expanded the discussion on the sEV metric to clarify its limitations, including that it requires retroactive assessment unlike our complementary metric, sFPC (Section 5).

Sincerely yours,

The Authors

---

### Decision · Action_Editor_uo4P · 2026-06-04

**Recommendation:** Accept as is

**Audience:**

Yes

**Audience Explanation:**

The reviewers unanimously agree that the findings of this paper would be of interest to at least some individuals, particularly those in the time-series forecasting community. They consistently highlight the practical relevance of the problem being addressed (forecast stability) and the potential for the proposed methods to be adopted by other researchers and practitioners in the field of forecasting.

**Claims And Evidence:**

Yes

**Claims Explanation:**

In the initial reviews, two reviewers answered "No" to this question, pointing out a couple of issues. For example, it was felt that the claims about computational efficiency were slightly mischaracterized and that the impact of Theorem 1 was not clearly explained. Strong skepticism was also expressed about the sensational results which seemed to contradict previous work. The authors provided detailed responses to the reviewers' concerns through a series of exchanges and also revised the paper accordingly. The authors successfully addressed the initial concerns through discussions and revisions, leading to a final consensus that the paper's claims are well-supported by evidence.